# The Pléiades Glacier Observatory: high resolution digital elevation models and ortho-imagery to monitor glacier change

Etienne Berthier[1], Jérôme Lebreton[1], Delphine Fontannaz[2], Steven Hosford[2], Joaquin M. C. Belart[3], Fanny Brun[4], Liss M. Andreassen[5], Brian Menounos[6,7], Charlotte Blondel[1]

[1] Université de Toulouse, LEGOS (CNES/CNRS/IRD/UT3), Toulouse, France

[2] Centre National d'Etudes Spatiales, Toulouse, France

[3] National Land Survey of Iceland, Akranes, Iceland

[4] IGE, Université Grenoble Alpes, CNRS, IRD, Grenoble INP, Grenoble, France

[5] Section for Glaciers, Ice and Snow, the Norwegian Water Resources and Energy Directorate (NVE), Oslo, Norway

[6] University of Northern British Columbia, Prince George, BC, Canada

[7] Hakai Institute, Campbell River, BC, Canada

*Correspondence to*: Etienne Berthier (etienne.berthier@univ-tlse3.fr)

*Abstract*. Spaceborne digital elevation models (DEMs) of glaciers are essential to describe their health, and their contribution to river runoff and to sea level rise. Publicly available DEMs derived from sub-meter satellite stereo-imagery were, up to now, mainly available in the polar regions and High Mountain Asia. Here, we present the Pléiades Glacier Observatory (PGO), a scientific programme acquiring Pléiades 0.7-m satellite stereo pairs for 140 sites from Earth's glacierized areas. The PGO product consists of freely-available DEMs at 2 m and 20 m ground sampling distance together with 0.5 m (panchromatic) and 2 m (multispectral) ortho-images. PGO stereo acquisitions began in July 2016 in the North Hemisphere and February 2017 in the South Hemisphere. Each site is revisited every five years (cloud permitting), close to the end of the melt season, to measure glacier elevation change with an average uncertainty of 0.49 m (95% confidence level, for a glacierized area of 1 km²), i.e. 0.1 m a$^{-1}$. PGO samples over 20,000 km² of glacierized terrain which represents about 3% of the Earth's glaciers area. This small sample, however, provides a first order estimate (within 0.07 m w.e./yr) of the global glacier mass change and its decadal evolution.

# 1. Introduction

Over the last two decades, the increase in spaceborne satellite imagery archives accelerated our ability to quantify glacier change (Pope et al., 2014; Berthier et al., 2023). Distribution of medium (10-30 m) resolution satellite archives (e.g., from Landsat, Advanced Spaceborne Thermal Emission and Reflection Radiometer - ASTER) and the open nature of new missions (e.g. the Sentinels from Copernicus), for example, provided imagery to construct improved glacier inventories (Pfeffer et al., 2014; RGI 7.0 Consortium, 2023), spatiotemporal analysis of glacier velocity (Millan et al., 2022) and elevation change (Hugonnet et al., 2021). These global observational products of glacier change are important calibration data to improve projections of future glacier mass change (Rounce et al., 2023).

Glaciology has also benefited from the use of very-high-resolution (VHR, i.e., sub-meter) optical sensors. Contrary to medium-resolution satellite missions, present-day very high resolution satellite missions do not allow a frequent and continuous global survey of the Earth's glaciers, but these missions are advantageous in a number of ways. The ability to quickly task these satellites provides a means for rapid response following natural disasters (Shugar et al., 2021; Kääb et al., 2021). Their sub-meter resolution translates into superior derived products (e.g. glacier outline, velocity, elevation, snow-line elevation) compared to those obtained from medium resolution imagery. This improved quality is needed to study fine scale processes (Sato et al., 2021; Brun et al., 2016; Loriaux and Ruiz, 2021), monitor small glaciers (Małecki, 2022), validate similar products derived from coarser images (Andreassen et al., 2022) and also calibrate glaciological mass balance measured in the field (Zemp et al., 2013; Wagnon et al., 2021; Andreassen et al., 2016). With the notable exceptions of the polar regions (Howat et al., 2019; Porter et al., 2018) and High Mountain Asia (Shean et al., 2020), however, access to this very high resolution data has remained limited for the glaciological community.

This article presents the Pléiades Glacier Observatory (PGO), an initiative by the French Space Agency (CNES) and the Laboratoire d'Etudes en Géophysique et Océanographie Spatiales (LEGOS) to facilitate access to very high resolution data (digital elevation models – DEMs – and ortho-imagery) from the Pléiades satellites. We present the coverage achieved since 2016 for 140 PGO glacierized sites and describe how the freely-available products are derived from Pléiades stereo-images. We also assess the quality of the PGO DEMs using near-contemporaneous accurate airborne laser scanning data in Norway and western Canada and evaluate the precision of the elevation change maps that are derived every five years. We conclude by considering how representative the geodetic mass balance derived for these PGO sites is for Earth's glaciers.

# 2. Design of the PGO project

## 2.1. Pléiades 1A and 1B satellites for glacier monitoring

CNES and Airbus Defense and Space respectively designed and operates the optical satellites Pléiades 1A and 1B (Gleyzes et al., 2012). Pléiades 1A was launched 17 December 2011 and 1B 2 December 2012. The image resolution of the panchromatic and multi-spectral bands are respectively initially 0.7 m and 2.8 m, then resampled by the ground segment to 0.5 m and 2 m. Pléiades images have a ~20 km swath, relatively large compared to other VHR satellites (e.g., 13 km for WorldView-3). In order to derive DEMs, stereo images can be acquired in an along-track pair about 40 seconds apart. Compared to earlier stereo

sensors (SPOT5-HRS, ALOS-PRISM and TERRA-ASTER Visible and Near-Infrared - VNIR), a clear advantage
for snow and ice monitoring is 12-bit encoding of the sensor (4096 grey levels) which significantly
increases the image contrast (Berthier et al., 2023).

Early results on several glaciers showed the usefulness of Pléiades data for measuring their topography
and its change with time (Berthier et al., 2014; Holzer et al., 2015). The 1-sigma uncertainty of these
74 Pléiades DEMs is about 1 m over gently sloping areas (Błaszczyk et al., 2019; Berthier et al., 2014). This
level of uncertainty is adequate to measure elevation changes, often exceeding several metres, at
76 seasonal (Belart et al., 2017; Beraud et al., 2023) to inter-annual (Bhattacharya et al., 2021) time scales.

Airbus operates Pléiades 1A and 1B commercially which does not include building a comprehensive
archive of images, at least not for glaciers. Furthermore, access to the data is difficult and cost
prohibitive, especially for users outside of the European Union. These challenges led us to initiate the
80 PGO program in 2016 as a way to monitor a selection of glacier sites around the globe and facilitate
access for the international glaciological community.

Despite the 12-bits encoding of the images, we observed saturated pixels for early Pléiades images
(2011–2015) on illuminated slopes (facing toward equator) at the time of image acquisition (10:30 to
84 11:00 local time). No saturation was observed in the polar regions due to the lower sun incidence angles.
To avoid this saturation in the tropics and mid-latitudes, a request is systematically made to Airbus DS to
86 lower the gain within the 60°N-60°S latitude bands. Technically, it consists in requesting to lower the
number of time delay and integration (TDI) stages from the default value of 13 to a value of 10. Finally, in
an earlier study, we found moderate added value of tri-stereo compared to a standard stereo coverage
(Berthier et al., 2014), likely because most of the imaged glaciers are moderately sloped. Tri-stereo
coverage being 50% more expensive for the project, PGO acquisitions are all performed in standard
stereo mode.

**2.2. Selected glacier targets and acquisitions campaigns**

Given the funding available for the PGO, an exhaustive survey of the ~700 000 km² glaciers on Earth is
94 not feasible. Our strategy is, instead, to focus on a discrete number of sites and propose some tailored
acquisitions. In particular, we are careful to task the Pléiades satellites during a time window prescribed
by experts in glacier research, in most cases at the end of the summer when the snow cover is the
lowest on and off glaciers. This is important because, when snow is present, the risk of image saturation
is higher and, if the snow layer is thick off glacier, the coregistration of the DEMs is more uncertain. Late
summer acquisitions also means that the images and DEMs will often be acquired close in time to the
100 glaciological field measurements or airborne campaigns which facilitates comparisons. Reduced snow
cover also means that most PGO ortho-images should be suitable to update glacier inventories
(Andreassen et al., 2022; Paul et al., 2011) and to delineate the snowline, a proxy for the equilibrium line
if observed close to the end of melt season (Pelto, 2010; Rabatel et al., 2013). Images in the PGO
database are almost cloud free because images acquired with more than 10% of clouds are not validated
and  the tasking continues. If a cloud free stereo-pair is not obtained during the user-defined time
period, the tasking is first extended by a few weeks (if relevant) or/and postponed to the following year.

A PGO site, based on a user-defined polygon, covers typically 100 to 500 km², and generally includes
108 dozens of glaciers. Site selection was performed following a call to the community through the World
Glacier Monitoring Service (WGMS, Zurich), the agency in charge of compiling and disseminating
standardised datasets on glacier fluctuations. The reason to go through the WGMS was that Pléiades

repeat DEMs have a high potential to calibrate (field) glaciological mass balance estimates (Zemp et al., 2013) and also help to assess the regional representativeness of the glaciers monitored in the field. The PGO covers several WGMS benchmark glaciers. We also included iconic glaciers (e.g., Perito Moreno in Argentina ; Kilimanjaro in Tanzania) and, as much as possible, we attempted to ensure that the PGO samples all main glacierized regions on Earth. The PGO only samples a few sites in the Arctic regions (including Alaska) because these glaciers are regularly imaged by the ArcticDEM project (Porter et al., 2018). Among the 19 first order glacier regions defined by the global terrestrial network for glaciers (GTN-G, 2023), only the Russian Arctic is not sampled by the PGO as no request came from the research community for this region. Overall, the PGO acquires imagery over 140 targets (Figure 1, Table 1).

For funding reasons, not all 140 sites can be observed the same year. We thus designed an acquisition program made of 10 original campaigns, five in each hemisphere. These campaigns occur during the summer and early autumn (i.e. from July to October in the north hemisphere and from January to May in the south hemisphere). During each of these campaigns, the Pléiades satellites attempt to acquire images over 10 to 30 glacier sites. The first PGO campaign took place in summer 2016 in the northern hemisphere and the last one in summer 2021 in the southern hemisphere.

Since July 2021 in the northern hemisphere (and February 2022 in the southern), the PGO has entered into "repeat mode", i.e. stereo coverage is repeated five years after previous acquisitions (cloud permitting). The choice of this 5-yr time lag between acquisitions was driven by (i) the wish to have a high signal-to-noise ratio on the measurement of the rate of elevation change, and (ii) the consideration that the volume-to-mass conversion factor is not well-constrained for periods shorter than 5-years (Huss, 2013).

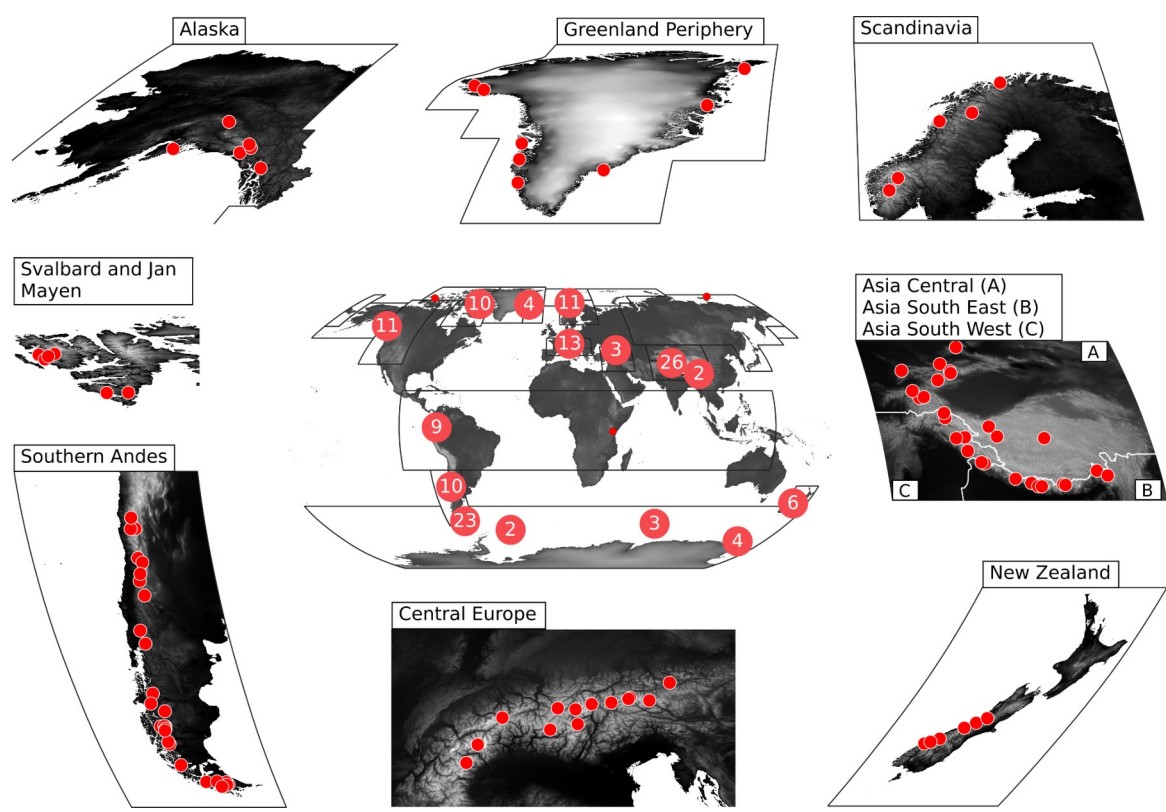

**Figure 1. Map of the distribution of the 140 PGO sites. The central panel shows the number of sites in the main glacier regions and the peripheral panels highlight the distribution of the sites for a few regions of dense spatial**
**coverage.**

**Table 1. Summary of the areas and number of glaciers covered during the first 10 PGO original campaigns. NH**
**stands for Northern Hemisphere, SH for Southern Hemisphere. See also Table 3 for the distribution of sites among the 19 GTN-G first order glacier regions. The columns "total and glacier areas" correspond to the full**
**coverage. The real area coverage by PGO is in fact slightly lower due to data gaps in the DEMs.**

| Campaign | Number of sites | Number of stereo pairs | Total area km² | Glacier area km² | Number of glaciers* |
|---|---|---|---|---|---|
| 2016 NH | 18 | 30 | 7163 | 2514 | 771 |
| 2017 SH | 14 | 28 | 4970 | 1819 | 813 |
| 2017 NH | 29 | 52 | 11,262 | 4434 | 1469 |
| 2018 SH | 9 | 22 | 3671 | 1535 | 365 |
| 2018 NH | 13 | 26 | 4719 | 1826 | 573 |
| 2019 SH | 5 | 23 | 3352 | 1911 | 221 |
| 2019 NH | 14 | 34 | 6229 | 1909 | 1019 |
| 2020 SH | 12 | 21 | 4338 | 1491 | 670 |
| 2020 NH | 14 | 27 | 5276 | 2065 | 784 |
| 2021 SH | 12 | 19 | 3509 | 1870 | 125 |
| **Total** | **140** | **282** | **54,489** | **21,374** | **6810** |

\* Counting only glaciers for which at least 50% of the area is covered.

## 2.3. The PGO products

The PGO products consist of the DEMs and related ortho-images derived automatically from the stereo image pair, and the 5-year maps of elevation difference calculated once a PGO site has been observed
again by the Pléiades satellites.

### 2.3.1. DEMs and ortho-images

Airbus Defense and Space provides Pléiades stereo-pairs at the "primary" processing level. We then generate DEMs and ortho-images using the Ames Stereo Pipeline (ASP) (Beyer et al., 2018 ; Shean et al.,
2016), version 3.0.0, release 2021-10-05 (https://github.com/NeoGeographyToolkit/StereoPipeline). ASP is a suite of free and open source tools designed for processing stereo images captured from satellites
and other platforms. It is extensively used in glaciology to generate DEMs from Maxar WorldView/GeoEye (Shean et al., 2016; Willis et al., 2015), ASTER VNIR  on board TERRA (Brun et al.,
2017; Shean et al., 2020),  Pléiades (Marti et al., 2016; Deschamps-Berger et al., 2020) and Planet SkySat-C (Bhushan et al., 2021) images.

A key step for the generation of a DEM is the correlation between the two images of the stereo-pair. Several algorithms are available in ASP that can lead to different results. Deschamps-Berger et al. (2020)
showed that the choice of the photogrammetric options, and in particular the correlator, has an impact on the precision and completeness of the elevation difference over stable terrain and snow-covered
areas. We used their preferred set of photogrammetric options, based on the Semi-Global Matching (SGM) correlator (Hirschmuller, 2008). SGM has the advantage of providing enhanced DEM detail/quality

and fewer data gaps. However, we observed that in some cases (Figure 2), SGM tended to fill the DEM
with noisy data in textureless areas of the images (cast shadows, areas covered with fresh snow, and in
the case of Fedchenko in Figure 2 image saturation). For this reason, we also processed the stereo-pairs
using the block-matching (BM) correlator with a set of processing parameters taken from (Willis et al.,
2015; Marti et al., 2016). We provide both versions (SGM and BM) and leave it to the user with their
local knowledge of the study area to decide which version of the DEM (or a combination of both) is the
most appropriate for a given study. We produced 2 and 20 m DEMs from the native point clouds
generated by ASP. The 20-m DEM is a smoother version that can be useful for testing some
methodologies on smaller files and for generating more complete ortho-images as it contains less data
gaps.

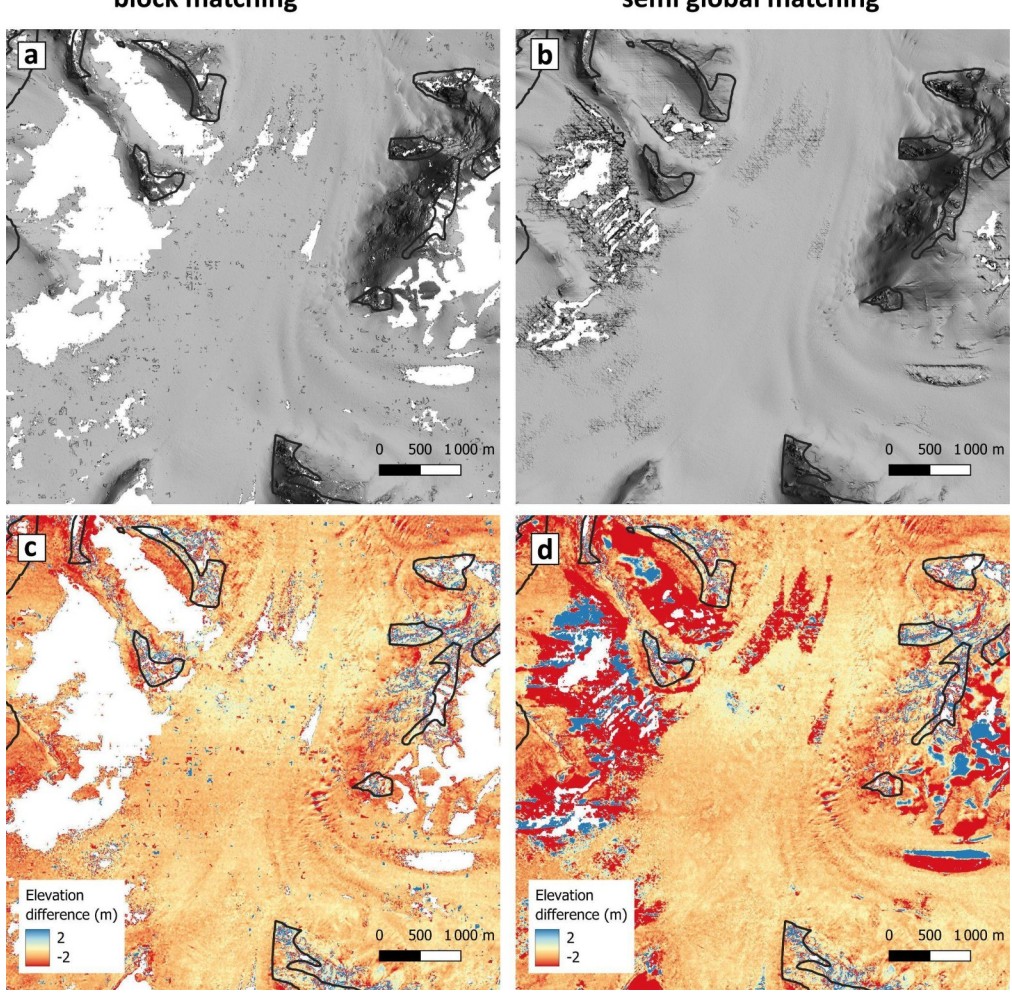

**Figure 2. Comparison of the Pléiades 2-m DEMs derived using the block-matching(left) and semi-global matching**
**(right) algorithms of the Ames Stereo Pipeline (ASP) for the upper accumulation area of Fedchenko glacier**
**(Pamir, Central Asia). Upper panels a and b show shaded relief images of the 2019-08-01 DEMs. Lower panels c**
**and d show the elevation differences between these 2019-08-01 and the 2019-09-22 Pléiades DEMs. Note that**
**the locations where data gaps are present in the block-matching DEMs (white areas in panels a and c)**
**correspond to unrealistically high/low values in the semi-global matching elevation difference map (panel d).**
**These gaps result mostly from saturation in the images.**

In our workflow, 0.5 m panchromatic and 2 m multispectral ortho-images are generated using the 20-m
DEM. Pansharpened images (i.e. multispectral images at 0.5 m resolution) are not calculated and
archived due to file storage limitations. These pansharpened images, however, could easily be generated

by the user using freely available tools such as *pansharp* in ASP or *otbcli_Pansharpening* in the Orfeo
ToolBox (https://www.orfeo-toolbox.org/).

The official absolute geolocation accuracy is 8.5 m (CE90, Circular Error at a confidence level of 90 %) for
Pléiades-1A and 4.5 m for Pléiades-1B (Lebègue et al., 2015) without ground control points (GCPs).
Further, Pléiades DEMs derived without GCPs can be biased in height by as much as 10 to 20 m. To avoid
such horizontal and vertical shifts and to ensure an improved consistency of the PGO database, all DEMs
were coregistered to the Copernicus GLO-30 DEM (GLO-30) using a publicly available implementation of
the Nuth and Kääb (2011) algorithm  (Shean et al., 2023) . GLO-30, an edited version of the TanDEM-X
DEM, has a 30-m ground sampling distance and was chosen as a reference DEM because it is currently
the best global void free DEM publicly available (Franks and Rengarajan, 2023). According to ESA and
AIRBUS (2022), Its absolute vertical accuracy is better than 4m (90% linear error) and its absolute
horizontal accuracy is better than 6m (90% circular error). Given the time lag between the radar images
used to produce the TanDEM-X DEM (2011 to 2015, (Rizzoli et al., 2017)) and the PGO acquisitions,
coregistration was performed on stable terrain, masking out glaciers as inventoried in the RGI v6.0 (RGI
Consortium, 2017). For a few test sites, we found that the 3D translation vector were almost unchanged
when using the 20 m instead of the 2 m DEM. Hence, the 3D translation vectors were computed using
the 20 m DEMs only (a ground sampling distance closer to the one of GLO-30) and the shifts were
applied to all PGO products (2-m and 20-m DEMs and all ortho-images). Coregistration to GLO-30 is
performed separately for BM and SGM DEMs.

Figure 3 shows one of the PGO products (DEM and ortho-images) and the elevation difference to GLO-30
before and after coregistration for a portion of the Purogangri ice cap over the Tibetan Plateau. An
example of the product metadata report that accompanies each PGO product is available in Appendix
A1.

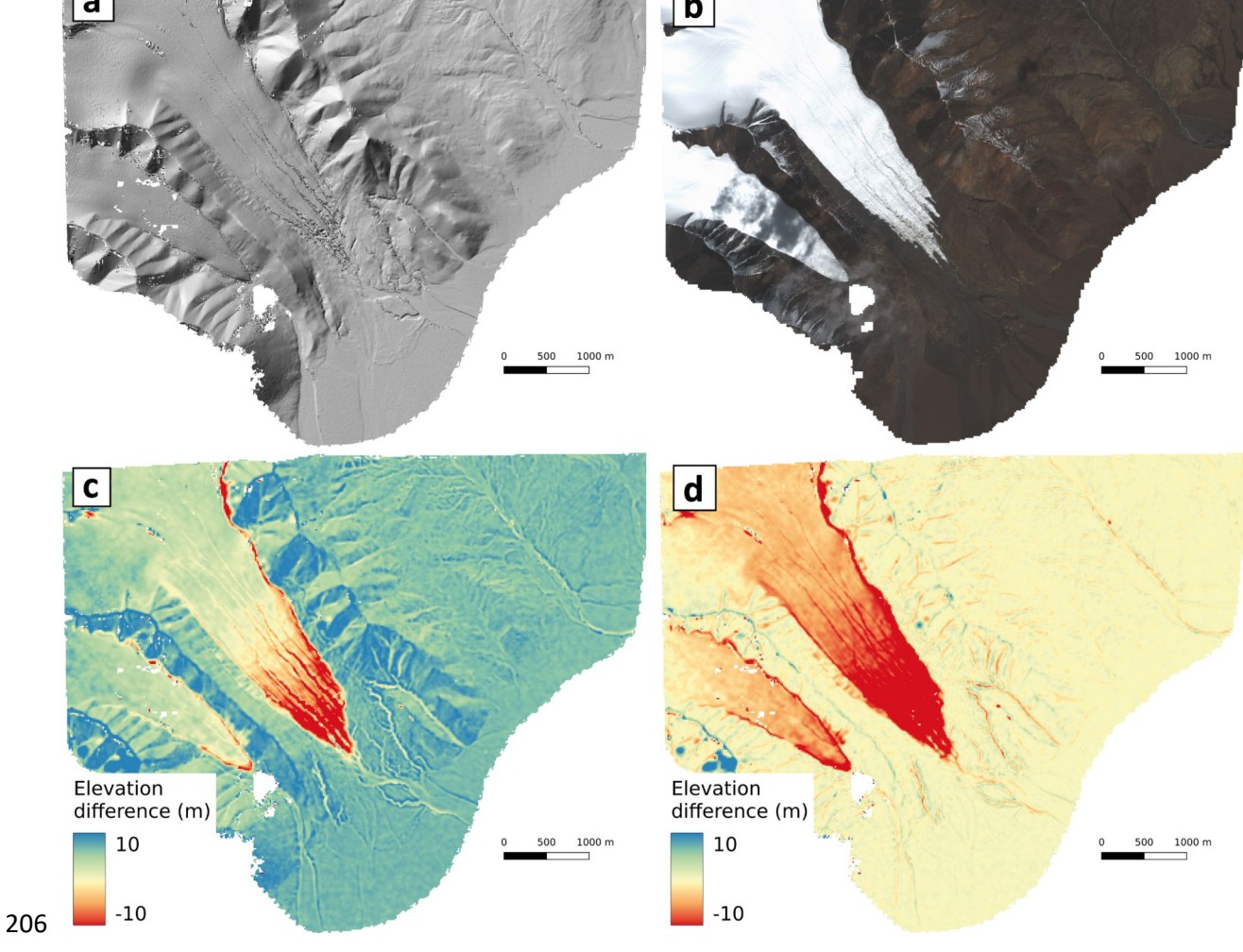

**Figure 3. A sample PGO product for the Purogangri ice cap over the Tibetan Plateau (PGO ID: 2018-10-**
208 **03_0458515_Purogangri_ASC). (a) Shaded relief image of the block-matching DEM; (b) multi-spectral 2-m ortho-**
**image © CNES 2018, Distribution Airbus DS; (c) Elevation difference of the Pléiades DEMs with the Copernicus 30**
**DEM prior (c) and after (d) coregistration. For this specific case, the shift vector of the PGO DEMs to GLO-30**
**were: dEast = −1.8 m ; dNorth = 4.8 m ; dZ = −6.6 m. Coregistration reduced the normalized median absolute**
**deviation (NMAD) off glacier from 0.81 m to 0.48 m.**

**2.3.2. Maps of elevation changes**

Once Pléiades acquisitions are repeated over a site, we generate DEMs from the most recent Pléiades
imagery and compare these to the older DEMs to map five (sometimes six or more) years of glacier
elevation change (Figure 4). This is achieved in two steps: first the most recent Pléiades DEM is
218 coregistered to the older one (derived at the same ground sampling distance and using the same
correlator) on the stable terrain as described above. Next, remaining spatially-coherent elevation biases
are corrected by fitting first a fifth order polynomial in the across-track direction (Gardelle et al., 2013)
and next a spline fit along-track (Falaschi et al., 2023). The latter is needed to correct low-frequency
undulating biases due to the unmodeled attitude error ("jitter") of the Pléiades satellite platform at a
frequency of about 1 Hz (Deschamps-Berger et al., 2020). These along-track biases are not systematic
and have a typical amplitude of 1–2 m and a wavelength of about 4 km. We note that the order of these
corrections (first across-track then along-track) were taken from Gardelle et al. (2013) but were not

studied further and could be the topic of future analysis. We also emphasise that the quality of these bias corrections depends on the availability of sufficient and well-distributed stable terrain. We therefore strongly encourage users to check the relevance of these automatic corrections using the plots associated with each elevation difference map and, if necessary, generate themselves the elevation change map using the PGO DEMs.

The jitter is especially strong for Pléiades 1B since the year 2021 due to an issue with the satellite platform. These across-track and along-track corrections are only efficient if there is a sufficient amount of well-distributed stable terrain around the glaciers. In the case of the Tuyuksu site (Figure 4), successive corrections allow reducing the dispersion of the residuals by almost a factor of two, e.g. the normalized median absolute deviation (NMAD) is lowered from 2.8 to 1.5 m. The along-track undulations are not entirely removed (Figure 4c), however. Thus, we invite the users to check statistics and do visual inspection of the difference maps on stable terrain to assess the quality of the corrections (see also Figure A3 in Berthier et al., in press).

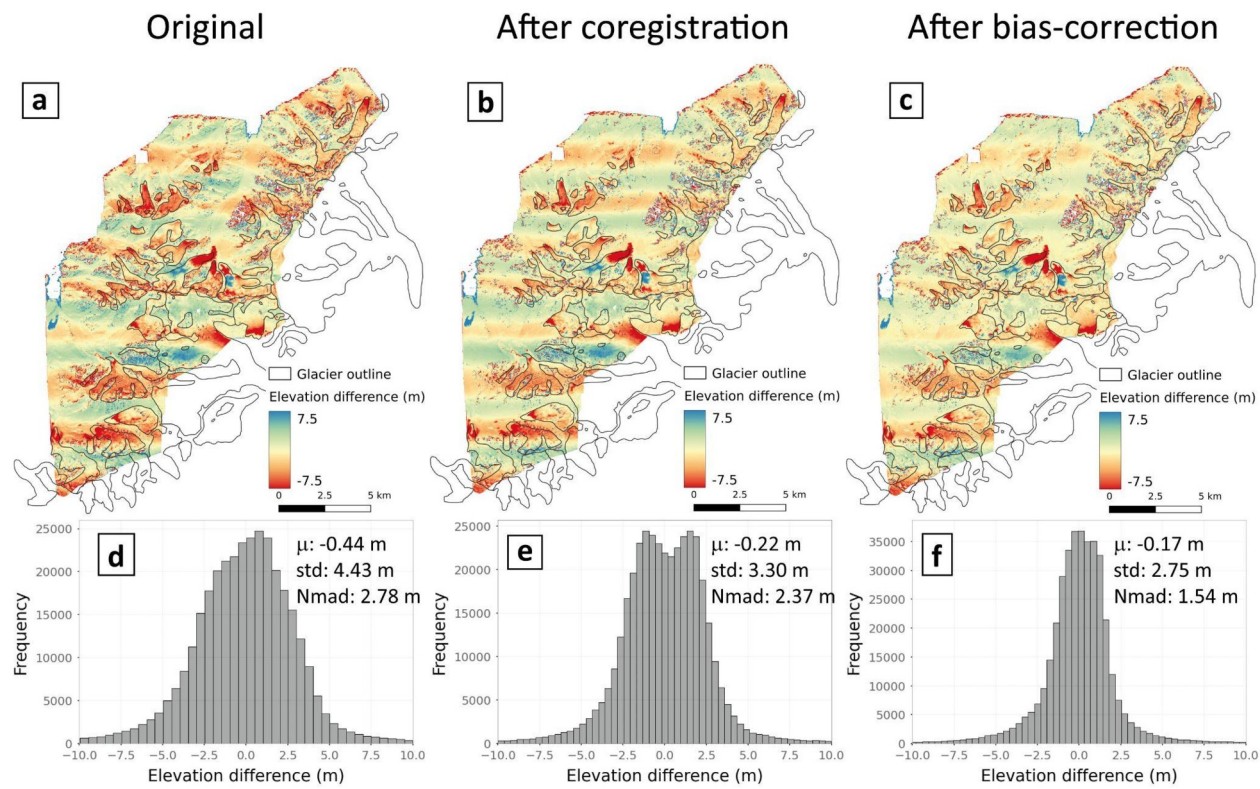

**Figure 4. PGO elevation difference map before and after two correctionson the Tuyuksu (Central Asia) site in Kazakhstan. The upper panels (a, b, c) show the elevation differences maps from August 2016 to August 2021 and the lower panels (d, e, f) the distribution of the elevation differences off glaciers. Maps and histograms are shown before coregistration (a, d), after coregistration (b, e) and after bias correction (c, f). (PGO ID: 2016-08-27_0545099_Tuyuksu_ASC ; 2021-08-21_0546043_Tuyuksu_ASC, both derived from Pléiades 1B images).**

Two, three (and sometimes more) stereo pairs are often needed to cover entirely a single PGO site in a campaign year. After five years, we thus generate the elevation change maps for all possible pairs of overlapping DEMs, at 2 and 20 m ground sampling distance and for the two algorithms (SGM and BM, Fig. 2). Hence, numerous elevation change maps are computed and we leave it to the users to decide which combination works best for their needs. Basic statistics are provided for each elevation change

map (e.g., standard deviation and NMAD off glacier, as in Figure 4) to guide the users in their choice.

# 3. Evaluation of the PGO datasets

**3.1. Evaluation of the DEMs**

**3.1.1 Quality of the coregistration to GLO-30**

We assess the quality of the coregistration of 259 PGO DEMs to GLO-30 (Figure 5) off-glacier. The spread of the residuals are similar in both easting and northing directions with standard deviations of 5 to 6 m,

and the standard deviation is slightly larger than 7 m in the vertical direction. The median shift is almost 0 m in easting direction, whereas the PGO DEMs are slightly shifted (4.5 m) toward the North compared

to GLO-30. This northward shift is larger for DEMs derived from Pléiades 1A images (5.8 m) than from Pléiades 1B (3.2 m) and is especially strong at high (north and south) latitudes, reaching up to 20 m at

80° North in Svalbard. We have no explanation for this small systematic northward shift which is under investigation at the French Space Agency (CNES). PGO DEMs are, on average, 2.4 m lower than GLO-30.

This vertical shift could be due to winter snow affecting the GLO-30 (derived from individual Tandem-X DEMs acquired year round) but not affecting the PGO DEMs, acquired only in summer. This vertical

offset is larger for DEMs derived from Pléiades 1B images (3.9 m) than from Pléiades 1A (1.1 m) We note that these horizontal and vertical shift values (mean/standard deviation) do not represent the absolute

geolocation performance of the Pléiades DEMs as they are also influenced by any mis-registration of GLO-30 itself.

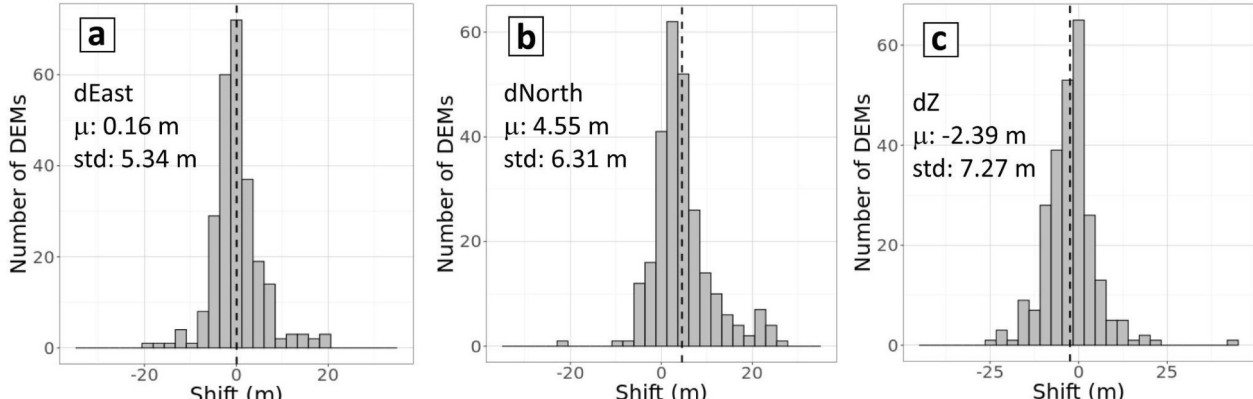

**Figure 5. Distributions of the shifts in the easting (a), northing (b) and vertical (c) directions between 259 PGO DEMs acquired between 2016 and 2021 (10 first campaigns) and GLO-30 off-glacier. "μ" stands for the mean,**

**"std" for the standard deviation. The figure shows translation components for the block-matching DEMs, as the mean and standard deviation for the semi-global matching DEMs were nearly identical.**

Coregistration to GLO-30 failed or led to unreliable horizontal shifts (> 30 m) for about 10% of the sites. Examples of problematic sites include Livingstone Island (Subantarctic and Antarctic Islands) where GLO-

30 displays large artefacts, possibly due to errors during the unwrapping of the TanDEM-X interferograms. Hence for seven DEMs out of nine on this Island, we applied no coregistration.

Coregistration also failed in a few cases where very limited stable terrain was available (e.g., on Balleny Islands around Antarctica). When coregistration failed or was judged unreliable, the Pléiades DEM were

left unchanged (i.e. not shifted) and the unsuccessful coregistration was identified on the metadata sheet accompanying each PGO product.

### 284 3.1.2 Comparison of close-in-time PGO DEMs in their overlapping areas

As several Pléiades DEMs are sometimes needed to cover a PGO site, they include overlapping areas
where the DEMs acquired a few days/weeks apart can be compared. These overlapping areas provide an opportunity to assess the performance of the coregistration, the so-called "triangulation" in Nuth and
Kääb (2011). Indeed, after coregistration to GLO-30, we expect two overlapping Pléiades DEMs to be well-coregistered, and residual shifts between the DEMs can be interpreted as residual coregistration
errors (Figure 6).

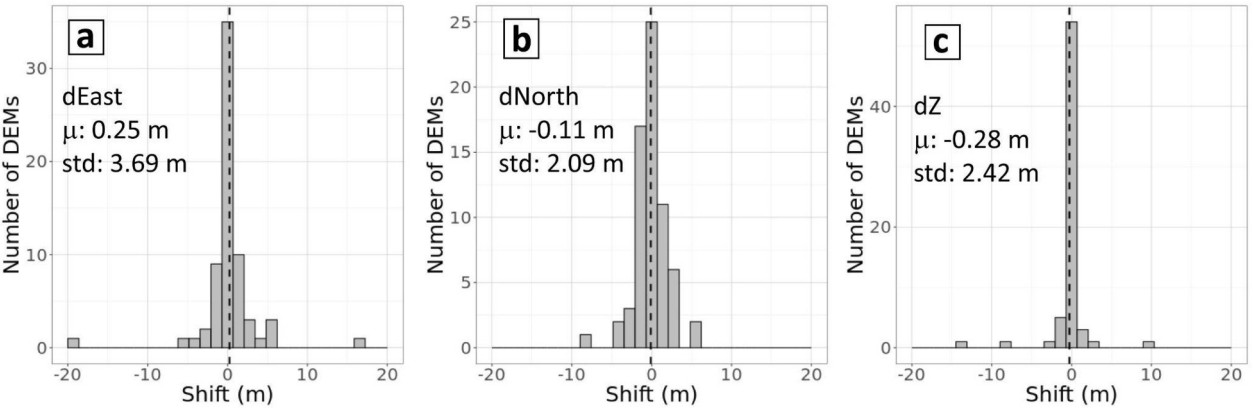

**Figure 6. Distributions of the shifts in the easting (a), northing (b) and vertical (c) directions between PGO DEMs over their overlapping portions (n=64). We only show the results for the 2 m block-matching DEMs. Results are**
**similar for the semi-global matching DEMs and at both resolutions (2 m and 20 m).**

The mean residuals are very close to 0 m in all directions and the standard deviations range from 2 to 4
296 m. This reflects the quality of the PGO DEM coregistration with the reference GLO-30 product. We note that a few PGO DEMs show relative co-registration errors of over 10 m. They correspond to sites in areas
of high relief (e.g., glacier Fedchenko in Tadjikistan or Makalu in Nepal) where GLO-30 is subjected to large errors.

### 300 3.1.3 Evaluation of the PGO DEMs using near-contemporaneous lidar data

In Norway and western Canada, three independent airborne lidar campaigns acquired data within less
than 1 day of a Pléiades stereo acquisition (Table 2). This ideal situation allows us to evaluate the performance of the PGO DEMs because of negligible elevation change on all surfaces (glacier, snow,
permafrost). The simultaneity of the surveys allows comparison of the uncertainties of the PGO DEMs on and off glacier, an important aspect as, in general, one has to assume that the off glacier terrain is
representative of the glacier terrain (Hugonnet et al., 2022). Uncertainties based on repeated lidar acquisitions over stable terrain typically yield errors (~0.1 m) that are almost one order of magnitude
smaller than those of the PGO DEMs.Hence the elevation difference mainly reflects the uncertainties of the PGO DEMs, although ALS errors can be higher that 0.1 m in steep terrain. Details about the western
Canada lidar surveys can be found in Pelto et al. (2019) and for the Norway surveys in TerraTec AS (2018; 2019a; 2019b) and in Andreassen et al. (2023).

The lidar pointclouds were interpolated into 1 m gridded DEMs using ASP's routine *point2dem*. For the comparison, we coregistered each PGO DEM (i.e., BM and SGM) with each synchronous lidar. The DEM

coregistration was done using the RGI v6.0 (RGI Consortium, 2017) glacier inventory as a mask to define
the stable terrain because this is the only inventory available for coregistration on all PGO sites..

Observed elevation differences (Figure 7) are in general near 0, but there are also some artefacts and
differences between BM vs SGM products.

Further, we calculated different statistics to characterise DEM uncertainties, based on the maps of
elevation difference between Pléiades and lidar (Fig. 7): NMAD off-glacier and on-glacier, median off-
320 glacier and on-glacier (Table 3). For these statistics, on- and off- glacier terrain was classified using high
resolution glacier outlines manually digitized on the Pléiades orthoimages and a hillshade representation
of the lidar DEMs.This improved glacier inventory was needed as RGI outlines were outdated and we
wanted to have the best possible separation between glacier and stable terrain.

**Table 2. Characteristics of the lidar surveys used to evaluate the PGO DEMs.**

| Region | Surveyed glaciers | Glacier area (km²) / evaluation area (km2) | Date PGO/lidar YYYY-MM-DD | PGO /Geostore ID | Lidar density p/m² | Avg Slope on/off glacier |
|---|---|---|---|---|---|---|
| Western Canada | Peyto | 47.0 / 94.6 | 2016-09-13 / 2016-09-13 | 2016-09-13_1912075_Wapta_WNA / DS_PHR1B_201609131912075_FR1_PX_W117N51_0616_02636 | 1 | 13°/28° |
| North Norway | Langfjordjøkelen | 6.4 / 17.1 | 2018-09-01 / 2018-09-01 | NaN / DS_PHR1B_201809011030275_FR1_PX_E021N70_0604_01124* | 2 | 12°/31° |
| South Norway | Hellstugubreen, Gråsubreen, Vestre Memurubreen, Austre Memurubreen | 19.7 / 42.7 | 2019-08-27 / 2019-08-26 | 2019-08-27_1102544_Jotunheinmen_SCA / DS_PHR1B_201908271102544_FR1_PX_E008N61_0615_01712 | 2 | 11°/26° |

*Langfjordjøkelen was surveyed by the PGO one year earlier, 8 September 2017. This 2018 Pléiades
stereo pair was not acquired as part of the PGO, this is why we only provide the ID of the Pléiades stereo
pair in the Geostore Airbus D&S catalogue. The processing used for this non-PGO DEM was identical to
328 PGO DEMs.

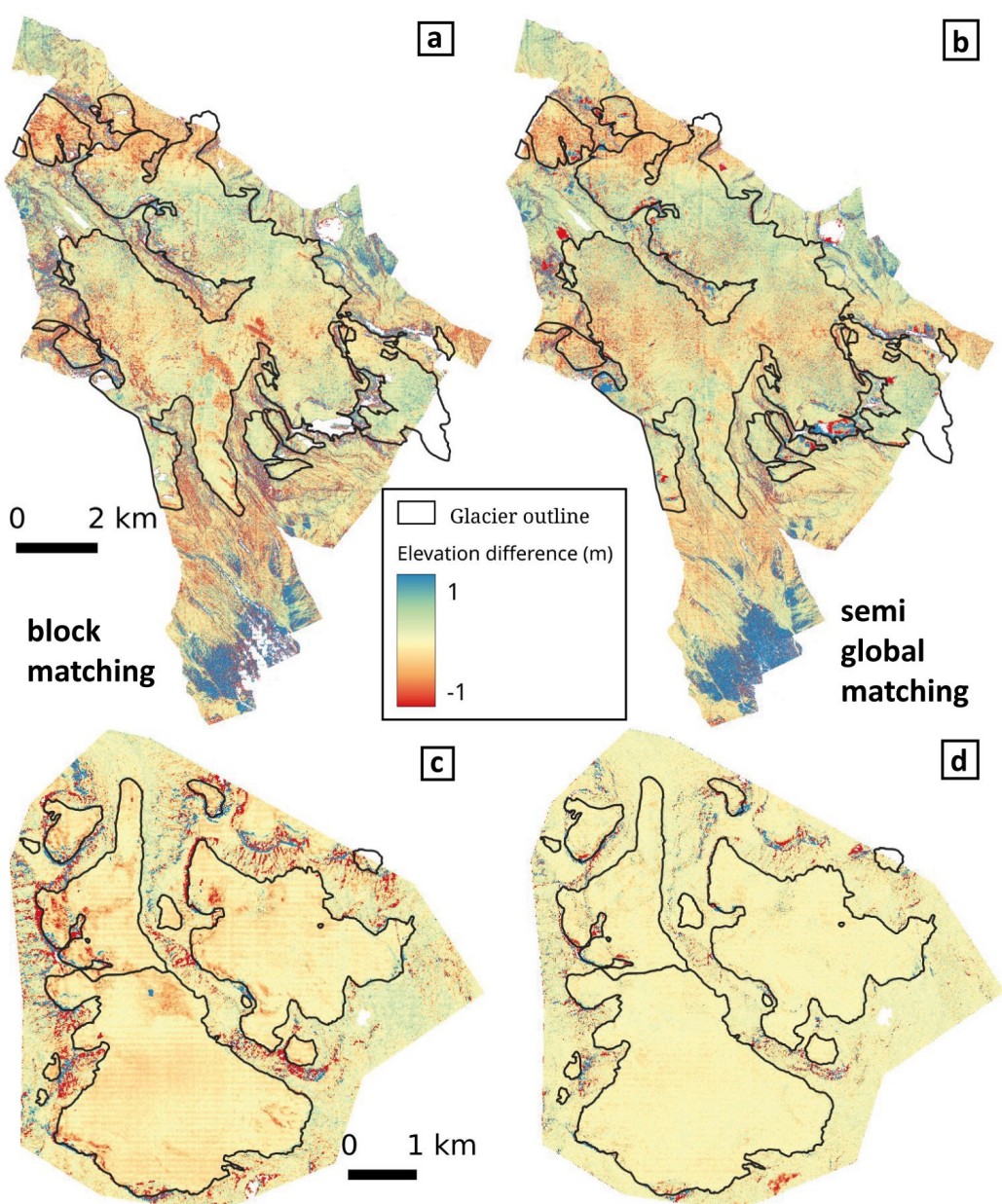

Figure 7. Map of elevation differences between PGO and Lidar DEMs acquired the same day over Peyto Glacier (13 September 2016, Canada, panels a and b) and one day apart over Hellstugubreen (26 and 27 August 2019, Norway, panels c and d). The left column shows the two block-matching DEMs, the right column the semi-global matching DEMs. We do not show the map of elevation difference for other glaciers in Norway (Langfjordjøkelen, Gråsubreen) because the patterns are highly similar.

**Table 3. Statistics on the elevation differences (m) between the PGO 2 m DEMs and the Lidar DEMs acquired the same day. BM - Block matching. SGM - Semi-global matching. "Hellstugubreen" stands for "Hellstugubreen, Gråsubreen, Vestre Memurubreen, Austre Memurubreen".**

|  | Median Dh off glac (m) | Median Dh on glac (m) | NMAD off glac (m) | NMAD on glac (m) |
|---|---|---|---|---|
| 2016 Peyto - BM | 0.02 | −0.01 | 0.59 | 0.36 |
| 2016 Peyto - SGM | 0.03 | 0.00 | 0.46 | 0.41 |
| 2018 Langfjordjøkelen - BM | 0.01 | −0.19 | 0.67 | 0.14 |
| 2018 Langfjordjøkelen - SGM | 0.01 | −0.14 | 0.54 | 0.17 |
| 2019 Hellstugubreen - BM | −0.01 | −0.12 | 0.38 | 0.12 |
| 2019 Hellstugubreen - SGM | 0.00 | −0.09 | 0.29 | 0.15 |

As a result of the co-registration process, the median elevation differences off glaciers are very close to 0 m. Over glacierized terrain, biases are also modest. Almost null for Peyto Glacier, they are slightly negative for the Norwegian sites but always within 0.2 m. Conversely, the dispersion of the residuals are slightly larger for the Canadian site, with a NMAD of about 0.4 m (a result of uncorrected jitter), while it ranges between 0.12 and 0.21 m for the glaciers in Norway. We note that the NMAD are systematically larger off glaciers than on glaciers which confirms that using the off glacier terrain to infer the uncertainty on glaciers is a conservative approach. Interestingly the choice of the correlation algorithm (BM or SGM) has a different influence on and off glaciers. SGM results in lower NMAD off glaciers whereas using BM leads to reduced NMAD on glaciers.

The median elevation difference and its spread (quantified using the NMAD) are rather constant with elevation (Figure 8, only shown for the Peyto site, Canada). Off glacier, the positive elevation differences at low elevations are explained by the presence of vegetation (see also the southernmost portion of the map in Figure 7a-b). The Pléiades summer DEMs map the height of the canopy (Piermattei et al., 2019) while the lidar maps the bare ground below the vegetation. The bias and the NMAD are constant up to slopes of 50°. Above, the dispersion of the elevation difference increases rapidly (on and off glacier) and the median difference departs from 0. These results indicate that a good practice is to exclude areas of high relief (e.g., slopes larger than 50°) during coregistration and when computing the glacier-wide mean elevation changes.

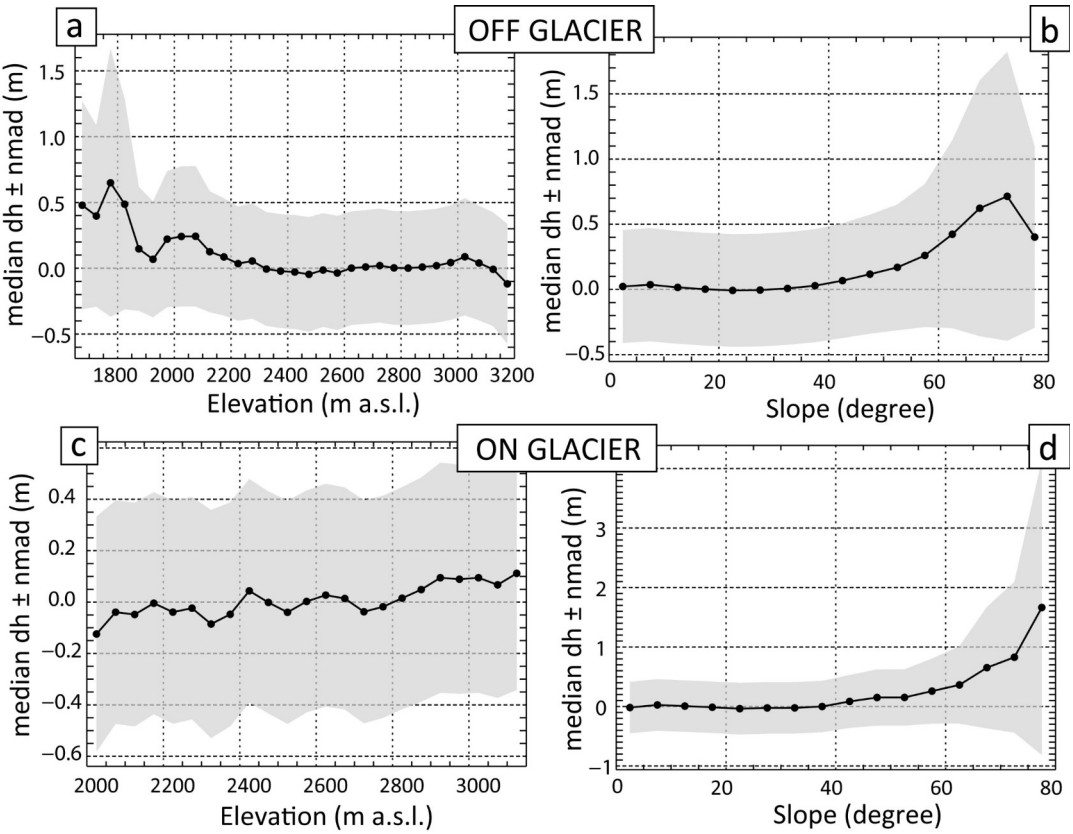

**Figure 8. Median elevation differences (dh) between the Pléiades semi-global matching 2 m DEMs and the lidar DEMs for the Peyto Glacier site (Canada). Points show median and shaded area shows NMAD of dh values (PGO DEM minus Lidar DEM) within each 50 m elevation bin (left) and each 5 degree slope bin (right) off glaciers (upper panels) and on glaciers (lower panels).**

Overall, these evaluations using lidar data suggest that glacier elevation changes can be measured from Pléiades DEMs with a sub-meter accuracy, with a minor influence of the processing algorithm. We note that these evaluations are performed on relatively small glaciers with abundant nearby stable terrain which is required for the coregistration and the bias corrections. So these results may not be readily transferable to larger glaciers.

## 3.2. Uncertainty of the PGO glacier elevation changes

Uncertainties in the elevation difference from repeat Pléiades DEMs have previously been quantified with differential GNSS measurements with centimeter accuracy. In the Mont Blanc massif, such measurements are repeated each year in early September along four transverse profiles on the Mer de Glace and on Argentière glaciers. For the 2021–22 mass balance year, the mean bias of the elevation difference was lower than 0.3 m and its standard deviation lower than 0.4 m (Berthier et al., in press). Similar values were found for elevation difference of Mera Glacier in Nepal from 2012 to 2018, with a mean bias of −0.24 m and standard deviation of 0.52 m (Wagnon et al., 2021).

Here, we quantified the uncertainty of the elevation changes systematically, taking advantage of the depth of the PGO archive. We used the elevation difference off glacier (as mapped in RGI v6.0) as a proxy of the uncertainty on glaciers, with the assumption that elevation difference should be 0 over "stable" terrain, and any observed residual is considered as error. This is a conservative choice as the errors of the DEMs tend to increase with slope (Toutin, 2002; Lacroix, 2016; Hugonnet et al., 2022) and the average slopes are often gentler on glaciers than on nearby ice-free terrain (see also section 3.1.3).

This is also conservative because during the five year time span separating the PGO DEMs, the off glacier terrain has evolved due to e.g., vegetation changes, destabilisation of recently deglaciated slopes. We

calculated uncertainties (at the 95% confidence level) on the mean elevation change over a given area (ranging from 0.01 km² to 10 km²) using the patch method (Miles et al., 2018; Dussaillant et al., 2018).

For a given patch size, we extract the 95th percentile of the absolute mean elevation difference. We analysed 58 PGO elevation difference maps for which the off glacier terrain covered at least 50 km²

(Figure 9).

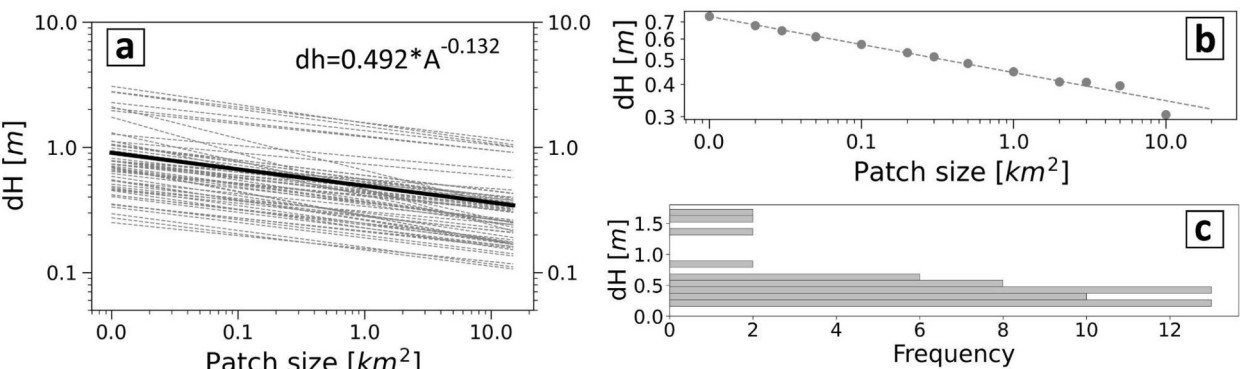

**Figure 9. (a) Uncertainties (dh) at the 95% confidence level (2-sigma) for 58 PGO maps of elevation changes as a**
396 **function of the averaging area. The dashed lines correspond to individual maps of elevation changes obtained from the 2-m BM DEMs and for which the stable terrain occupies more than 50 km². The thick black line**
**corresponds to the mean of all these individual lines and its equation is provided. (b) Example of the uncertainty (at the 95% confidence interval) as a function of the patch size for one of the PGO repeat surveys on**
**Langfjordjokelen in Norway. (c) Distribution of the uncertainties for the 58 elevation difference maps and a patch size of 1 km².**

We observe a relatively large spread of the uncertainties on the elevation differences despite the fact that they are all derived from repeat Pléiades DEMs. For example, the 2-sigma uncertainties for a 1

404 km² patch size range from 0.15 m up to 1.5 m. The largest uncertainties (between 1.2 and 1.5 m, n=6) correspond to maps of elevation difference affected by a larger jitter in the Pléiades DEMs and only

406 partly corrected by our along-track spline correction. This is for example the case for the Tuyuksu (Central Asia) 2016–2021 elevation difference maps shown in Figure 4. Excluding these anomalous six

maps, the remaining uncertainties (95% confidence level) are on average 0.38 m for a 1 km² patch size with a limited spread (n=52, min=0.15 m, max=0.83 m, standard deviation = 0.15 m). The variance of the

mean slope off glacier only explains a small fraction (13%) of the variance in these uncertainties. These mean uncertainties are in agreement with the one derived from same-day lidar surveys (section 3.1.3).

## 4. Are PGO sites representative of the Earth's glaciers?

ASTER VNIR, on board the TERRA platform, is the only sensor in orbit providing publicly-available global coverage using optical stereoscopic images. Recently, it was used to generate maps of elevation

changes and hence to calculate glacier-wide mass balances for almost all the Earth's glaciers from 2000 to 2019 (Hugonnet et al., 2021). However, ASTER will stop acquiring images in 2026 (or 2027) and no

satellite mission is scheduled to provide publicly-available, global coverage with stereo images. Very high

resolution sensors like Pléiades are not fully dedicated to science applications and, currently, do not have the capability to replace ASTER. It is useful, however, to assess whether the 140 glacier sites surveyed by the PGO provide a reasonable assessment of global glacier mass change.

To determine the representativeness of the PGO sampling, we extracted from the Hugonnet et al. (2021) database, the glacier-wide mass balance of glaciers intersecting the PGO sites (named hereafter 'PGO glaciers'). For glaciers only partly covered in a PGO site, we retained those with at least 50% coverage. There are about 6800 PGO glaciers and, in area, they cover 2.5% of the world's glaciers (Table 3). By region, the coverage is highly heterogeneous and varies from 0% in the Russian Arctic to almost 47% in New Zealand. We clarify here that, in this entire analysis, none of the mass balances were derived from PGO elevation change maps. All mass balances are from the Hugonnet et al. (2021) database.

For each GTN-G first order glacier region, we then computed the region-wide mass balances as the area-weighted sum of the PGO glacier-wide mass balances and compared these regionally-aggregated values with corresponding values using the full sample from Hugonnet et al. (2021). Three periods were considered, 2000–2019, i.e. the full period for which the uncertainties are the smallest in Hugonnet et al. database and also two sub-periods, 2000–09 and 2010–19, to test the ability of PGO glaciers to capture the change in mass balance from one decade to another (Figure 10).

At global scale, excluding the unsampled Russian Arctic, the global mass balance during 2000–19 was −0.39±0.02 w.e./yr (Hugonnet et al., 2021). Using only the values for PGO glaciers (Table 4), the global mass balance is more negative (−0.46 m w.e./yr). PGO glaciers capture rather well the acceleration of the mass loss that occurred from 2000–09 to 2010–19. The full sample indicates a drop of the mass balance by 0.05 m w.e./yr between the two periods, PGO glaciers see an almost identical drop of 0.07 m w.e./yr.

At the scale of the 18 individual GTN-G first order regions (Figure 10, Table 4, Russian Arctic excluded), the mass balance differences between the full sample and PGO glaciers are larger. When the 20-yr period is considered, the differences in region-wide mass balance can be as large as 0.34 m w.e./yr (region: Iceland) with a standard deviation of 0.16 m w.e./yr (n=18). Again, PGO glaciers perform better at capturing the change in mass balance between the two 10-yr periods: the maximum difference is 0.21 m w.e./yr (region: Western Canada and USA) and the minimum difference is −0.15 m w.e./yr (regions: South Asia West and Subantarctic and Antarctic Islands), the standard deviation being 0.09 m w.e./yr. For 10 out of 18 RGI regions, the change in region-wide mass balance is observed by PGO glaciers with an error of less than 0.05 m w.e./yr.

Hence, even if the PGO sites were not chosen to represent the World's glaciers, they still provide a reasonable estimate of their mass balance and this sample is able to capture their temporal changes. Yet, one strong complication to use these glaciers for a global mass change analysis would be related to the fact that the Pléiades acquisitions on the 140 PGO glacier sites are not performed simultaneously but using a moving temporal window (Table 1).

It should be noted that there are uncertainties in the Hugonnet et al. (2021) data and that they are not necessarily representative for smaller samples of glaciers or shorter periods (e.g., Andreassen et al., 2023; Berthier et al., 2023). At local scale and for periods of a few months or years, repeated lidar or other high resolution DEMs (e.g. PGO) give more accurate results.

**Table 3. Fraction of the Earth's glacier sampled by the PGO. The number and area of glaciers refer to the RGI v6.0 inventory except in region 12 (Caucasus and Middle East) where the Global Land Ice Measurements**
**from Space (GLIMS) outlines are used, as in Hugonnet et al. (2021).**

| | GTN-G region | Number of glaciers | Glacier area km² | Number of PGO sites | Number of PGO glaciers* | Area of PGO glaciers (in % of the total) |
|---|---|---|---|---|---|---|
| 1 | Alaska | 27,108 | 86,725 | 6 | 190 | 1.0 |
| 2 | Western Canada and USA | 18,855 | 14,524 | 5 | 268 | 3.5 |
| 3 | Arctic Canada North | 4556 | 105,111 | 4 | 22 | 0.4 |
| 4 | Arctic Canada South | 7415 | 40,888 | 2 | 54 | 0.8 |
| 5 | Greenland Periphery | 19,306 | 89,717 | 8 | 255 | 1.9 |
| 6 | Iceland | 568 | 11060 | 1 | 17 | 1.6 |
| 7 | Svalbard and Jan Mayen | 1615 | 34187 | 6 | 60 | 3.2 |
| 8 | Scandinavia | 3417 | 2949 | 5 | 238 | 17.3 |
| 9 | Russian Arctic | 1069 | 51,592 | 0 | 0 | 0 |
| 10 | North Asia | 5151 | 2410 | 2 | 113 | 7.1 |
| 11 | Central Europe | 3927 | 2092 | 13 | 882 | 33.3 |
| 12 | Caucasus and Middle East | 3516 | 1336 | 3 | 344 | 25.8 |
| 13 | Central Asia | 54,429 | 49,303 | 12 | 1185 | 4.5 |
| 14 | South Asia West | 27,988 | 33,568 | 5 | 301 | 1.7 |
| 15 | South Asia East | 13,119 | 14,734 | 9 | 624 | 7.9 |
| 16 | Low Latitudes | 2939 | 2341 | 9 | 220 | 12.6 |
| 17 | Southern Andes | 15,908 | 29,429 | 30 | 894 | 10.3 |
| 18 | New Zealand | 3537 | 1162 | 6 | 935 | 46.8 |
| 19 | Subantarctic and Antarctic Islands | 2752 | 132,867 | 14 | 208 | 2.2 |
| | **Global** | **217,715** | **705,995** | **140** | **6810** | **2.5** |
| | **Global excl. Russian Arctic** | **216,106** | **654,405** | **140** | **6810** | **2.7** |

* We only count glaciers for which at least 50% of the area is covered.

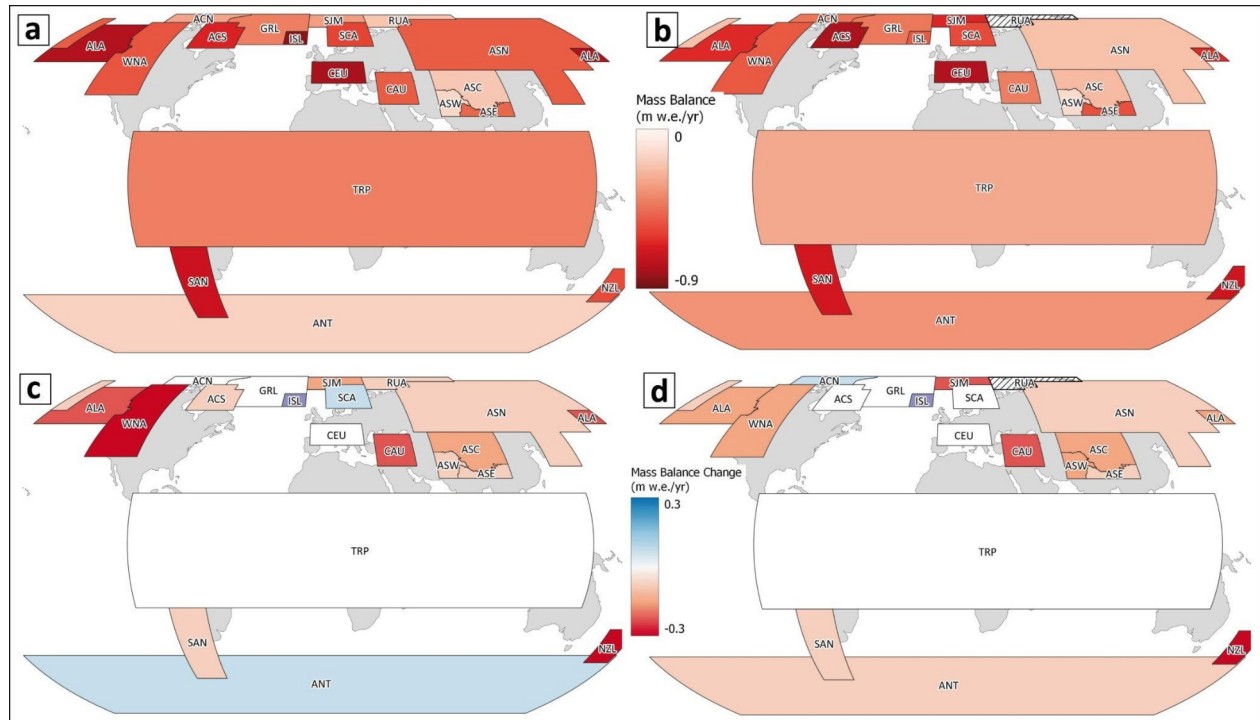

**Figure 10. Comparison of the 2000-2019 region-wide mass balance calculated using the entire Hugonnet et al. (2021) dataset (a) and using only the glaciers sampled by the PGO (b). The lower panels show the changes in region-wide mass balance between 2000-2009 and 2010-2019 for (c) all glaciers and for (d) the subset of glaciers sampled by the PGO. All mass balances are from the Hugonnet et al. (2021) database (i.e., none were derived from PGO elevation change maps).**

**Table 4. Regional and global mass balance (in m w.e./yr) from the entire RGI sample (Hugonnet et al., 2021) and from the PGO glaciers (this study). MB stands for mass balance. Delta_MB corresponds to the change**

**in region-wide mass balance from 2000–09 to 2010–19. All mass balances are from the Hugonnet et al. (2021) database (i.e. none were derived from PGO elevation change maps).**

|  | GTN-G region | MB 2000-19 ALL | MB 2000-19 PGO | Delta_MB ALL | Delta_MB PGO |
|---|---|---|---|---|---|
| 1 | Alaska | −0.77 | −0.63 | 0.12 | 0.16 |
| 2 | Western Canada and USA | −0.52 | −0.51 | −0.10 | 0.11 |
| 3 | Arctic Canada North | −0.29 | −0.43 | -0.18 | −0.09 |
| 4 | Arctic Canada South | −0.65 | −0.79 | −0.21 | −0.06 |
| 5 | Greenland Periphery | −0.40 | −0.42 | 0.00 | −0.04 |
| 6 | Iceland | −0.85 | −0.51 | 0.36 | 0.32 |
| 7 | Svalbard and Jan Mayen | −0.31 | −0.64 | −0.28 | −0.39 |
| 8 | Scandinavia | −0.57 | −0.55 | 0.05 | −0.01 |
| 9 | Russian Arctic | −0.20 | NaN | −0.06 | NaN |
| 10 | North Asia | −0.50 | −0.21 | 0.28 | 0.31 |
| 11 | Central Europe | −0.80 | −0.77 | 0.05 | 0.00 |
| 12 | Caucasus and Middle East | −0.50 | −0.40 | 0.08 | 0.12 |
| 13 | Central Asia | −0.19 | −0.23 | −0.02 | −0.05 |
| 14 | South Asia West | −0.14 | −0.13 | 0.08 | −0.07 |
| 15 | South Asia East | −0.47 | −0.53 | −0.05 | −0.08 |
| 16 | Low Latitudes | −0.40 | −0.28 | 0.12 | 0.10 |
| 17 | Southern Andes | −0.70 | −0.68 | 0.02 | 0.01 |
| 18 | New Zealand | −0.55 | −0.69 | −0.10 | −0.18 |
| 19 | Subantarctic and Antarctic Islands | −0.16 | −0.34 | −0.10 | −0.25 |
|  | **Global excl. Russian Arctic** | **−0.39** | **−0.46** | **−0.05** | **−0.07** |

## 5. Conclusion

The Pléiades Glacier Observatory is an initiative by the French Space Agency (CNES) and LEGOS to facilitate access to very high resolution digital elevation models, elevation change maps, and, after

signing a licence, ortho-images of glaciers. Such data are useful to calculate glacier geodetic mass balances, but also to support other glaciology oriented applications, such as updating glacier outlines,

extracting glacier hypsometry or qualitatively documenting glacier changes. The PGO aims at managing the Pléiades acquisitions, and distributing products that are tailored for glaciological applications, and as

user friendly as possible. The acquisitions started in 2016 and during the first five years, acquired stereo-pairs over 140 target sites around the globe, selected through a call to the glaciological community.

Since 2021, these acquisitions have been progressively repeated to produce maps of elevation change over five years. At the time of writing, already 31 publications used PGO data to examine glacier

changes.

We quantified the uncertainties of the DEMs (after coregistration to the Copernicus GLO-30 DEM) and
492 elevation change maps derived from repeat Pléiades DEMs. This was done with two methods: (1) comparison to near-contemporaneous accurate lidar surveys, and (2) using residual elevation difference
values on nearby stable terrain to estimate corresponding uncertainty on glacier surfaces. Both methods agree broadly on the uncertainties, and as a rule of thumb, the mean glacier-wide elevation differences
have a 2-sigma uncertainty of about 0.5 m for a glacier of 1 km² or larger.

Pléiades satellites are planned to orbit until 2026. Access to data from their successors (Pléiades Neo) is
498 not yet secured for the scientific community and the cost may be prohibitive. It should be a priority for the space agencies to continue to provide high resolution stereo-imagery to scientists to observe the
500 imprint of climate change on the Earth surface and in particular on glaciers.

## Data availability statement

Pléiades Glacier Observatory DEMs and elevation change products are under CC-BY-NC licence and freely available at: https://a2s-dissemination.u-strasbg.fr/#!

Licensing issues prevent  open distribution of primary  Pléiades products and ortho-images. These images are available after signing the Pléiades institutional scientific licence to be requested to the
French space agency CNES (dinamis@cnes.fr).

The scripts used to generate the DEMs and ortho-images and to coregister them to GLO-30 are available
at: https://zenodo.org/uploads/12909586.

## Author contributions

EB designed the PGO program with contribution from DF and SH. JL and EB generated the DEMs and elevation change maps. JMCB, LMA and BM provided Lidar data and all related analysis. CB worked on
the regional representativity of the PGO sites. All authors contributed to the discussion of the results. EB prepared the manuscript with contributions from all co-authors.

## Competing interests

One author (EB) is a member of the editorial board of The Cryosphere.

## Acknowledgments

E. Berthier and J. Lebreton acknowledge support from the French Space Agency (CNES). B.
Menounos acknowledges funding from the Natural Sciences and Engineering Research Council of Canada and the Tula Foundation. L.M. Andreassen acknowledges the internal NVE project
'N80524 Regionalt massebalanseestimat av norske breer'. This is also a contribution to the International Association of Cryospheric Sciences (IACS) working group on Regional Assessments
of Glacier Mass Change (RAGMAC).

 **Appendix**

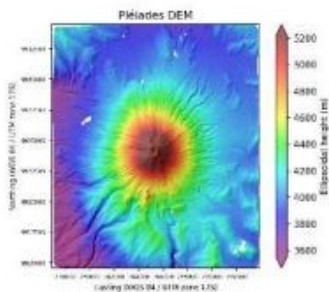

## Pléiades Glacier Observatory : DEM

Date : 2016-11-15
Site : Cotopaxi_TRP

### DEM information

| | |
|---|---|
| **Coordinate system** | UTM 17 south - EPSG 32717 |
| **Correlation algorithm** | Semi Global Matching (SGM) |
| **DEM resolution** | 2 m and 20 m |
| **Reference for height** | Ellipsoidal Height (WGS84) |
| **Shift vector to Copernicus GLO-30 (m)** | dx=-2.69; dy=-2.64; dz=+1.74 |
| **Base-to-Height ratio (B/H)** | 0.42 |

### Source images

**PHR**    DS_PHR1B_201611151534305_FR1_PX_W079S01_0708_01575

**PHR**    DS_PHR1B_201611151535093_FR1_PX_W079S01_0708_01604

### Copyright

Pléiades © CNES Year_of_acquisition, Distribution Airbus D&S

### Archive structure

```
└─2016-11-15_1535065_Cotopaxi_TRP
  ├─BM
  ├─2016-11-15_1535065_Cotopaxi_TRP_footprint.shp
  ├─2016-11-15_1535065_Cotopaxi_TRP_footprint.dbf
  ├─2016-11-15_1535065_Cotopaxi_TRP_footprint.prj
  ├─2016-11-15_1535065_Cotopaxi_TRP_footprint.shx
  └─SGM
    ├─2016-11-15_1535065_Cotopaxi_TRP_1B_DEM_SGM_2m.tif
    ├─2016-11-15_1535065_Cotopaxi_TRP_1B_DEM_SGM_20m.tif
    ├─README_SGM_DEM.pdf
    ├─PREVIEW_2016-11-15_1535065_Cotopaxi_TRP_1B_DEM_SGM_20m.png
    └─Coreg_2016-11-15_1535065_Cotopaxi_TRP_1B_DEM_SGM_20m_vs_Cop30.png
```

### Description

DEMs and orthoimages where generated from raw Pléiades images using the Ames Stereo Pipeline [Beyer et al., 2018]. The set of processing parameters used for DEM generation are from [Marti et al., TC, 2016] for block matching -BM- and from [Deschamps-Berger et al., 2020] for semi global matching -SGM.

All DEMs and orthoimages are coregistered on the Copernicus GLO-30 DEM using the demcoreg tool [Shean et al., 2021].

Acknowledgement statement: The Pléiades images/DEMs used in this study was provided by the Pléiades Glacier Observatory initiative of the French Space Agency (CNES) and Laboratoire d'Etudes en Géophysique et Océanographie Spatiales (LEGOS).

When an image is shown in a presentation, website or an article, the copyright should be (Pléiades © CNES Year_of_acquisition, Distribution Airbus D&S).

We remind to cope with the licence rules regarding (no) data sharing and no commercial use.

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

Ap

pendix Figure A1: Example of the fact sheet accompanying each PGO product, here the semi-global matching (SGM) DEMs over the Cotopaxi area acquired 15 November 2016.

**Appendix Table A1: The 140 sites of the Pléiades Glacier Observatory. The list is ordered chronologically by campaign (HN for North Hemisphere, HS for South Hemisphere). The name of each site is followed by the 3 letters of the GTN-G first order region it belongs to. The table also lists the latitude, longitude of each site, the number of TDI stages that were used during the image acquisitions and the number of stereo pairs needed to cover the entire site.**

| Campaign | Site_Region | Latitude (°) | Longitude (°) | Nb TDI stages | Nb stereo pairs |
|---|---|---|---|---|---|
| 2016_HN | Antisana_TRP | 0 | -78 | 10 | 1 |
| 2016_HN | Bologna_WNA | 62 | -128 | 10 | 3 |
| 2016_HN | Broggerhalvoya_SJM | 78.7 | 12.6 | 13 | 3 |
| 2016_HN | Columbia_WNA | 52 | -117.5 | 10 | 2 |
| 2016_HN | Cotopaxi_TRP | -0.7 | -78.43 | 10 | 1 |
| 2016_HN | Garibaldi_WNA | 50 | -123 | 10 | 2 |
| 2016_HN | Grisefiord_ACN | 76.5 | -82.5 | 13 | 2 |
| 2016_HN | Gulkana_ALA | 63.5 | -145.5 | 10 | 2 |
| 2016_HN | Kongsfjord_SJM | 79 | 12.6 | 13 | 2 |
| 2016_HN | Meighen_ACN | 80 | -99.5 | 13 | 1 |
| 2016_HN | Melville_ACN | 75.5 | -115 | 13 | 1 |
| 2016_HN | Ortles_CUE | 46.5 | 10.5 | 10 | 2 |
| 2016_HN | Sonnblickkees_CUE | 47 | 12.5 | 10 | 2 |
| 2016_HN | Svetisen_SCA | 66.5 | 14 | 10 | 1 |
| 2016_HN | Tuyuksu_ASC | 43 | 77 | 10 | 2 |
| 2016_HN | Wapta_WNA | 51.5 | -116.5 | 10 | 1 |
| 2016_HN | Wolverine_ALA | 60.5 | -149 | 10 | 1 |
| 2016_HN | Yasghil_ASW | 36.5 | 75.5 | 10 | 1 |
| **2017_HS** | Alvear_SAN | -55 | -68 | 10 | 1 |
| **2017_HS** | BahiaDelDiablo_ANT | -63.75 | -67.5 | 13 | 2 |
| **2017_HS** | GardenEden_NZL | -43.25 | 170.75 | 10 | 4 |
| **2017_HS** | Gourdon_ANT | -64.25 | -67.5 | 13 | 2 |
| **2017_HS** | Hudson_SAN | -46 | -73 | 10 | 2 |
| **2017_HS** | Lautaro_SAN | -49 | -73.5 | 10 | 2 |
| **2017_HS** | MtAspiring_NZL | -44.5 | 168.5 | 10 | 3 |
| **2017_HS** | MtCook_NZL | -43.5 | 170.25 | 10 | 3 |
| **2017_HS** | Olivares_SAN | -33 | -70 | 10 | 2 |
| **2017_HS** | Peteroa_SAN | -35.5 | -70.5 | 10 | 1 |
| **2017_HS** | RioToro_SAN | -49 | -73 | 10 | 1 |
| **2017_HS** | SierraBeauvoir_SAN | -54 | -68.5 | 10 | 3 |
| **2017_HS** | Tronador_SAN | -41.15 | -71.9 | 10 | 1 |
| **2017_HS** | Ushuaia_SAN | -55 | -68.5 | 10 | 1 |
| 2017_HN | Elbrus_CAU | 43.25 | 42.5 | 10 | 1 |
| 2017_HN | Fedchenko_ASC | 38.75 | 72.15 | 10 | 6 |
| 2017_HN | GranParadis_CEU | 45.5 | 7 | 10 | 2 |
| 2017_HN | Hansbreen_SJM | 77 | 15.5 | 13 | 1 |
| 2017_HN | Hornbreen_SJM | 77 | 17 | 13 | 1 |
| 2017_HN | Kaffioyra_SJM | 78.5 | 12.5 | 13 | 1 |
| 2017_HN | Kaunertal_CEU | 47 | 10.75 | 10 | 2 |
| 2017_HN | Langfjordjokelen_SCA | 70 | 22 | 13 | 2 |
| 2017_HN | Langtang_ASE | 28.25 | 85.7 | 10 | 2 |
| 2017_HN | Lingmarksbraeen_GRL | 69.25 | -53.5 | 13 | 1 |
| 2017_HN | Lombardy_CEU | 46.25 | 10 | 10 | 2 |
| 2017_HN | Lunana_ASE | 28 | 90.25 | 10 | 2 |
| 2017_HN | Olsen_GRL | 74.75 | -22 | 13 | 2 |
| 2017_HN | Oraefajokull_ISL | 64 | -16.5 | 13 | 2 |
| 2017_HN | Pasterze_CEU | 47 | 12.75 | 10 | 2 |
| 2017_HN | Qaanaaq_GRL | 77.5 | -69.5 | 13 | 2 |

| | | | | | |
|---|---|---|---|---|---|
| 2017_HN | Qasigiannguit_GRL | 64 | -51 | 13 | 2 |
| 2017_HN | QuelccayaIceCap_TRP | -14 | -70.75 | 10 | 2 |
| 2017_HN | RedRockCliff_GRL | 77 | -67.5 | 13 | 1 |
| 2017_HN | Rhone_CEU | 46.5 | 8.5 | 10 | 2 |
| 2017_HN | RikhaSamba_ASE | 28.75 | 83.5 | 10 | 1 |
| 2017_HN | Sarek_SCA | 77 | 17.5 | 13 | 2 |
| 2017_HN | Silvretta_CEU | 47 | 10 | 10 | 1 |
| 2017_HN | Stubai_CEU | 47 | 11 | 10 | 2 |
| 2017_HN | Trambau_ASE | 28 | 86.5 | 10 | 2 |
| 2017_HN | Valpelline_CEU | 45.5 | 7 | 10 | 2 |
| 2017_HN | Variegated_ALA | 60 | -139.2 | 10 | 3 |
| 2017_HN | Venediger_CEU | 47 | 12.75 | 10 | 1 |
| 2017_HN | Zillertal_CEU | 47 | 11.75 | 10 | 1 |
| **2018_HS** | Chico_SAN | -49 | -73 | 10 | 5 |
| **2018_HS** | Cocuy_TRP | 6.5 | -72.25 | 10 | 1 |
| **2018_HS** | Grey_SAN | -51 | -73.5 | 10 | 4 |
| **2018_HS** | Huascaran_TRP | -9.05 | -77.6 | 10 | 1 |
| **2018_HS** | PeritoMoreno_SAN | -50.5 | -73 | 10 | 3 |
| **2018_HS** | Rolleston_NZL | -43 | 171.5 | 10 | 1 |
| **2018_HS** | SanLorenzo_SAN | -47.5 | -72.25 | 10 | 2 |
| **2018_HS** | SantaMarta_TRP | 10.84 | -73.7 | 10 | 2 |
| **2018_HS** | Tupungato_SAN | -33.5 | -69.75 | 10 | 3 |
| 2018_HN | AruCo_ASC | 34 | 82.25 | 10 | 1 |
| 2018_HN | BashKhaindy_ASC | 41 | 76 | 10 | 2 |
| 2018_HN | Dachstein_CEU | 47.5 | 13.5 | 10 | 1 |
| 2018_HN | Karabatkak_ASC | 42 | 78.25 | 10 | 1 |
| 2018_HN | Kketau_ASC | 45 | 80.5 | 10 | 1 |
| 2018_HN | Konsvegen_SJM | 78.75 | 13 | 10 | 2 |
| 2018_HN | LemonCreek_ALA | 58.5 | -134.5 | 10 | 1 |
| 2018_HN | Makalu_ASE | 27.75 | 87 | 10 | 2 |
| 2018_HN | Mittivakkat_GRL | 65.75 | -35.5 | 10 | 3 |
| 2018_HN | Purogangri_ASC | 34 | 89 | 10 | 3 |
| 2018_HN | Satopanth_ASE | 30.75 | 79.5 | 10 | 4 |
| 2018_HN | Thana_ASE | 28 | 90.75 | 10 | 2 |
| 2018_HN | White_ACN | 79.5 | -91 | 13 | 2 |
| **2019_HS** | AguaNegra_SAN | -30.25 | -69.75 | 10 | 3 |
| **2019_HS** | Heard_ANT | -53 | 73.5 | 10 | 4 |
| **2019_HS** | Livingstone_ANT | -62.5 | -60.5 | 13 | 9 |
| **2019_HS** | SanQuintin_SAN | -47 | -73.75 | 10 | 6 |
| **2019_HS** | Universidad_SAN | -34.5 | -70.25 | 10 | 1 |
| 2019_HN | Aktru_ASN | 50 | 87.5 | 10 | 2 |
| 2019_HN | Aqqutikitsoq_GRL | 67.15 | -53 | 10 | 3 |
| 2019_HN | Barkrak_ASC | 42.15 | 71 | 10 | 3 |
| 2019_HN | Bezengi_CAU | 43 | 43.2 | 10 | 2 |
| 2019_HN | DeLongIslands_ASN | 76.75 | 148.75 | 13 | 2 |
| 2019_HN | Grinnell_ACS | 62.6 | -66.75 | 10 | 1 |
| 2019_HN | HolmLand_GRL | 80.35 | -17 | 10 | 10 |
| 2019_HN | Jotunheinmen_SCA | 61.5 | 8.5 | 10 | 3 |
| 2019_HN | Kilimanjaro_TRP | -3 | 37.5 | 10 | 1 |
| 2019_HN | Kolka_CAU | 42.75 | 44.5 | 10 | 2 |
| 2019_HN | Parlung24K_ASE | 29.75 | 95.75 | 10 | 2 |
| 2019_HN | ParlungN4_ASE | 29 | 97 | 10 | 2 |
| 2019_HN | TerraNivae_ACS | 62.3 | -66.5 | 10 | 2 |
| 2019_HN | Zulmart_ASC | 38.85 | 73 | 10 | 1 |
| **2020_HS** | DaviesDome_ANT | -64 | -58 | 10 | 2 |
| **2020_HS** | Domuyo_SAN | -36.6 | -70.4 | 10 | 1 |

| | | | | | |
|---|---|---|---|---|---|
| **2020_HS** | EsteroDerecho_SAN | -30.4 | -70.4 | 10 | 2 |
| **2020_HS** | Fiordland_NZL | -44.7 | 168 | 10 | 2 |
| **2020_HS** | GlaciarDeLosTres_SAN | -49.3 | -73 | 10 | 1 |
| | GranCampoNevado_SAN | -52.75 | -73 | 10 | 2 |
| **2020_HS** | | | | | |
| **2020_HS** | Huila_TRP | 3 | -76 | 10 | 1 |
| **2020_HS** | Kerguelen_ANT | -49.25 | 69 | 10 | 3 |
| **2020_HS** | Mocho_SAN | -40 | -72 | 10 | 1 |
| **2020_HS** | Olivine_NZL | -44.5 | 168.4 | 10 | 5 |
| **2020_HS** | PascuaLama_SAN | -29.3 | -70 | 10 | 1 |
| **2020_HS** | Schiaparelli_SAN | -54.5 | -70.8 | 10 | 1 |
| 2020_HN | Abramov_ASC | 39.6 | 71.5 | 10 | 3 |
| 2020_HN | AkShirak_ASC | 41.8 | 78.3 | 10 | 6 |
| 2020_HN | Altar_TRP | -1.7 | -78.4 | 10 | 1 |
| 2020_HN | ChhotaShigri_ASW | 32.2 | 77.5 | 10 | 2 |
| 2020_HN | Chimborazo_TRP | -1.5 | -78.8 | 10 | 1 |
| 2020_HN | Disappointment_ALA | 60.5 | -138.5 | 10 | 2 |
| 2020_HN | Gangotri_ASE | 33.8 | 76.3 | 10 | 2 |
| 2020_HN | Guliya_ASC | 35.3 | 81.5 | 10 | 1 |
| 2020_HN | Hardangerjokulen_SCA | 60.5 | 7.4 | 10 | 1 |
| 2020_HN | Kluane_ALA | 60.9 | -139.5 | 10 | 3 |
| 2020_HN | Koshik_ASW | 36.9 | 75.4 | 10 | 1 |
| 2020_HN | Ladakh_ASW | 34 | 77.5 | 10 | 1 |
| 2020_HN | Meager_WNA | 50.6 | -123.5 | 10 | 1 |
| 2020_HN | Zanskar_ASW | 33.8 | 76.3 | 10 | 1 |
| **2021_HS** | Astrolabe_ANT | -66.8 | 140 | 13 | 3 |
| **2021_HS** | BallenyIsland1_ANT | -66.4 | 162.5 | 13 | 1 |
| **2021_HS** | BallenyIsland2_ANT | -66.7 | 163.25 | 13 | 1 |
| **2021_HS** | BallenyIsland3_ANT | -67.5 | 164.75 | 13 | 1 |
| **2021_HS** | DrygalskiIsland_ANT | -65.7 | 92.5 | 13 | 2 |
| **2021_HS** | LavoisierIsland_ANT | -66.2 | -66.75 | 13 | 1 |
| **2021_HS** | Marinelli_SAN | -55.5 | -69.6 | 10 | 2 |
| **2021_HS** | MontaguIs_ANT | -58.5 | -26.4 | 10 | 1 |
| **2021_HS** | Roncagli_SAN | -54.75 | -69.2 | 10 | 4 |
| **2021_HS** | SouthOrkney_ANT | -60.7 | -44.6 | 13 | 1 |
| **2021_HS** | Viedma_SAN | -49.5 | -73.1 | 10 | 2 |
| **2021_HS** | WarsawIcefield_ANT | -62.2 | -58.6 | 13 | 1 |

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
