# Peer review of "The Pléiades Glacier Observatory: high resolution digital elevation models and ortho-imagery to monitor glacier change"

_EGUsphere, 2024_

## Referee Comment (RC2)

**Review of "The Pléiades Glacier Observatory: high resolution digital elevation models and ortho-imagery to monitor glacier change" by Berthier et al.**

https://egusphere.copernicus.org/preprints/2024/egusphere-2024-250/

David Shean

May 26, 2024

Note: I initially completed a detailed review of the paper on March 20, 2024. Unfortunately, due to a number of factors in professional and personal life, I was unable to comlpete, and I am only now reviewing notes and finalizing the review on May 26, 2024. I apologize to the editor and authors for the delayed review.

**Summary**

This paper documents the Pleiades Glacier Observatory (PGO) program and associated data products providing elevation measurements from 2016-2017 to present for 140 sites, including a sample of ~6810 glaciers spanning nearly all glacierized regions on Earth. The authors provide some background on the program, data collection campaign design (based on community input), and methods used to prepare the datasets (stereo image processing, DEM co-registration, elevation difference correction). The geolocation accuracy of the DEM products is evaluated using external datasets (GLO-30, same-day lidar) and internal repeat measurements. Residual vertical error/uncertainty is evaluated using difference products relative to airborne lidar and repeat PGO DEM measurements over stable terrain, with an assessment of uncertainty reduction from area averaging. A final analysis considers whether the glaciers covered by PGO sites are representative of respective regional glacier mass balance. To accomplish this the authors did not compute mass balance from PGO DEMs, but sampled and aggregated subsets of existing mass balance measurements (Hugonnet et al., 2021) for the glaciers with coverage in the PGO catalog.

Overall, this is a nice paper documenting a valuable resource for the community, and I am eager to see others begin to use the PGO data products moving forward. This publication summarizes nearly a decade of work to establish and maintain the PGO monitoring program, which is no small feat.

In general, my comments are minor, mostly involving suggestions to improve wording, and clear up confusion. I offered several general suggestions and comments around best practices. I hope the authors consider these moving forward, though I do not feel these are essential for publication, and I understand the substantial effort that would be required to reprocess the entire PGO archive before initial release. It's more important to get products out, even if they are not perfect and will be improved in the future.

One of my main questions is whether TC is really the best journal for this paper. This work is relevant to the glaciology and high-mountain cryosphere/hazard communities. But much of the focus is on the data products, including methods and detailed cal/val. There are limited "cryosphere science" results in the paper, which is not a complaint, as I value data/methods papers, but it makes me wonder whether another journal (ESSD?) might be more appropriate?

For an operational observation/monitoring program like this, it is essential to use version-controlled processing and data products for provenance and understanding of potential issues and evolution of the methods (and associated products). Ideally, low-level image products (Level-1B) products would be available, low-level DEM (uncorrected) products, in addition to high-level, co-registered and corrected DEM and difference products. I also wonder about the evolution of the proprietary processing routines used by Airbus to prepare the delivered products over the years (there should at least be version numbers in the metadata that can be tracked) - presumably the current Level-0 to Level-1B processor is better than what was used 10 years ago.

Unfortunately, I was unable to access the "open" data products during my review! I followed the instructions (
https://www.legos.omp.eu/pgo/wp-content/uploads/sites/10/2024/03/Access_PGO-DSM_v6.pdf)
prepared by the authors, which state "this is when you have to wait delay (up to 7 days!) for your account to be active." After 3-4 weeks, I still did not have them appear in my account.
The different platforms are a bit confusing to navigate (though not suggesting current NASA distribution is necessarily easier). But this "custom approval" approach is really not conducive to supporting new users or easy "on-demand" data access.

[Figure]

One suggestion - I understand the budget constraints and option to define custom polygons around glaciers of interest for delivery of clipped Level-1B image products, instead of the full Level-1B images (full sensor swath). The "extra" pixels for full-width images, and resulting DEM coverage, would provide a much larger sample of "stable" terrain, which would improve

co-registration and correction beyond the relatively narrow buffer of the clipped products around priority glaciers. Perhaps an arrangement can be made with Airbus to deliver the full Level-1B images for internal processing and correction, but then PGO would only release the clipped DEMs.

**Major comments**

88-89: I disagree with the assertion that there is limited value of tri-stereo, esp for steep relief. Also Berthier et al (2014) says "There is a moderate added value of a tri-stereo" not "limited". Even if glaciers have moderate surface slopes, tri-stereo will provide improved accuracy over the adjacent exposed steep terrain needed for high-quality co-registration results with limited DEM coverage. The tri-stereo also vastly improves relative bundle adjustment for the pair and enables more sophisticated corrections (like three-line-array "jitter" correction). Yes, it may be more expensive, and it may not make sense for the PGO focus (mountain glaciers only) and limited budget, but need to separate these constraints from its inherent value for improved 3D stereo reconstruction in high-relief areas.

Consider reporting the "before" and "after" co-registration metrics over stable surfaces for each DEM - these stats are exported as json from demcoreg, along with a summary png showing DEMs, difference maps, surfaces used, and histograms. The "after" median and nmad stats can be used to provide proxy for the accuracy of each DEM (at least relative to GLO-30)

191-192: Doing co-reg with 20 m DEMs seems fine, but just note that the success depends on the roughness and relief of the scene. If the only available roughness of the surface is at small length scales of ~2-10 m, you want to use the 2 m DEMs. But this is rarely the case for mountain terrain, where significant roughness is present at ~10-1000s of m. I'm confused, however, because the text states that co-registration used 20 m DEMs, but then Figure 6 shows results for the 2 m DEMs, and Figure 5 does not state what was used.

I recommend more careful documentation of which products were used when co-registering a PGO DEM (2 m) to GLO-30 (30 m) vs. comparing two PGO DEMs (both 2 m), and how this might impact results. Fortunately, the Nuth and Kääb slope- and aspect-dependent co-registration approach should perform well for mixed or coarse resolution, as long as there are sufficient distributions of slope and aspect values over static control surfaces (AKA "stable terrain").

210-221: I don't understand why a 5th-order polynomial is needed for the cross-track direction, and there is no justification provided here. I just don't see these artifacts in Figure 4, and I worry about over-fitting and introducing bias over adjacent glaciers from a limited sample. If there is a root cause related to the CCD geometry model in the Pleaides-HR camera model, or systematic "cross-track jitter" during collection of stereo pairs, then that provides justification for high-order polynomial correction, but I'm not sure that's the case.

229-234: Is a mosaic/composite created for the 2+ DEMs over a single site for a given campaign/year? This strategy of providing many versions of difference maps seems like it could be overwhelming for the user, and the independent co-registration efforts will result in different translations for a given DEM.

Also, in terms of complication/volume - if for a given site, you have 2-3 DEMs at $t_0$, 2-3 DEMs at $t_1$, that results in potentially 9 difference maps, then x2 for the two resolutions, and x2 for the BM and SGM, so potentially 30+ products 😬

Personally, I would recommend distribution of the co-registered DEM products, one set of difference maps that you consider the "best," and then a set of tools that allow users to create and correct their own difference maps from different products if desired.

239-248: Are the observed horizontal CE90 values within the vendor spec of 8.5 m and the stated GLO30 horizontal geolocation accuracy (~4 m if I recall correctly)? I believe the 8.5m CE90 is for off-nadir angles up to 30°.

The systematic +4.5 m northward geolocation error is puzzling. It would be useful to know how the GLO-30 products were processed to "match" the DEM at each site (in local UTM zone, with height above the WGS84 ellipsoid in some ITRF realization (presumably ITRF2014 given acquisition dates) - ideally a version-controlled script in an open repo documenting this.
A map of this error for all PGO sites might reveal latitudinal dependence, distance from UTM zone center (though scaling error within the zone should not introduce errors like this), or some other cause.
Another potential explanation could be related to the sun-synchronous polar orbits and/or more N-S scan direction azimuth - perhaps timing errors or in-track pointing could result in systematic geolocation error, but it seems like Airbus would have caught and fixed this.

While some of the systematic 2.4 m vertical error could be due issues around penetration through snow in the GLO-30 products, I'm suspicious that this could be systematic error in the ASP v3.0 output (note similar vertical bias magnitude for WV/GE documented in Shean et al., 2016 and Fig 4 in Shean et al (2019) PIG paper in TC). This could likely be related to the (implementation and/or lack of) atmospheric refraction and velocity aberration correction for the version of ASP you are running, and/or some of the 2.0 m uncertainty associated with definition of the WGS84 Ensemble (EPSG:4326) vs a specific ITRF/WGS84 realization (e.g., https://spatialreference.org/ref/epsg/7664/ - see WKT2). Note that the latest ASP includes new, validated refraction/aberration corrections using the USGS Community Sensor Model, and support for PROJ9 and unambiguous output product 3D CRS definitions using WKT2.

256-263: A 30 m threshold for "unreliable" sounds high for image geolocation spec of 8.5 m CE90 combined with GLO-30 geolocation spec of ~4 m.
Ideally, one would mask the GLO-30 products before co-registraiton using the various AUX products containing error estimates and flags for artifacts, though I'm not sure about AUX layer availability for latest GLO-30 products. See example and scripts used to mask the 90-m TanDEM-X products here: https://github.com/dshean/tandemx

Earlier, the authors suggest users should use their judgment when evaluating the corrections (including co-registration and systematic bias removal), but here it sounds like there are some automated checks to apply or not apply the co-registration?  Or were these checks performed manually?  If doing manual review, maybe flag other potentially problematic difference map products. Or as with co-registration, do not apply systematic bias removal (5th order poly, spline) if the resulting output is worse?

Section 3.1.3 needs some tightening - see line-by-line comments.

I was pretty confused when I started reading about PGO mass balance from 2000-2009, since the PGO DEMs don't start until 2016-2017.  I suggest more clearly stating that you are not comparing mb measurements from PGO DEMs, but simply re-aggregating Hugonnet et al. results for a subset of glaciers with PGO coverage.

An area coverage threshold of 50% was used to identify PGO glaciers.  How was the mean mass balance computed for these glaciers?  Hypsometric interpolation?  Or just a simple mean for areas with valid coverage, which could be biased if 50% is only over ablation area or only over accumulation area. What happens if you only consdier glaciers with 95% more coverage? Do you get the same result?

The question of "Are PGO sites representative of the Earth's glaciers?" is important, but a bit subjective. They could be, but you can also get "the right answer for the wrong reasons" using an independent dataset. I think the more pertinent question for this effort might be "Are PGO mass balance measurements accurate?" I realize that many other papers have demonstrated Pleiades DEMs offer accurate glacier mass balance measurements, but it would have been nice to see some mass balance computed using the available repeat PGO DEMs compared with Hugonnet mass balance from the same period.

What percentage of the PGO glaciers already have 5-year repeat coverage?

The Figure A3 of Berthier et al., in press was cited but not available during review.

**Minor comments**

22: Consider "The PGO product consists of freely-available DEMs posted at 2 m…" and deleting the next sentence (maybe too detailed for abstract)
23: Consider "PGO stereo acquisitions began…"
35: Add commas after parenthetical citations for this list
39: Suggest "very-high-resolution (VHR, i.e., sub-meter)"
41: VHR can provide a global *sample*, it just doesn't provide continuous global coverage
57: "airborne laser scanning"
57: I tend to use the term "near-contemporaneous acquisition" but "almost simultaneously" is fine

58: "glaciated" vs. "glacierized"

58: I suggest moving the mention of the number of sites to an earlier sentence "We present the coverage achieved for 140 PGO sites…"

63: I was under the impression that the official mission/instrument name was "Pléiades-HR 1A"

66: In my opinion, the ~20 km swath makes PHR stand out among other VHR platforms with narrow swath widths (~7-13 km). I suggest you mention this here, and the value for sampling many mountain glaciers in "one snapshot"

67: Would be best to list the actual times for a typical stereo pair and triplet geometry. Is it ~30 seconds for the triplet interval?  Sounds too short for a pair with expected B/H at 7 km/s.

68: Suggest ASTER VNIR (with acronym definition), as that is the specific instrument name

69: For the record, most modern VHR systems (and medium resolution systems) are 11-12 bits these days.  Pleiades is not necessarily unique in this respect.

70: More bits don't necessarily help over textureless surfaces.  Suggest removing.  More bits help when you have very bright and very dark surfaces in the same scene and a single exposure.

75: Suggest "exceeding" instead of "reaching"

77: Suggest rewording this sentence. There is a global archive, it is just focused on urban areas and other commercial/defense priority targets.

83: Delete "well"

83: In my experience, the saturation is worse for snow-covered, equator-facing slopes in early spring, when solar incidence angles are closer to surface normal

84: check time formatting requirements for TC

87-88: Is the TDI value of 13 for 13 lines?  These units are surprising - most other sensors use 8-128 lines of TDI, with $2^n$ steps

96: I think you're convolving snow cover and snow depth here with "lowest" - the albedo of fresh snow is high, so saturation can be an issue, but a thin layer of snow cover (a few cm) is not a problem for co-registration.

104: The image is not really rejected during tasking, right?  Some real-time observations or a forecast model is used to prioritize tasking and if a site is cloudy, no image is acquired.

107: What determines the area of the site? With a 20 km swath, a 100 $km^2$ image is only 5 km long?  I'm guessing these areas refer to some user-defined polygon for the order and delivery of the stereo images.

117: Define GTN-G

118-119: Consider starting the paragraph with a summary sentence about the number of sites and coverage in each region, then talking about details of site selection

120: Check tense in these sections "can" vs. "could" - some of what is reported is based on past decisions, or summarizing what was done and results. But I think the PGO acquisitions are continuing?  If so, can continue using present tense throughout.

126: suggest putting "repeat mode" into quotes

142-143: suggest "stereo image pair"

146: remove accents from "stereo-pairs"

147: It would be useful to know more details about how ASP was run (e.g., correlator kernel sizes).  This is likely too much detail for this paper, so in addition to the Deschamps-Berger

(2020), maybe best to cite a github repo containing the scripts used to generate the PGO DEMs/orthos (for full open science and version control of PGO products moving forward).

I recommend you consider the updated MGM workflow outlined in Bhushan and Shean (https://zenodo.org/records/4554647) and Bhushan et al. (J Glac, 2024, soon to be in press, hopefully)?

147: The ASP Earth citation guidelines also recommend citing Shean et al (2016), as this is what is used for Pleiades camera models:

https://github.com/NeoGeographyToolkit/StereoPipeline?tab=readme-ov-file#citation

Suggest citing the Zenodo DOI for the specific release rather than a link to the general github page. Note that there have been several important bug fixes and improvements in more recent ASP versions (>4.0). It would be useful to consider upgrading and documenting software version number in the product metadata, rather than freezing to an older ASP version.

150: Should be "Maxar WorldView/GeoEye" recommend including Planet SkySat-C (Bhushan et al., 2021)

151: ASTER is mentioned multiple times earlier in the manuscript without definition of acronym, suggest moving to first location, and specifying VNIR instrument

153: suggest "images" rather than "satellites"

159: "highly-resolved topography" - suggest "enhanced DEM detail/quality and fewer data gaps"

167: "smoother version" for 20 m DEMs - not necessarily. This is just posted on a coarser grid.

176: Earlier, the PAN resolution was stated to be 0.7 m.

176: Do you generate orthoimages from both stereo views, or just the more nadir view? The latter seems like a good compromise, and what most users will want.

Also to clarify, the orthoimages are generated using the original 20-m BM DEM, before any co-registration? This will lead to self-consistent orthoimage, but are you then applying the resulting co-registration translation to the orthoimages? If not, then the ortho and the DEM will not be self-consistent.

The "right" way to do this is to perform a bundle adjustment, align sparse 3D triangulated matches (or an initial dense stereo DEM) to the reference elevation data, then apply the corresponding transformation to the original camera models and rerun stereo processing to create self-consistent DEMs and orthoimages. Not suggesting you need to go back and reprocess, but something consider for the future. Shashank Bhushan's latest repos include notebooks and scripts following this workflow, and we are working to wrap all of this within ASP to make it easier/faster for the user.

177-180: Could cut sentences about pansharpening that you didn't actually do, so not really relevant.

181: Geolocation performance -> should be clear that this is "The absolute geolocation accuracy"

185: "are" vs. "were" - just double-check tense throughout paper based on what is done, in archive, vs. what is ongoing.

185-186: Cite the specific version of demcoreg using Zenodo DOIs: https://zenodo.org/records/7730376.

Suggest avoiding possessive 's after Nuth and Kaab (2011)

185: Include citation for GLO-30 and/or COP30 with documentation at first mention, suggest stating the stated global vertical and horizontal accuracy metrics, and noting that accuracy decreases and there are more artifacts/blunders in steep mountain terrain.

191: Suggest considering "3D translation vector" in this section to distinguish from other co-registration approaches that involve rotation.

197: Maybe consider "product metadata report" or "product quality report" rather than "fact sheet"

206: Suggest changing "again" to something more descriptive about repeat observation for a given site after 5 years

209: Just say "using the demcoreg package, as described above"

212: Rather than "jitter" suggest "unmodeled attitude error ("jitter")" as there is also "jitter" during TDI, which is what many in the electrical engineering community think when they hear "jitter." The cause of the triangulated stereo DEM artifacts is attitude error in the metadata of one or both of the primary image products, introducing horizontal and vertical geolocation error in the resulting triangulated point.

215: It would be best to clearly indicate whether figures are showing DEMs from PHR-1A or PHR-1B throughout the text, so we remember that PHR-1B will have larger attitude error artifacts.

220-221: Users should check statistics for residuals after your corrections, and do visual inspection of the difference maps. (I think this is what you are trying to say, but "check the stable terrain" is ambiguous)

229: Suggest "entirely cover a single PGO site in a campaign year" - avoid confusion about repeat coverage.

233-234: How are these stats provided? They are not in the report provided in Appendix 1.

248: Somewhere, should state the geolocation accuracy of GLO-30.

269: "can be interpreted as residual co-registration error"

276: "quality of the PGO DEM co-registration with the reference GLO-30 product"

277: suggest "relative co-registration errors of over 10 m" rather than "residual shifts" as these are relative offsets between your two independently co-registered DEMs. See major comment about masking GLO-30 prior to co-registration - surface relief is only one factor in GLO-30 quality/accuracy.

280: suggest "...three independent airborne lidar campaigns acquired data within less than 1 day of a Pleiades stereo acquisition (Table 2)"

280-281 - Must provide citations for these ALS data (with information on how they were collected and processed), and ideally links to open data archive for the specific products/versions used. I believe these were all ACO products in Canada, not sure about Norway. Details on instrument, altitude, etc will help interpret the observed differences with PGO DEMs over surfaces with variable roughness (e.g., crevasses). While most ALS data should be significantly more accurate/detailed than PGO DEMs, there is a huge range in ALS data quality, often related to instrument specs and processing approach.

282: suggest being clear that there is negligible elevation change over all surfaces in the scenes, including glaciers, snow and exposed terrain. Can consider moving sentence from lines 285-288 here for more continuity.

282-284: I'm glad you pointed out that ALS data has ~0.1 m uncertainty. Note that the ALS errors can (and often will) be much higher than 0.1 m over steep surfaces. I would delete the "Hence the elevation difference directly reflects the uncertainties of the PGO DEMs" as this is not true. You can just say you neglect the ALS error. There is also some unknown lidar penetration in snow/ice depending on grain size and lidar wavelength (which is why we need citations with more information about the ALS data used - Near-IR vs. green)

289: I'm confused here - lidar point clouds are not triangulated. From what I can tell, the authors gridded the lidar point clouds using the ASP point2dem tool with 1 m posting.

288: Since there is another co-registration with lidar, I'm not sure why this is relevant, suggest deleting.

291: Don't need to cite Shean/NuthKaab again (has nothing to do with lidar)

291-293: I'm confused here - the authors just stated that the simultaneous products allow one to look at on-glacier and off-glacier surfaces. Why are they masking here?

293: I don't understand why this bilinear resampling was needed, and this was not mentioned during earlier co-reg with GLO-30 (much larger difference in grid cell size) - the demcoreg tool you use will do this on the fly (bicubic is default).

296-298: I think I understand why updated glacier masks were created manually here (likely better stats), but it's a bit odd to use RGI6 for masking during co-reg and then a different mask for evaluation.

279: Suggest "near-contemporaneous" instead of "simultaneous" - they may be same day, but can have time offset of several hours (potentially significant elevation difference during big melt days)

317: Suggest "As a result of the co-registration process"

319-320: There are clear residual "jitter" artifacts in this PGO DEM in Canada which likely explains most of the observed NMAD.

323-324: I see some "patches" with large apparent elevation difference over crevasses fields and other surfaces where we expect BM to perform poorly compared to SGM. Using NMAD rather than STD here will ignore these "outliers" but depending on application, "superior" is subjective (e.g., if you're studying crevasse dimensions or small-scale roughness, SGM will always be superior)

327-328: While ALS does obtain returns from beneath the canopy, if you are not isolating first/only or ground returns in the ALS point clouds, then ASP is going to produce some weighted average of all available ALS point elevations in each grid cell. You will not end up with a DTM (and I'm not sure that a citation to Piermattei should be included here).

My recommendation would be to mask the vegetation, as this affects your stats, and you need to create a DSM, and need to deal with height percentiles (e.g., 90th percentile) when gridding lidar points for comparison with stereo products. This is tricky, and an important gap that requires more attention from our community in my opinion - ALS is great, but it needs to be carefully processed when used as "truth" for stereo evaluations.

330-332: And to exclude these areas during co-registration. Demcoreg by default only uses slopes within range of 0.1-40°

332: "relief" not "reliefs"

338-339: I'm not sure you can claim "decimetric accuracy" - I interpret that to mean the accuracy is ~0.1 m. The bias is 0.0-0.2 m, and NMAD is 0.3-0.7 m, so combined is potentially closer to ~1 meter. I think "sub-meter" might be more appropriate.

341: "likely facilitate" should be "is required" or "improve" - there's no question that stable terrain is requirement for success

342: Really, it is a matter of the glacier vs. stable terrain area and distribution.  If you have a 20x40 km PGO DEM, you can obtain good results for larger glaciers and ice caps.  If you're ordering smaller ~100 km² images, max glacier size decreases.

343: The earlier section 3.1 is also attempting to estimate "uncertainty of the elevation changes". The first paragraph in this section seems to be using point measurements to do the same thing (validation) as with the lidar analysis?

I think this section is supposed to focus on PGO_DEM minus PGO_DEM error, rather than comparing against an external reference? Consider revisiting section heads and organization for how the different evaluations are presented, maybe moving the GNSS paragraph somewhere else (perhaps before the lidar section, since that is also PGO DEM minus external ALS_DEM)

345: suggest "differential GNSS measurements with centimeter accuracy" rather than "centimetric GNSS measurement"

It's not clear to me why the GNSS measurements can't be used to evaluate the DEM accuracy directly, assuming they are acquired on the same day.

353: Should explicitly state the assumption elevation difference should be 0 over "stable" terrain, and any observed residual is considered error.

375: "peculiar" word choice.  "Anomalous" may be more appropriate

375-377: Fine to throw these out if there is a clear threshold or other justification - ties in with earlier comment about thresholds to determine whether co-registration failed.

382: ASTER VNIR

387: If we had two PHR satellites dedicated to glacier observations, or a constellation of shared PHR satellites, it might.  But the limited number of available PHR (two) and competition prevents this. In the coming decade, we should have several constellations of VHR imaging satellites capable of stereo.

389: suggest "reasonable assessment of global glacier mass change"

390: include year for Hugonnet citation, and fix other instances in this section

398: "compared these regionally aggregated PGO values with corresponding values using the full sample from Hugonnet et al. (2021)"

399: Given the 2016-2017 start date for PGO, comparing with 2000-2019 values is hard to justify, even if the Hugonnet uncertainty is lowest for the longer time period.

404: "slightly more negative" is used to describe 18% more negative.  I'm not sure I would call that "slightly."

404-405: I am not clear on how 2000-2009 mb is measured using PGO DEMs that only begin in 2016.

OK, I now realize what was done here.  See major comment about improving language to remove confusion that you are comparing mb from PGO DEMs with mb from Hugonnet et al (2021).

418: "perform even better" - this could be misinterpreted as comparison with external studies. Suggest "and capture temporal evolution"

425: "...more accurate results for mean glacier mass balance" - not to be confused with earlier PGO DEM accuracy evaluation using lidar.

454: I think its important to be clear that you are evaluating calibrated/corrected PGO DEM products, not the original DEMs prepared using original Level-1B metadata

456: I know what you mean by "stable terrain as a proxy of the uncertainty on glaciers" but as written, this doesn't make sense - be clear that the "residual elevation difference values on nearby stable terrain was used to estimate corresponding uncertainty on glacier surfaces"

457: "the mean glacier-wide elevation difference has…" - be clear that this is area average, not just "elevation differences"

458-460: not sure this belongs in conclusions, as it is not discussed at all in the paper, and the focus is not on glacier mass balance.  If anything, suggest moving to "discussion" portion of section 4.

**Figures**

Suggest redefining all acronyms in figure captions, as readers skimming figures won't necessarily know that BM = Block Matching.

Figure 1: It might be useful/informative to use a color ramp for marker color to signify the number of glaciers covered at each site in the regional panels (earlier it was mentioned that most sites cover dozens of glaciers)

Figure 2: It would be useful to show at least one of the input orthoimages here, so we know *why* there are data gaps - is there a cloud, or is it fresh, textureless snow?  Note for future processing, some of the latest recommended ASP MGM options reduce the cross-hatch artifacts shown in 2b.

Figure 3: Not a requirement, but it would be nice to see metrics on errors before and after if you are going to report the translation components in a figure caption.

Figure 4: Suggest rewording caption, as you're not really showing processing steps, rather showing an example of difference map products before and after two corrections.  Maybe add titles over each column (something like "Original", "After Co-registration", "After Bias Correction") to make it easier to interpret.

Figure 5: Suggest "The figure shows translation components for the BM DEMs, as the mean and standard deviation for the SGM DEMs were nearly identical"

State whether this was the 2 m or 20 m DEMs

Is the cm unit correct here: "a few tenth of centimeters"?  This is "a few mm", no?

Figure 6: See comments about whether figures show 2 or 20 m DEMs.

Figure 7: Not clear which is BM and which is SGM - add titles or modify caption.  I think a is BM and b is SGM?

I'm a little surprised by the observed differences between c and d if the only difference is the input PGO DEM using BM (c) vs. SGM (d).  This potentially indicates that the independent co-registration approach (co-registering BM to lidar and independently co-registering SGM to

lidar) is introducing bias that could be incorrectly interpreted as due to the correlation approach (or terrain properties).

It would be useful to have one sentence summarizing the "takeaway" message for this figure. Observed elevation differences are near 0, but there are also some artifacts and differences between BM vs SGM products.

Figure 8: Since presenting signed difference values, be clear about which DEM was subtracted from the other. "Difference between" is ambiguous.

Suggest "Points show median and shaded area shows NMAD of dh values within each X m elevation bin (left) and each X degree slope bin (right)"

It's not completely clear why we would expect a relationship between dh and elevation, unless there are systematic scaling issues in the PGO DEMs. Slope is much more important, and slope and elevation are not independent (i.e., generally, higher slope values at higher elevations in these mountains).

Figure 10: Don't need 's for Hugonnet et al (2021) dataset.

**Tables**

Table 1: Suggest moving number of stereo pairs after number of sites.  In a single campaign, are there any sites with multiple stereo pairs?

Does the Glacier area column refer to orthoimage coverage or the DEM valid data coverage? The latter should be less due to data gaps from failed correlation.

Table 2: Seems odd to just report the glacier area, as you report stats for on and off glacier. Suggest including the "total area" of intersection used for co-reg/evaluation, and then glacier area.

Suggest include date and time for each collection (see earlier comment about potential melt between stereo and lidar acquisition time offset)

Suggest "Lidar density (pt/m$^2$)"

It's a bit odd to use a non-PGO DEM for this evaluation (which is not reproducible).  At least state that the processing used for the non-PGO DEM was identical to PGO DEMs.

Table 4: "balance" singular, not plural.  The PGO glaciers (this study) is a bit misleading, as it makes it sounds like you measured mass balance using PGO DEMs.  As requested in earlier comments, make sure this is clear.

**Data Availability**

468: Suggest "Licensing issues prevent open distribution of primary products and orthoimages. The PGO images are available…

Have you considered distributing the entire archive through AWS Open Data Registry (like ArcticDEM/REMA) or other on-demand, direct-access cloud provider, rather than the current combination of university portals? The ability to subset and stream from COGs in the cloud is much more efficient than downloading entire products (hence the cloud-optimized GeoTiff 🙂)

As requested in a few places, for open science and reproducibility, it would be best to distribute all processing and analysis scripts in a version-controlled public archive, so PGO users can

understand how the products were created, and how the processing has evolved (and whether they should download a new version of their DEM).

Appendix 1
Great to see this pdf bundled with each product.
Consider including a context map for the site here in addition to the browse image.
Our team (led by Ben Purinton) is working to centralize scripts (https://github.com/uw-cryo/asp_plot) to generate standardized "quality report" pdfs for ASP output. It would be great to work together to develop a community package and integrate within core ASP tools.

---

## Author Comment (AC2)

**Reply to reviewer#2**

**Response: We thank David Shean for providing a thorough and constructive review of our manuscript. Please find below our preliminary answers to the main comments. The response to the specific/minor comments will be detailed in our final response letter if the editor considers that our manuscript is appropriate for The Cryosphere.**

**Summary (including already major comments)**

One of my main questions is whether TC is really the best journal for this paper. This work is relevant to the glaciology and high-mountain cryosphere/hazard communities. But much of the focus is on the data products, including methods and detailed cal/val. There are limited "cryosphere science" results in the paper, which is not a complaint, as I value data/methods papers, but it makes me wonder whether another journal (ESSD?) might be more appropriate?

**Response: Our rationale to target The Cryosphere is that our dataset is specifically designed for and of interest to the glaciological community. We note that the REMA paper by Howat et al. (https://doi.org/10.5194/tc-13-665-2019) had a similar scope (i.e. "data products, including methods and detailed cal/val"), did not include glaciological findings and was published also in The Cryosphere. It is actually one of the most cited articles in The Cryosphere. Our strong preference is to keep our manuscript in The Cryosphere should the editor decide it is publishable and within scope.**

For an operational observation/monitoring program like this, it is essential to use version-controlled processing and data products for provenance and understanding of potential issues and evolution of the methods (and associated products). Ideally, low-level image products (Level-1B) products would be available, low-level DEM (uncorrected) products, in addition to high-level, co-registered and corrected DEM and difference products. I also wonder about the evolution of the proprietary processing routines used by Airbus to prepare the delivered products over the years (there should at least be version numbers in the metadata that can be tracked) - presumably the current Level-0 to Level-1B processor is better than what was used 10 years ago.

**Response: As described in our data availability statement, we are not allowed to redistribute low-level Pléiades imagery. This should change by the end of 2024 (or rather sometimes in 2025 to be more realistic) with a program named the Pléiades World heritage (PWH, described very briefly here: https://dinamis.data-terra.org/acces-aux-donnees/) thanks to which the full Pléiades archive will be opened to the academic sector (details of licensing to be confirmed). This will satisfy the request of expert users, who might wish to process low-level data. We stress that the PGO was designed to satisfy the need of fellow glaciologists directly interested in the coregistered DEMs. We note that the coregistration of the PGO DEMs to GLO-30 is a simple translation so a user could, if needed, easily obtain the uncorrected DEMs.**

Unfortunately, I was unable to access the "open" data products during my review! I followed the instructions prepared by the authors, which state "this is when you have to wait a delay (up to 7 days!) for your account to be active." After 3-4 weeks, I still did not have them appear in my account. The different platforms are a bit confusing to navigate (though not suggesting current NASA distribution is necessarily easier). But this "custom approval" approach is really not conducive to supporting new users or easy "on-demand" data access.

**Response: It was unfortunate that the reviewer could not access these data in an easy manner. Due to licensing constraints, we did not have a lot of freedom to choose our**

**distribution platform. We apologize to the referee and would have gladly shown the reviewer how to solve this data access issue. This lag of several days between the initial request and the final access should be solved by the end of year 2024. Two "sample" products (in the Alps and Tibetan Plateau) are available until 7 July 2024 here: https://filesender.renater.fr/?s=download&token=50321ce9-ab69-4ddd-a5e2-56dbb4e0de 36**

One suggestion - I understand the budget constraints and option to define custom polygons around glaciers of interest for delivery of clipped Level-1B image products, instead of the full Level-1B images (full sensor swath). The "extra" pixels for full-width images, and resulting DEM coverage, would provide a much larger sample of "stable" terrain, which would improve co-registration and correction beyond the relatively narrow buffer of the clipped products around priority glaciers. Perhaps an arrangement can be made with Airbus to deliver the full Level-1B images for internal processing and correction, but then PGO would only release the clipped DEMs.

**Response: Budget constraints were unavoidable, and we preferred to cover more glaciers than order entire images with a lot of off-glacier terrain. In the long term, it will be one of the added values of the PGO to have "forced" the Pléiades satellites to acquire imagery over glaciers. These full level-1B images are now stored in the image catalogue and will be available, at some point (see Pléiades World Heritage program). This was important because Airbus does not aim at building an archive: they acquire images on demand from the customers (and commercial requests are rarely on glaciers).**

**Major comments (including also some specific comments)**

limited value of tri-stereo

**Response: We are unaware of studies that show tri-stereo yield clearly superior results on natural terrain (not urban), but we would be happy to include those and modify the statement if the referee knows of studies that demonstrate this.**

"before" and "after" co-registration metrics

**Response: These metrics are distributed with the data product (that the reviewer could unfortunately not access). See sample products now available on a FTP server.**

2m vs 20 m coregistration

**Response: We will clarify the text (and possible change Figure 6) to make sure there is no confusion**

I recommend more careful documentation of which products were used when co-registering a PGO DEM (2 m) to GLO-30 (30 m) vs. comparing two PGO DEMs (both 2 m), and how this might impact results.

**Response: This general comment was not clear to us. Maybe one of the specific comments will help us address this? We will make sure we clarify the specification of the GLO-30 DEM and the workflow for DEM coregistration.**

5th-order polynomial is needed for the cross-track direction.

**Response: It is true that the across track biases are generally smaller than along track biases. In our full response (if the editor agrees our article is suitable to TC), we will show several examples where the spatially-structured biases should be corrected. A fifth-order polynomial performed suitably well to remove bias. We note that the corrections are all graphically (.jpg or .png) illustrated in the data products downloaded by the users. In case**

**those generic corrections are not satisfying (and this is indeed sometimes the case), the user can easily come back to the DSM.**

mosaic/composite created for the 2+ DEMs over a single site for a given campaign/year?

**Response: We prefer not to create mosaics at the cost of distributing many products. The Pléiades DSM are sometimes acquired several months apart during a period of strong melt (and hence elevation changes) so we prefer to leave it to the users to make the mosaic and maybe apply a seasonal correction.**

Are the observed horizontal CE90 values within the vendor spec of 8.5 m and the stated GLO30 horizontal geolocation accuracy (~4 m if I recall correctly)? I believe the 8.5m CE90 is for off-nadir angles up to 30°.

**Response: We did not compute CE90 but will check this during revision.**

The systematic +4.5 m northward geolocation error is puzzling AND systematic 2.4 m vertical error

**Response: We agree. We will check with CNES and Airbus DS to see if they made similar observations while following the geometric performance of the Pleiades satellites. We hope they will be able to share their knowledge for a satellite mission which is ageing and hence not their priority. Our plan is to continue to explore the source of these biases during the revision of our manuscript.**

A 30 m threshold for "unreliable" sounds high for image geolocation spec of 8.5 m CE90 combined with GLO-30 geolocation spec of ~4 m

**Response: We will share examples of the dem_align.py outputs where shifts of 15-20 m were found and did not result from lack of stable terrain or unwrapping errors in GLO-30. The observation of such shifts justifies the use of a 30-m threshold. Maybe the geolocation performances of Pléiades or GLO-30 cited by the reviewer do not hold everywhere on Earth? And a CE90 value of 8.5 m also leaves room for a few cases for which the horizontal offset will reach two to three times this amount.**

Ideally, one would mask the GLO-30 products before co-registraiton using the various AUX products containing error estimates and flags for artifacts, though I'm not sure about AUX layer availability for latest GLO-30 products

**Response: There are indeed some quality layers for GLO-30 products, in particular a Heigh Error Mask. However, as stated in the documentation "these are random errors and do not include any kind of systematic errors, such as elevation offsets related to erroneous orbital parameters." So the main artefacts in the GLO-30 DEM (from phase unwrapping errors) are not flagged in the auxiliary data. We think that the DEM coregistration (based on a large number of pixels) is sufficiently robust to outliers so that the results should not be affected by these pixels with lower confidence. We also note that coregistering again the entire database of DEMs is a major endeavour and justified only if significant improvements are achieved.**

Earlier, the authors suggest users should use their judgment when evaluating the corrections (including co-registration and systematic bias removal), but here it sounds like there are some automated checks to apply or not apply the co-registration? Or were these checks performed manually?

**Response: There were manual checks. This is also why we provide some plots (including the output of dem_align.py) so that the user knows the corrections that were applied and how they impacted the residuals off glacier. We propose to include a few such plots in the appendix of our article.**

I was pretty confused when I started reading about PGO mass balance from 2000-2009, since the PGO DEMs don't start until 2016-2017. I suggest more clearly stating that you are not comparing mb measurements from PGO DEMs, but simply re-aggregating Hugonnet et al. results for a subset of glaciers with PGO coverage.

**Response: We acknowledge that this section was confusing and will be reworked.**

An area coverage threshold of 50% was used to identify PGO glaciers. How was the mean mass balance computed for these glaciers? Hypsometric interpolation? Or just a simple mean for areas with valid coverage, which could be biased if 50% is only over ablation area or only over accumulation area. What happens if you only consider glaciers with 95% more coverage? Do you get the same result?

**Response: We only use the 50% coverage to decide if a glacier is included or not in our subsequent analysis. Then no interpolation has to be performed by us because the mass balance data comes from Hugonnet et al. and is not recalculated from the elevation change maps (neither from PGO nor from Hugonnet et al. dh maps). Our revisions will ensure this point is clear to the reader.**

The question of "Are PGO sites representative of the Earth's glaciers?" is important, but a bit subjective. They could be, but you can also get "the right answer for the wrong reasons" using an independent dataset. I think the more pertinent question for this effort might be "Are PGO mass balance measurements accurate?" I realize that many other papers have demonstrated Pleiades DEMs offer accurate glacier mass balance measurements, but it would have been nice to see some mass balance computed using the available repeat PGO DEMs compared with Hugonnet mass balance from the same period.

**Response: The question "Are PGO mass balance measurements accurate?" is partly answered by our analysis of the uncertainty of the elevation change maps using Lidar or "stable" terrain. Of course, one would have to properly propagate these elevation change uncertainties to take into account other sources of errors from the density conversion factor and the outlines. We are reluctant to use the Hugonnet et al. elevation change maps to evaluate the PGO ones because (i) the time periods differ and (ii) the precision of the PGO dh maps are in most cases superior. See for example the comparison of the elevation change maps on Meighen Ice Cap (Figure 4 in https://doi.org/10.1088/1361-6633/acaf8e) and evaluation of the glacier-wide mean elevation change for several glaciers in Norway (Fig. 15 in https://doi.org/10.1017/aog.2023.70)**

What percentage of the PGO glaciers already have 5-year repeat coverage?

**Response: As this percentage is evolving every semester, we do not think this is really useful for the article itself.**

The Figure A3 of Berthier et al., in press was cited but not available during review.

**Response: The figure A3 is available: https://doi.org/10.1017/jog.2023.100**

---

## Author Response (AR1)

Etienne Berthier
14 av Ed Belin
31400 Toulouse
etienne.berthier@univ-tlse3.fr

29 July 2024

Dear Editor,

Please find enclosed a revised version of our manuscript (MS) entitled, "The Pléiades Glacier Observatory: high resolution digital elevation models and ortho-imagery to monitor glacier change". To facilitate your assessment, we uploaded a clean and track-change version of the revised MS.

We thank the editor and the reviewers for their positive and constructive evaluation of our study. Below, you will find a copy of all comments and, in blue, a point-by-point response to them. When relevant, the revised text is provided in italics.

We hope that these corrections/clarifications make our paper suitable for publication in *The Cryosphere*.

   Yours sincerely,

Etienne Berthier on behalf of the co-authors

**Reply to reviewer#1**

In their TCD manuscript "The Pléiades Glacier Observatory: high-resolution digital elevation models and ortho-imagery to monitor glacier change" Berthier et al. introduce a contentiously growing topographic dataset of 140 glacierized areas around the world termed the Pléiades Glacier Observatory (PGO). For each region, one Pleiades stereo acquisition is acquired every 5 years to monitor the glacier's state and health at the globally distributed benchmark sites. The data will be processed to Digital Elevation Models (DEMs) and orthoimagery and will be made publicly available.

This is a very valuable dataset for the glaciological society to document ongoing glacier changes around the world and I want to thank the authors for making this happen. Overall the manuscript is well structured and written and I suggest publication after correcting some minor points which I list in the following.

**We thank the anonymous reviewer for the positive review of our manuscript and kind comments.**

Specific comments:

L19-20: short general information about the Pleiades satellite mission should be included in the abstract. **We added "*Pléiades 0.7-m satellite stereo pairs*". Not much space in the abstract to provide more details about Pléiades but we note, Section 2.1 is dedicated to this.**

L27-28: is this shown in the manuscript? (Well, yes in section 4) Can you quantify it here?

**We added "*(within 0.07 m w.e./yr)*" to make the statement more quantitative.**

L39: what about InSAR and altimetry? Maybe include a brief intro in this paragraph or at least that these methods also exist?

**We focus the introduction on optical imagery. We could discuss INSAR, altimetry (radar and laser) and also gravimetry, but to keep the introduction brief and focused, we prefer to maintain the emphasis on optical sensors. We do, however, refer to a new review paper (Berthier et al., 2023) that describes these other approaches.**

L73: here also Holzer et al., 2015 could be cited. **Good point, reference added.**

L77-81: this is important and I and probably the entire glaciological community are very thankful for your efforts. **Thanks for such an encouraging statement.**

L185: What are the specifications of the Copernicus GLO-30 DEM (e.g. resolution (from the naming I guess 30 m...), accuracy etc.)? **We now provide the ground sampling distance, absolute vertical and horizontal accuracy and the doi.**

L239: off-glacier I presume, but maybe write it explicitly. **A good point. We added "*off-glacier*" in the text and in the legend of Figure 5.**

L246-L248: I agree with this point and it also raises the question why the authors did not coregister to a more precise dataset such as ICESat/ICESat-2 altimetry? Is there a particular reason? I could imagine track spacing and surface slope are an issue?

**The reviewer is right that summer laser altimetry data would be ideal, at least to correct a final vertical shift. However, especially at low latitudes, the track spacing is such that few to no laser elevation measurements would intersect with the Pléiades footprint (max 20 km wide but often much less given the shape of our sites).**

L396: what is GTN-G again? **"global terrestrial network for glaciers". Added in the text. See https://www.gtn-g.ch/**

**Reply to reviewer#2, David Shean**

Note: I initially completed a detailed review of the paper on March 20, 2024. Unfortunately, due to a number of factors in professional and personal life, I was unable to comlpete, and I am only now reviewing notes and finalizing the review on May 26, 2024. I apologize to the editor and authors for the delayed review.

**We thank D. Shean for the time taken to perform a very detailed review. Some of his comments are now even influencing the work of the "geometric image quality" team at CNES.**

Summary

This paper documents the Pleiades Glacier Observatory (PGO) program and associated data products providing elevation measurements from 2016-2017 to present for 140 sites, including a sample of ~6810 glaciers spanning nearly all glacierized regions on Earth. The authors provide some background on the program, data collection campaign design (based on community input), and methods used to prepare the datasets (stereo image processing, DEM co-registration, elevation difference correction). The geolocation accuracy of the DEM products is evaluated using external datasets (GLO-30, same-day lidar) and internal repeat measurements. Residual vertical error/uncertainty is evaluated using difference products relative to airborne lidar and repeat PGO DEM measurements over stable terrain, with an assessment of uncertainty reduction from area averaging. A final analysis considers whether the glaciers covered by PGO sites are representative of respective regional glacier mass balance.

To accomplish this the authors did not compute mass balance from PGO DEMs, but sampled and aggregated subsets of existing mass balance measurements (Hugonnet et al., 2021) for the glaciers with coverage in the PGO catalog.

Overall, this is a nice paper documenting a valuable resource for the community, and I am eager to see others begin to use the PGO data products moving forward. This publication summarizes nearly a decade of work to establish and maintain the PGO monitoring program, which is no small feat.

In general, my comments are minor, mostly involving suggestions to improve wording, and clear up confusion. I offered several general suggestions and comments around best practices. I hope the authors consider these moving forward, though I do not feel these are essential for publication, and I understand the substantial effort that would be required to reprocess the entire PGO archive before initial release. It's more important to get products out, even if they are not perfect and will be improved in the future. One of my main questions is whether TC is really the best journal for this paper. This work is relevant to the glaciology and high-mountain cryosphere/hazard communities. But much of the focus is on the data products, including methods and detailed cal/val. There are limited "cryosphere science" results in the paper, which is not a complaint, as I value data/methods papers, but it makes me wonder whether another journal (ESSD?) might be more appropriate?

**Our rationale to target *The Cryosphere* (TC) is that our dataset is specifically designed for and of interest to the glaciological community. We note that the REMA paper by Howat et al. (https://doi.org/10.5194/tc-13-665-2019) had a similar scope (i.e. "data products, including methods and detailed cal/val"), did not include glaciological findings and was published also in TC. It is actually one of the most cited articles in this journal. Our strong preference is to keep our manuscript in TC and we are happy that the editor decided it fits within scope.**

For an operational observation/monitoring program like this, it is essential to use version-controlled processing and data products for provenance and understanding of potential issues and evolution of the methods (and associated products). Ideally, low-level image products (Level-1B) products would be available, low-level DEM (uncorrected) products, in addition to high-level, co-registered

and corrected DEM and difference products. I also wonder about the evolution of the proprietary processing routines used by Airbus to prepare the delivered products over the years (there should at least be version numbers in the metadata that can be tracked) - presumably the current Level-0 to Level-1B processor is better than what was used 10 years ago.

**As described in our data availability statement, we are not allowed to redistribute low-level Pléiades imagery. This should change by the end of 2024 (or rather sometimes in 2025 to be more realistic) with a program named the Pléiades World heritage (PWH, described very briefly here: https://dinamis.data-terra.org/acces-aux-donnees/) thanks to which the full Pléiades archive will be opened to the academic sector (details of licensing to be confirmed). This will satisfy the request of expert users, who might wish to process low-level data. We stress that the PGO was designed to satisfy the needs of fellow glaciologists directly interested in the coregistered DEMs. We note that the coregistration of the PGO DEMs to GLO-30 is a simple translation so a user could, if needed, easily obtain the uncorrected DEMs.**

Unfortunately, I was unable to access the "open" data products during my review! I followed the instructions (

https://www.legos.omp.eu/pgo/wp-content/uploads/sites/10/2024/03/Access_PGO-DSM_v6.pdf) prepared by the authors, which state "this is when you have to wait delay (up to 7 days!) for your account to be active." After 3-4 weeks, I still did not have them appear in my account. The different platforms are a bit confusing to navigate (though not suggesting current NASA distribution is necessarily easier). But this "custom approval" approach is really not conducive to supporting new users or easy "on-demand" data access.

**It was unfortunate that David Shean could not access these data in an easy manner. Due to licensing constraints, we did not have a lot of freedom to choose our distribution platform. We apologize to the referee and would have gladly demonstrated how to solve this data access issue. This lag of several days between the initial request and the final access should be solved by the end of year 2024. Two "sample" products (in the Alps and Tibetan Plateau) are available until 28 August 2024:**

 **https://filesender.renater.fr/?s=download&token=b143ff4c-70e4-4dd4-84f3-24924d8e1fe0**

One suggestion - I understand the budget constraints and option to define custom polygons around glaciers of interest for delivery of clipped Level-1B image products, instead of the full Level-1B images (full sensor swath). The "extra" pixels for full-width images, and resulting DEM coverage, would provide a much larger sample of "stable" terrain, which would improve co-registration and correction beyond the relatively narrow buffer of the clipped products around priority glaciers. Perhaps an arrangement can be made with Airbus to deliver the full Level-1B images for internal processing and correction, but then PGO would only release the clipped DEMs.

**Budget constraints were unavoidable, and we preferred to cover more glaciers than order entire images with a lot of off-glacier terrain. In the long term, it will be one of the added values of the PGO to have "forced" the Pléiades satellites to acquire imagery over glaciers. These full level-1B images are now stored in the image catalogue and will be available, at some point (see above comment related to the Pléiades World Heritage program). This is important because Airbus does not aim at building an archive: they acquire images on demand from the customers (and commercial requests are rarely on glaciers).**

**Major comments**

88-89: I disagree with the assertion that there is limited value of tri-stereo, esp for steep relief. Also Berthier et al (2014) says "There is a moderate added value of a tri-stereo" not "limited". Even if glaciers have moderate surface slopes, tri-stereo will provide improved accuracy over the adjacent exposed steep terrain needed for high-quality co-registration results with limited DEM coverage.

The tri-stereo also vastly improves relative bundle adjustment for the pair and enables more sophisticated corrections (like three-line-array "jitter" correction). Yes, it may be more expensive, and it may not make sense for the PGO focus (mountain glaciers only) and limited budget, but need to separate these constraints from its inherent value for improved 3D stereo reconstruction in high-relief areas.

**We changed "limited" to "moderate", indeed reflecting more appropriately the conclusion of the 2014 study.**

Consider reporting the "before" and "after" co-registration metrics over stable surfaces for each DEM - these stats are exported as json from demcoreg, along with a summary png showing DEMs, difference maps, surfaces used, and histograms. The "after" median and nmad stats can be used to provide proxy for the accuracy of each DEM (at least relative to GLO-30).

**These metrics are distributed with the data product which was part of the datasets that Dr. Shean could not access. See sample products now available on a FTP server until 28 August 2024**

**https://filesender.renater.fr/?s=download&token=b143ff4c-70e4-4dd4-84f3-24924d8e1fe0**

191-192: Doing co-reg with 20 m DEMs seems fine, but just note that the success depends on the roughness and relief of the scene. If the only available roughness of the surface is at small length scales of ~2-10 m, you want to use the 2 m DEMs. But this is rarely the case for mountain terrain, where significant roughness is present at ~10-1000s of m. I'm confused, however, because the text states that co-registration used 20 m DEMs, but then Figure 6 shows results for the 2 m DEMs, and Figure 5 does not state what was used.

**We changed the text in section 2.3.2: "first the most recent Pléiades DEM is coregistered to the older one (derived at the same ground sampling distance and using the same correlator) on the stable terrain". This is to clarify that when computing PGO elevation change maps, the coregistration is done on the oldest Pléiades DEM using the same correlator and at the same ground sampling distance. This is why in Figure 6 we show the residuals using the 2 m DEMs.**

I recommend more careful documentation of which products were used when co-registering a PGO DEM (2 m) to GLO-30 (30 m) vs. comparing two PGO DEMs (both 2 m), and how this might impact results. Fortunately, the Nuth and Kääb slope- and aspect-dependent co-registration approach should perform well for mixed or coarse resolution, as long as there are sufficient distributions of slope and aspect values over static control surfaces (AKA "stable terrain").

**We clarified the specification of the GLO-30 DEM and the workflow for DEM coregistration.**

210-221: I don't understand why a 5th-order polynomial is needed for the cross-track direction, and there is no justification provided here. I just don't see these artifacts in Figure 4, and I worry about over-fitting and introducing bias over adjacent glaciers from a limited sample. If there is a root cause related to the CCD geometry model in the Pleaides-HR camera model, or systematic "cross-track jitter" during collection of stereo pairs, then that provides justification for high-order polynomial correction, but I'm not sure that's the case.

**We confirm that across track biases are generally smaller than along track biases. Below, we show several examples where the spatially-structured biases should, we think, be corrected. A fifth-order polynomial performed suitably well to remove bias as used in Gardelle et al. (2013, http://dx.doi.org/10.5194/tc-7-1263-2013). We note that the corrections are all graphically (.jpg or .png) illustrated in the data products downloaded by the users. In case those generic corrections are not satisfying (and this is indeed sometimes the case), the user can easily come back to the uncorrected DEM and apply the relevant correction.**

[Figure]

Figure R1. Three examples where a 5th order polynomial fit to the residuals off glacier was applied to reduce the uncertainties in the PGO DEM.
The examples are for product Gulkana_ALA_2016-09-01_2128501_2021-09-09_2122269_SGM_2m (upper left), Lombardy_CEU_2017-08-01_1032561_2022-08-21_1033056_SGM_2m (upper right) and Olsen_GRL_2017-07-17_1358160_2022-07-22_1413096_SGM_2m (lower left).

229-234: Is a mosaic/composite created for the 2+ DEMs over a single site for a given campaign/year? This strategy of providing many versions of difference maps seems like it could be overwhelming for the user, and the independent co-registration efforts will result in different translations for a given DEM. Also, in terms of complication/volume - if for a given site, you have 2-3 DEMs at t 0 , 2-3 DEMs at t 1, that results in potentially 9 difference maps, then x2 for the two resolutions, and x2 for the BM and SGM, so potentially 30+ products 😬 Personally, I would recommend distribution of the co-registered DEM products, one set of difference maps that you consider the "best," and then a set of tools that allow users to create and correct their own difference maps from different products if desired.

**We prefer not to create mosaics at the cost of distributing many products. The Pléiades DSM are sometimes acquired with a seasonal offset of several months during a period of strong melt (and hence elevation changes) so we prefer to leave it to the users to make the mosaic and maybe apply a seasonal correction.**

239-248: Are the observed horizontal CE90 values within the vendor spec of 8.5 m and the stated GLO30 horizontal geolocation accuracy (~4 m if I recall correctly)? I believe the 8.5m CE90 is for off-nadir angles up to 30°.

**We computed horizontal CE90 and found values of 16.7 m for Pléiades 1A and 5.8 m for Pléiades 1B. The better location of Pléiades 1B is well-known (Lebegue et al., 2015, cited in the article). See also the response below.**

The systematic +4.5 m northward geolocation error is puzzling. It would be useful to know how the GLO-30 products were processed to "match" the DEM at each site (in local UTM zone, with height above the WGS84 ellipsoid in some ITRF realization (presumably ITRF2014 given acquisition dates) - ideally a version-controlled script in an open repo documenting this. A map of this error for all PGO sites might reveal latitudinal dependence, distance from UTM zone center (though scaling error

within the zone should not introduce errors like this), or some other cause. Another potential explanation could be related to the sun-synchronous polar orbits and/or more N-S scan direction azimuth - perhaps timing errors or in-track pointing could result in systematic geolocation error, but it seems like Airbus would have caught and fixed this.

**We thank David Shean for asking us to investigate this unexpected systematic offset. This has led to more analysis and many exchanges with colleagues from the French Space Agency (CNES).**

**First and most importantly for the PGO article and database, we could verify that GLO-30 was a safe reference, did not exhibit any large horizontal shift and that there was no issue with coregistration. We assessed this in Svalbard, where we could verify that the shift vectors (dX and dY) were similar (within 1-2 m) when co-registering a dozen of PGO DEMs to the 30-m Cop-30 DEM, the 10-m Arctic DEM mosaic or the local 20-m DEM from the North Polar Institute (https://data.npolar.no/dataset/dce53a47-c726-4845-85c3-a65b46fe2fea).**

**An analysis of the northward geolocation error (dNorth) with latitude has revealed that the shift is mainly due to Pléiades 1A and concentrated at high latitudes (Figure R2). The image quality team at CNES is now looking into that carefully. They suggest that this could be due to an empirical correction of the image geolocation that has a dependency with latitude and is poorly-constrained at polar latitudes due to the lack of validation sites.**

**See revised section 3.1.1.**

[Figure]

**Figure R2: Geolocation error as a function of latitude for 135 PGO DEMs derived from Pléiades 1A (left) and 123 DEMs derived from Pléiades 1B.**

While some of the systematic 2.4 m vertical error could be due issues around penetration through snow in the GLO-30 products, I'm suspicious that this could be systematic error in the ASP v3.0 output (note similar vertical bias magnitude for WV/GE documented in Shean et al., 2016 and Fig 4 in Shean et al (2019) PIG paper in TC). This could likely be related to the (implementation and/or lack of) atmospheric refraction and velocity aberration correction for the version of ASP you are running, and/or some of the 2.0 m uncertainty associated with definition of the WGS84 Ensemble (EPSG:4326) vs a specific ITRF/WGS84 realization (e.g., https://spatialreference.org/ref/epsg/7664/ - see WKT2). Note that the latest ASP includes new, validated refraction/aberration corrections using the USGS Community Sensor Model, and support for PROJ9 and unambiguous output product 3D CRS definitions using WKT2.

**We computed one of the DEMs using the last version of ASP at the time of writing (v3.4) and found almost no changes (within 1-2 cm in height). We computed the mean vertical error for Pléiades 1A and 1B and found it was more negative (-3.9 m) for 1B than for 1A (-1.1 m). Hence, a**

**sensor specific correction may be partly responsible for the systematic vertical error and this is also under investigation at CNES. See revised section 3.1.1.**

256-263: A 30 m threshold for "unreliable" sounds high for image geolocation spec of 8.5 m CE90 combined with GLO-30 geolocation spec of ~4 m.

**The above discussion shows that the 8.5 m CE90 for geolocation does not hold for Pléiades 1A and, hence, we feel that this max 30 m shift is justified.**

Ideally, one would mask the GLO-30 products before co-registraiton using the various AUX products containing error estimates and flags for artifacts, though I'm not sure about AUX layer availability for latest GLO-30 products. See example and scripts used to mask the 90-m TanDEM-X products here: https://github.com/dshean/tandemx

**There are indeed some quality layers for GLO-30 products, in particular a Heigh Error Mask. However, as stated in the documentation "these are random errors and do not include any kind of systematic errors, such as elevation offsets related to erroneous orbital parameters." So the main artefacts in the GLO-30 DEM (from phase unwrapping errors) are not flagged in the auxiliary data. We think that the DEM coregistration (based on a large number of pixels) is sufficiently robust to outliers so that the results should not be affected by these pixels with lower confidence. As discussed above, for a dozen of DEMs in Svalbard, we found a similar shift vectors (within 1-2 m) using GLO-30, Arctic DEM and the local DEM from the North Polar Institute. We also note that coregistering the entire database of DEMs a second time would be a major endeavour and justified only if significant improvements are achieved.**

Earlier, the authors suggest users should use their judgment when evaluating the corrections (including co-registration and systematic bias removal), but here it sounds like there are some automated checks to apply or not apply the co-registration? Or were these checks performed manually? If doing manual review, maybe flag other potentially problematic difference map products. Or as with co-registration, do not apply systematic bias removal (5th order poly, spline) if the resulting output is worse?

**A coregistration was considered as unreliable when the absolute shift vector was larger than 30 m. So this is an automated check.**

Section 3.1.3 needs some tightening - see line-by-line comments.

**Thank you. This section was revised according to the specific comments.**

I was pretty confused when I started reading about PGO mass balance from 2000-2009, since the PGO DEMs don't start until 2016-2017. I suggest more clearly stating that you are not comparing mb measurements from PGO DEMs, but simply re-aggregating Hugonnet et al. results for a subset of glaciers with PGO coverage.

**We reworked this section to clarify that all mass balances are from the Hugonnet et al. (2021) dataset. We added "*We clarify here that, in this entire analysis, none of the mass balances were derived from PGO elevation change maps. All mass balances are from the Hugonnet et al. (2021) database.*"**

An area coverage threshold of 50% was used to identify PGO glaciers. How was the mean mass balance computed for these glaciers? Hypsometric interpolation? Or just a simple mean for areas with valid coverage, which could be biased if 50% is only over ablation area or only over accumulation area. What happens if you only consdier glaciers with 95% more coverage? Do you get the same result?

**We only use the 50% coverage (by PGO) to decide if a glacier should be included or not in our subsequent analysis. No interpolation (gap filling) was performed by us because the mass**

**balance data comes from Hugonnet et al. 2021 and is not recalculated from the elevation change maps (neither from PGO nor from Hugonnet et al. dh maps). Our revisions should make this point clear to the reader (see answer to the above comment).**

The question of "Are PGO sites representative of the Earth's glaciers?" is important, but a bit subjective. They could be, but you can also get "the right answer for the wrong reasons" using an independent dataset. I think the more pertinent question for this effort might be "Are PGO mass balance measurements accurate?" I realize that many other papers have demonstrated Pleiades DEMs offer accurate glacier mass balance measurements, but it would have been nice to see some mass balance computed using the available repeat PGO DEMs compared with Hugonnet mass balance from the same period.

**The reviewer questions "Are PGO mass balance measurements accurate?" is partly answered by our analysis of the uncertainty of the elevation change maps using Lidar or "stable" terrain. Of course, one would have to properly propagate these elevation change uncertainties to take into account other sources of errors from the density conversion factor and the outlines. We are reluctant to use the Hugonnet et al. (2021) elevation change maps to evaluate the PGO ones because (i) the time periods differ and (ii) the precision of the PGO dh maps are in most cases vastly superior to the ASTER ones. See for example the comparison of the elevation change maps on Meighen Ice Cap (Figure 4 in Berthier et al. 2023 https://doi.org/10.1088/1361-6633/acaf8e) and evaluation of the glacier-wide mean elevation change for several glaciers in Norway (Fig. 15 in Andreassen et al. 2023 https://doi.org/10.1017/aog.2023.70)**

What percentage of the PGO glaciers already have 5-year repeat coverage?

**As this percentage is evolving every semester, we do not think this is really useful for the article itself.**

The Figure A3 of Berthier et al., in press was cited but not available during review.

**Sorry. The figure A3 is available: https://doi.org/10.1017/jog.2023.100**

**Minor comments**

22: Consider "The PGO product consists of freely-available DEMs posted at 2 m…" and deleting the next sentence (maybe too detailed for abstract). **Agreed**

23: Consider "PGO stereo acquisitions began…". **Agreed**

35: Add commas after parenthetical citations for this list. **Added**

39: Suggest "very-high-resolution (VHR, i.e., sub-meter)". **Agreed**

41: VHR can provide a global sample, it just doesn't provide continuous global coverage. **"continuous" added**

57: "airborne laser scanning". **Added**

57: I tend to use the term "near-contemporaneous acquisition" but "almost simultaneously" is fine. **Changed as suggested**

58: "glaciated" vs. "glacierized". **Glacierized is indeed more appropriate**

58: I suggest moving the mention of the number of sites to an earlier sentence "We present the coverage achieved for 140 PGO sites…" **Changed as suggested**

63: I was under the impression that the official mission/instrument name was "Pléiades-HR 1A". **Even in official pages of CNES, -HR is omitted so, for the sake of concision, we think it is not necessary here.**

66: In my opinion, the ~20 km swath makes PHR stand out among other VHR platforms with narrow swath widths (~7-13 km). I suggest you mention this here, and the value for sampling many mountain glaciers in "one snapshot". **The 20-km swath is now mentioned.**

67: Would be best to list the actual times for a typical stereo pair and triplet geometry. Is it ~30 seconds for the triplet interval? Sounds too short for a pair with expected B/H at 7 km/s. **Here we mentioned just stereo and provide a 40s time interval corresponding to a base to height value of 0.4**

68: Suggest ASTER VNIR (with acronym definition), as that is the specific instrument name. **Visible and Near-Infrared added**

69: For the record, most modern VHR systems (and medium resolution systems) are 11-12 bits these days. Pleiades is not necessarily unique in this respect. **OK**

70: More bits don't necessarily help over textureless surfaces. Suggest removing. More bits help when you have very bright and very dark surfaces in the same scene and a single exposure. **Removed**

75: Suggest "exceeding" instead of "reaching" **changed**

77: Suggest rewording this sentence. There is a global archive, it is just focused on urban areas and other commercial/defense priority targets. **Changed to *comprehensive***

83: Delete "well" **deleted**

83: In my experience, the saturation is worse for snow-covered, equator-facing slopes in early spring, when solar incidence angles are closer to surface normal. **Thanks, sentence reworked.**

84: check time formatting requirements for TC **changed**

87-88: Is the TDI value of 13 for 13 lines? These units are surprising - most other sensors use 8-128 lines of TDI, with 2 n steps **Our wording (number of TDI stage) is in agreement with Materne et al., 2006 (see https://adsabs.harvard.edu/pdf/2006ESASP.621E..66M)**

96: I think you're convolving snow cover and snow depth here with "lowest" - the albedo of fresh snow is high, so saturation can be an issue, but a thin layer of snow cover (a few cm) is not a problem for co-registration. **We added "*if the snow layer is thick off glacier*" to deconvolve both cases.**

104: The image is not really rejected during tasking, right? Some real-time observations or a forecast model is used to prioritize tasking and if a site is cloudy, no image is acquired. *Good point.* **We changed the text to *"Images in the PGO database are almost cloud free because any images acquired with more than 10% of clouds are not validated and is rejected during the tasking continues."***

107: What determines the area of the site? With a 20 km swath, a 100 km 2 image is only 5 km long? I'm guessing these areas refer to some user-defined polygon for the order and delivery of the stereo images. **We changed the text to "*A PGO site, based on user-defined polygons, covers typically 100 to 500 km²...*"**

117: Define GTN-G **defined**

118-119: Consider starting the paragraph with a summary sentence about the number of sites and coverage in each region, then talking about details of site selection **This point was not obvious to us as the previous paragraph end with the number of sites.**

120: Check tense in these sections "can" vs. "could" - some of what is reported is based on past decisions, or summarizing what was done and results. But I think the PGO acquisitions are continuing? If so, can continue using present tense throughout. **Tense adjusted.**

126: suggest putting "repeat mode" into quotes. **Quotes added.**

142-143: suggest "stereo image pair". **Changed**

146: remove accents from "stereo-pairs". **Thanks, the French touch… :)**

147: It would be useful to know more details about how ASP was run (e.g., correlator kernel sizes). This is likely too much detail for this paper, so in addition to the Deschamps-Berger (2020), maybe best to cite a github repo containing the scripts used to generate the PGO DEMs/orthos (for full open science and version control of PGO products moving forward). **The scripts used to generate the DEMs and ortho-images and to coregister them to GLO-30 are available at: https://zenodo.org/uploads/12909586**

I recommend you consider the updated MGM workflow outlined in Bhushan and Shean (https://zenodo.org/records/4554647) and Bhushan et al. (J Glac, 2024, soon to be in press, hopefully)? **We also tested MGM over the Lidar sites and found little difference to SGM. It could be superior in some contexts, even though it is always difficult to prove it unequivocally. Reprocessing the entire archive is a large endeavour and we are not very keen on adding a third DEM product.**

**Below is a table with some preliminary testing (only for three sites in Norway), where we show the NMAD on-glacier and off-glacier. These results suggest very minor statistical differences between SGM and MGM, hence we opted to keep only two versions of the product.**

| | Perct. of data gaps on glac | Median Dh off glac | Median Dh on glac | NMAD off glac | NMAD on glac |
|---|---|---|---|---|---|
| 2018 Langfjord- BM | 0.2% | 0.005 | -0.186 | 0.669 | 0.139 |
| 2018 Langfjord- & - SGM | 0.0% | 0.009 | -0.142 | 0.536 | 0.169 |
| **2018 Langfjord- & - MGM** | 0.0% | 0.013 | -0.138 | 0.530 | 0.178 |
| 2019 Gråsubreen - BM | 2.3% | -0.001 | -0.104 | 0.254 | 0.122 |
| 2019 Gråsubreen - SGM | 2.2% | 0.001 | -0.080 | 0.224 | 0.205 |
| **2019 Gråsubreen - MGM** | 2.2% | 0.000 | -0.098 | 0.219 | 0.219 |
| 2019 Hellstugubreen/Mem BM | 0.8% | -0.021 | -0.124 | 0.450 | 0.119 |
| 2019 Hellstugubreen/Mem SGM | 0.6% | -0.002 | -0.093 | 0.321 | 0.142 |
| **2019 Hellstugubreen/Mem MGM** | 0.6% | 0.001 | -0.094 | 0.303 | 0.153 |

147: The ASP Earth citation guidelines also recommend citing Shean et al (2016), as this is what is used for Pleiades camera models: https://github.com/NeoGeographyToolkit/StereoPipeline?tab=readme-ov-file#citation **Shean et al. (2016) added.**

Suggest citing the Zenodo DOI for the specific release rather than a link to the general github page. Note that there have been several important bug fixes and improvements in more recent ASP versions (>4.0). It would be useful to consider upgrading and documenting software version number in the product metadata, rather than freezing to an older ASP version. **We assume D. Shean meant v3.4.0, the last release of ASP? A test on a DEM showed almost no change so updating to the new version does not seem to be a priority. But this would need to be consolidated on several test sites. We preferred to freeze the version for the sake of homogeneity of the archive.**

150: Should be "Maxar WorldView/GeoEye" recommend including Planet SkySat-C (Bhushan et al., 2021); **Modified and reference added**

151: ASTER is mentioned multiple times earlier in the manuscript without definition of acronym, suggest moving to first location, and specifying VNIR instrument **Done**

153: suggest "images" rather than "satellites" **changed**.

159: "highly-resolved topography" - suggest "enhanced DEM detail/quality and fewer data gaps" **changed**

167: "smoother version" for 20 m DEMs - not necessarily. This is just posted on a coarser grid. **A coarser grid leads to smoothing of the details of the topography. We kept unchanged.**

176: Earlier, the PAN resolution was stated to be 0.7 m. **To avoid this confusion, we added L68: "resampled by the ground segment to 0.5 m and 2 m"**

176: Do you generate orthoimages from both stereo views, or just the more nadir view? The latter seems like a good compromise, and what most users will want. **One of the images is used arbitrarily to generate the ortho-image. Using the nadir most image would have been a good idea but this was not done. We note however that both images of the stereo pair are acquired with symmetric angles so it should have a strong influence.**

Also to clarify, the orthoimages are generated using the original 20-m BM DEM, before any co-registration? This will lead to self-consistent orthoimage, but are you then applying the resulting co-registration translation to the orthoimages? If not, then the ortho and the DEM will not be self-consistent. **Yes, the translation is applied to the ortho-image so they will be self-consistent with the DEM.**

The "right" way to do this is to perform a bundle adjustment, align sparse 3D triangulated matches (or an initial dense stereo DEM) to the reference elevation data, then apply the corresponding transformation to the original camera models and rerun stereo processing to create self-consistent DEMs and orthoimages. Not suggesting you need to go back and reprocess, but something to consider for the future. Shashank Bhushan's latest repos include notebooks and scripts following this workflow, and we are working to wrap all of this within ASP to make it easier/faster for the user. **This is indeed something we should test in the future as bundle adjustment was not performed here.**

177-180: Could cut sentences about pansharpening that you didn't actually do, so not really relevant. **We would like to keep these statements in the text as they may help some of the users, not aware of the tools available freely for pansharpening.**

181: Geolocation performance -> should be clear that this is "The absolute geolocation accuracy" **changed**

185: "are" vs. "were" - just double-check tense throughout paper based on what is done, in

archive, vs. what is ongoing. **Checked throughout.**

185-186: Cite the specific version of demcoreg using Zenodo DOIs: https://zenodo.org/records/7730376. **cited**

Suggest avoiding possessive 's after Nuth and Kaab (2011) **Avoided.**

185: Include citation for GLO-30 and/or COP30 with documentation at first mention, suggest stating the stated global vertical and horizontal accuracy metrics, and noting that accuracy decreases and there are more artifacts/blunders in steep mountain terrain. **Reference and accuracy metrics added.**

191: Suggest considering "3D translation vector" in this section to distinguish from other co-registration approaches that involve rotation. **Agreed.**

197: Maybe consider "product metadata report" or "product quality report" rather than "fact sheet". **Agreed, much better.**

206: Suggest changing "again" to something more descriptive about repeat observation for a given site after 5 years. **We wrote "*Once Pléiades acquisitions are repeated over a site*"**

209: Just say "using the demcoreg package, as described above" **Done.**

212: Rather than "jitter" suggest "unmodeled attitude error ("jitter")" as there is also "jitter" during TDI, which is what many in the electrical engineering community think when they hear "jitter." The cause of the triangulated stereo DEM artifacts is attitude error in the metadata of one or both of the primary image products, introducing horizontal and vertical geolocation error in the resulting triangulated point. **Changed.**

215: It would be best to clearly indicate whether figures are showing DEMs from PHR-1A or PHR-1B throughout the text, so we remember that PHR-1B will have larger attitude error artifacts. **Added to Figure 4, as this is the figure showing these undulation. We do not think it is necessary for other figures.**

220-221: Users should check statistics for residuals after your corrections, and do visual inspection of the difference maps. (I think this is what you are trying to say, but "check the stable terrain" is ambiguous). **Reworded as suggested.**

229: Suggest "entirely cover a single PGO site in a campaign year" - avoid confusion about repeat coverage. **Added.**

233-234: How are these stats provided? They are not in the report provided in Appendix 1. **The sample product will answer such questions. This is a jpg file showing the successive statistics on stable terrain.**

248: Somewhere, should state the geolocation accuracy of GLO-30. **Done L198.**

269: "can be interpreted as residual co-registration error". **"residual" added.**

276: "quality of the PGO DEM co-registration with the reference GLO-30 product". **Changed.**

277: suggest "relative co-registration errors of over 10 m" rather than "residual shifts" as these are relative offsets between your two independently co-registered DEMs. See major comment about masking GLO-30 prior to co-registration - surface relief is only one factor in GLO-30 quality/accuracy. **Changed. See answer to major comment.**

280: suggest "...three independent airborne lidar campaigns acquired data within less than 1 day of a Pleiades stereo acquisition (Table 2)" **Changed as suggested.**

280-281 - Must provide citations for these ALS data (with information on how they were collected and processed), and ideally links to open data archive for the specific products/versions used. I believe these were all ACO products in Canada, not sure about Norway. Details on instrument, altitude, etc will help interpret the observed differences with PGO DEMs over surfaces with variable roughness (e.g., crevasses). While most ALS data should be significantly more accurate/detailed than PGO DEMs, there is a huge range in ALS data quality, often related to instrument specs and processing approach.

**The equipment and processing steps used to produce the ALS data are the same as those described in Pelto et al. (2019), now cited in the paper. Work is in progress to release the entire ALS data obtained over glaciers in western Canada but that manuscript is not yet finished.**

282: suggest being clear that there is negligible elevation change over all surfaces in the scenes, including glaciers, snow and exposed terrain. Can consider moving sentence from lines 285-288 here for more continuity. **Changed.**

282-284: I'm glad you pointed out that ALS data has ~0.1 m uncertainty. Note that the ALS errors can (and often will) be much higher than 0.1 m over steep surfaces. I would delete the "Hence the elevation difference directly reflects the uncertainties of the PGO DEMs" as this is not true. You can just say you neglect the ALS error. There is also some unknown lidar penetration in snow/ice depending on grain size and lidar wavelength (which is why we need citations with more information about the ALS data used - Near-IR vs. green)

**We agree that ALS errors can be larger in steep terrain. We have modified the sentence to 'Hence the elevation difference mainly reflects the uncertainties of the PGO DEMs, although ALS errors can be higher that 0.1 m in steep terrain'**

289: I'm confused here - lidar point clouds are not triangulated. From what I can tell, the authors gridded the lidar point clouds using the ASP point2dem tool with 1 m posting.

**You're correct, this is a slight misunderstanding on our behalf about how point2dem creates the gridded DEM of a sparse pointcloud. After reviewing the documentation we noticed that the routine point2dem used to do a triangulation of a sparse pointcloud in a previous version of the software, but this method was deprecated (now it needs to be specified by using the argument --use-surface-sampling). We've rephrased this sentence in the main document.**

288: Since there is another co-registration with lidar, I'm not sure why this is relevant, suggest deleting. **True. Deleted.**

291: Don't need to cite Shean/NuthKaab again (has nothing to do with lidar). **Deleted.**

291-293: I'm confused here - the authors just stated that the simultaneous products allow one to look at on-glacier and off-glacier surfaces. Why are they masking here? **This is to put ourself in a realistic situation of elevation change mapping where coregistration can only use off glacier terrain (often only from RGI).**

293: I don't understand why this bilinear resampling was needed, and this was not mentioned during earlier co-reg with GLO-30 (much larger difference in grid cell size) - the demcoreg tool you use will do this on the fly (bicubic is default).

**You're correct. This sentence is the leftover of a previous run of statistics, in which we calculated the stats without and with co-registration. In this context the input resolution of the lidar did have more impact. We've deleted this sentence, since as you point out, the demcoreg tool already does this.**

296-298: I think I understand why updated glacier masks were created manually here (likely better stats), but it's a bit odd to use RGI6 for masking during co-reg and then a different mask for evaluation.

**RGIv6 (or now v7) is the standard mask available everywhere so it is used for coregistration. Then, as guessed by D. Shean, we wanted the best mask for clean statistics on/off glaciers. We changed the text.**

279: Suggest "near-contemporaneous" instead of "simultaneous" - they may be same day, but can have time offset of several hours (potentially significant elevation difference during big melt days). **Changed**

317: Suggest "As a result of the co-registration process". **Changed.**

319-320: There are clear residual "jitter" artifacts in this PGO DEM in Canada which likely explains most of the observed NMAD. **Mentioned.**

323-324: I see some "patches" with large apparent elevation difference over crevasses fields and other surfaces where we expect BM to perform poorly compared to SGM. Using NMAD rather than STD here will ignore these "outliers" but depending on application, "superior" is subjective (e.g., if you're studying crevasse dimensions or small-scale roughness, SGM will always be superior). **Superior deleted.**

327-328: While ALS does obtain returns from beneath the canopy, if you are not isolating first/only or ground returns in the ALS point clouds, then ASP is going to produce some weighted average of all available ALS point elevations in each grid cell. You will not end up with a DTM (and I'm not sure that a citation to Piermattei should be included here).

My recommendation would be to mask the vegetation, as this affects your stats, and you need to create a DSM, and need to deal with height percentiles (e.g., 90th percentile) when gridding lidar points for comparison with stereo products. This is tricky, and an important gap that requires more attention from our community in my opinion - ALS is great, but it needs to be carefully processed when used as "truth" for stereo evaluations.

**We think it is actually important to show the influence that vegetation can have on two different types of measurements (stereo vs. Lidar). It is maybe obvious for specialists of Lidar but probably not to all users of the PGO database. Even near-contemporaneous acquisitions can exhibit spatially structured biases due to different interactions with the surface.**

330-332: And to exclude these areas during co-registration. Demcoreg by default only uses slopes within range of 0.1-40°. **Coregistration added.**

332: "relief" not "reliefs" **corrected.**

338-339: I'm not sure you can claim "decimetric accuracy" - I interpret that to mean the accuracy is ~0.1 m. The bias is 0.0-0.2 m, and NMAD is 0.3-0.7 m, so combined is potentially closer to ~1 meter. I think "sub-meter" might be more appropriate. **Sub-meter used in the revised manuscript.**

341: "likely facilitate" should be "is required" or "improve" - there's no question that stable terrain is requirement for success. **Changed.**

342: Really, it is a matter of the glacier vs. stable terrain area and distribution. If you have a 20x40 km PGO DEM, you can obtain good results for larger glaciers and ice caps. If you're ordering smaller ~100 km 2 images, max glacier size decreases. **Agreed.**

343: The earlier section 3.1 is also attempting to estimate "uncertainty of the elevation changes". The first paragraph in this section seems to be using point measurements to do the same thing (validation) as with the lidar analysis? I think this section is supposed to focus on PGO_DEM minus

PGO_DEM error, rather than comparing against an external reference? Consider revisiting section heads and organization for how the different evaluations are presented, maybe moving the GNSS paragraph somewhere else (perhaps before the lidar section, since that is also PGO DEM minus external ALS_DEM). **In section 3.1, we evaluated a single PGO DEMs against an external reference. Here we now evaluate the elevation difference derived from two PGO DEMs (against GNSS or on stable terrain). For this reason we think it is more appropriate to keep the current order.**

345: suggest "differential GNSS measurements with centimeter accuracy" rather than "centimetric GNSS measurement" **changed.**

It's not clear to me why the GNSS measurements can't be used to evaluate the DEM accuracy directly, assuming they are acquired on the same day. **Simultaneity is important as melt can lead to significant elevation change over just a few days.**

353: Should explicitly state the assumption elevation difference should be 0 over "stable" terrain, and any observed residual is considered error. **Assumption clearly stated now.**

375: "peculiar" word choice. "Anomalous" may be more appropriate. **Changed as suggested.**

375-377: Fine to throw these out if there is a clear threshold or other justification - ties in with earlier comment about thresholds to determine whether co-registration failed. **Agreed.**

382: ASTER VNIR. **Corrected**

387: If we had two PHR satellites dedicated to glacier observations, or a constellation of shared PHR satellites, it might. But the limited number of available PHR (two) and competition prevents this. In the coming decade, we should have several constellations of VHR imaging satellites capable of stereo. **Changed to indicate that this is because the VHR imaging satellites are not purely dedicated to science.**

389: suggest "reasonable assessment of global glacier mass change" **Changed as suggested.**

390: include year for Hugonnet citation, and fix other instances in this section. **Changed.**

398: "compared these regionally aggregated PGO values with corresponding values using the full sample from Hugonnet et al. (2021)" **Changed.**

399: Given the 2016-2017 start date for PGO, comparing with 2000-2019 values is hard to justify, even if the Hugonnet uncertainty is lowest for the longer time period. **We think the revised text will be clearer now to convey that we only compare two samples (Global vs. PGO-like) of glaciers whose mass balances were only extracted from the Hugonnet et al. (2021) dataset.**

404: "slightly more negative" is used to describe 18% more negative. I'm not sure I would call that "slightly." **Slightly deleted.**

404-405: I am not clear on how 2000-2009 mb is measured using PGO DEMs that only begin in 2016.

OK, I now realize what was done here. See major comment about improving language to remove confusion that you are comparing mb from PGO DEMs with mb from Hugonnet et al (2021).

**We think it will be clearer now to the referees and readers that we only compare two samples (Global vs. PGO-like) of glaciers whose mass balances were only extracted from the Hugonnet et al. (2021) dataset.**

418: "perform even better" - this could be misinterpreted as comparison with external studies. Suggest "and capture temporal evolution"425: "...more accurate results for mean glacier mass balance" - not to be confused with earlier PGO DEM accuracy evaluation using lidar. **We changed to *"this sample is able to perform even better at capturing their temporal changes".***

454: I think its important to be clear that you are evaluating calibrated/corrected PGO DEM products, not the original DEMs prepared using original Level-1B metadata. **Clarified**

456: I know what you mean by "stable terrain as a proxy of the uncertainty on glaciers" but as written, this doesn't make sense - be clear that the "residual elevation difference values on nearby stable terrain was used to estimate corresponding uncertainty on glacier surfaces" **changed**

457: "the mean glacier-wide elevation difference has…" - be clear that this is area average, not just "elevation differences" **changed**

458-460: not sure this belongs in conclusions, as it is not discussed at all in the paper, and the focus is not on glacier mass balance. If anything, suggest moving to "discussion" portion of section 4. **We agree that this did not really belong to the manuscript. Deleted.**

Figures

Suggest redefining all acronyms in figure captions, as readers skimming figures won't necessarily know that BM = Block Matching. **Done.**

Figure 1: It might be useful/informative to use a color ramp for marker color to signify the number of glaciers covered at each site in the regional panels (earlier it was mentioned that most sites cover dozens of glaciers). **We think this suggestion would clutter the figure and so left it unchanged.**

Figure 2: It would be useful to show at least one of the input orthoimages here, so we know why there are data gaps - is there a cloud, or is it fresh, textureless snow? Note for future processing, some of the latest recommended ASP MGM options reduce the cross-hatch artifacts shown in 2b. **We computed the histograms and found image saturation where the gaps are concentrated. This is now mentioned in the text and the caption. Thanks for the advices about future processing using MGM.**

Figure 3: Not a requirement, but it would be nice to see metrics on errors before and after if you are going to report the translation components in a figure caption. **We added in the caption** *"Coregistration reduced the normalized median absolute deviation (NMAD) off glacier from 0.81 m to 0.48 m."*

Figure 4: Suggest rewording caption, as you're not really showing processing steps, rather showing an example of difference map products before and after two corrections. Maybe add titles over each column (something like "Original", "After Co-registration", "After Bias Correction") to make it easier to interpret. **All suggestions taken into account.**

Figure 5: Suggest "The figure shows translation components for the BM DEMs, as the mean and standard deviation for the SGM DEMs were nearly identical". State whether this was the 2 m or 20 m DEMs. Is the cm unit correct here: "a few tenth of centimeters"? This is "a few mm", no?

**All suggestions taken into account. Unit is not provided anymore but "a few tenth of centimeters" was right.**

Figure 6: See comments about whether figures show 2 or 20 m DEMs. **Here it seems clear that the figure shows the 2 m block-matching BM DEMs**

Figure 7: Not clear which is BM and which is SGM - add titles or modify caption. I think a is BM and b is SGM? **Title added and caption modified.**

I'm a little surprised by the observed differences between c and d if the only difference is the input PGO DEM using BM (c) vs. SGM (d). This potentially indicates that the independent co-registration approach (co-registering BM to lidar and independently co-registering SGM to lidar) is introducing bias that could be incorrectly interpreted as due to the correlation approach (or terrain properties). **We observe many local artefacts (Figure 7). However the bias (see column "Median Dh on glac**

**(m)" in Table 3) differ by only 0.05 m between the two correlation approaches so we do not really the point of the referee.**

It would be useful to have one sentence summarizing the "takeaway" message for this figure. Observed elevation differences are near 0, but there are also some artifacts and differences between BM vs SGM products. **Take away message added in the text.**

Figure 8: Since presenting signed difference values, be clear about which DEM was subtracted from the other. "Difference between" is ambiguous. **"*(PGO DEM minus Lidar DEM)*" added.**

Suggest "Points show median and shaded area shows NMAD of dh values within each X m elevation bin (left) and each X degree slope bin (right)". **Adopted**

It's not completely clear why we would expect a relationship between dh and elevation, unless there are systematic scaling issues in the PGO DEMs. Slope is much more important, and slope and elevation are not independent (i.e., generally, higher slope values at higher elevations in these mountains). **We agree but we also see in the literature that many colleagues are making unjustified corrections of the biases with altitude. So we think it is important to illustrate that the bias with altitude is small.**

Figure 10: Don't need 's for Hugonnet et al (2021) dataset. **Deleted**.

Tables

Table 1: Suggest moving number of stereo pairs after number of sites. In a single campaign, are there any sites with multiple stereo pairs?

**On average two stereo pairs are needed to cover a site but this is highly variable. We now added an appendix Table A1 listing all PGO sites and the number of stereo pairs needed to cover each site.**

Does the Glacier area column refer to orthoimage coverage or the DEM valid data coverage? The latter should be less due to data gaps from failed correlation.

**We clarify this point by adding: "*The columns "total and glacier areas" correspond to the full coverage. The real area coverage by PGO is in fact slightly lower due to data gaps in the DEMs.*"**

Table 2: Seems odd to just report the glacier area, as you report stats for on and off glacier. Suggest including the "total area" of intersection used for co-reg/evaluation, and then glacier area.

***Total area of intersection used for co-registration added to the table 2.***

Suggest include date and time for each collection (see earlier comment about potential melt between stereo and lidar acquisition time offset). **Dates are all listed in the Table. The exact time of the lidar surveys were not available, but a few hours of time lag (hence a few centimeters of melt) should not make a significant difference given the other sources of errors.**

Suggest "Lidar density (pt/m 2 )" **Done.**

It's a bit odd to use a non-PGO DEM for this evaluation (which is not reproducible). At least state that the processing used for the non-PGO DEM was identical to PGO DEMs.

**The case of near-contemporaneous acquisitions are rare. Hence, we prefer to keep this additional evaluation site in Norway even if the Pléiades images are not in the PGO database. We added that the processing of this stereo pair was identical.**

Table 4: "balance" singular, not plural. The PGO glaciers (this study) is a bit misleading, as it makes it sounds like you measured mass balance using PGO DEMs. As requested in earlier comments, make sure this is clear. **Clarified now.**

Data Availability

468: Suggest "Licensing issues prevent open distribution of primary products and orthoimages. **Changed**

The PGO images are available… **Changed**

Have you considered distributing the entire archive through AWS Open Data Registry (like ArcticDEM/REMA) or other on-demand, direct-access cloud provider, rather than the current combination of university portals? The ability to subset and stream from COGs in the cloud is much more efficient than downloading entire products (hence the cloud-optimized GeoTiff 🙂 ) **We have limited institutional flexibility on the choice of the distribution platform. It is true that when we see how convenient it is to obtain data from OpenTopography this is something we will consider for the future... Thanks for the suggestion.**

As requested in a few places, for open science and reproducibility, it would be best to distribute all processing and analysis scripts in a version-controlled public archive, so PGO users can understand how the products were created, and how the processing has evolved (and whether they should download a new version of their DEM).

**A zenodo repository with the processing scripts was created and is referred to in the Data availability section. https://zenodo.org/uploads/12909586**

Appendix 1

Great to see this pdf bundled with each product. Consider including a context map for the site here in addition to the browse image. Our team (led by Ben Purinton) is working to centralize scripts (https://github.com/uw-cryo/asp_plot) to generate standardized "quality report" pdfs for ASP output. It would be great to work together to develop a community package and integrate within core ASP tools.

**Something we will consider in case of a new release. The PDF was generated during the processing itself, adding the context map would require us to reprocess the entire database.**

**Finally we would like to deeply thank David Shean for the time taken in a busy schedule to perform this exceptionally detailed review of our work. This has led to many improvements in the manuscript, will shape future work and already stimulated an ongoing analysis by the image quality team of the French Space Agency.**

---

## Referee Report (RR1)

**Second Review of "The Pléiades Glacier Observatory: high resolution digital elevation models and ortho-imagery to monitor glacier change" by Berthier et al.**

https://egusphere.copernicus.org/preprints/2024/egusphere-2024-250/

David Shean

August 11, 2024

Many thanks to the authors for considering all of my detailed comments, and providing a thorough response document. I know how much time is involved to do this. The revised manuscript reads well, and I believe is ready for publication.

There are a few items below which I encourage the authors to consider in their final revision/proofs, as well as ongoing product distribution, and future efforts to update PGO products and use for analysis. I am recommending acceptance, and I don't require a detailed response to this second round of comments, but I know the PGO team will understand these points, and I am happy to follow up on any of this, provide further explanation, and discuss corrections outside of the review process. Hoping we can get these things right, to establish best practices and deliver the best possible products for the community.

**Product availability**

I was able to log in and download sample products!  I noted that transfer rates to Seattle, WA were ~300-500 KB/s.  The file I chose was 2023-11-11_0503585_Mera_ASE.tgz is 579 MB, so this required 15-30 minutes to transfer - not ideal, but fine.

After decompressing the archive, I noticed that the tif files were not tiled or compressed.  I ran my gdal_compress.sh utility (https://github.com/dshean/gdal_tools/blob/master/gdal_compress.sh) using lossless LZW compression for the 4 tif files, which significantly reduced file size:
- 2023-11-11_0503585_Mera_ASE_1B_DEM_BM_2m.tif from 673M to 203M
- 2023-11-11_0503585_Mera_ASE_1B_DEM_BM_20m.tif from 6.7M to 3.0M

I created a new tar.gz archive with these files, the total archive size was reduced from 579 MB to 387 MB. I recommend that the PGO team runs similar gdal_translate commands to tile and compress the tif files, creates new archives, and replaces the existing archives on the distribution site before official release. This will reduce archive storage and file transfer times for the community.

**Response to Reviewer #1:**

L246-L248: I agree with this point and it also raises the question why the authors did not coregister to a more precise dataset such as ICESat/ICESat-2 altimetry? Is there a particular reason? I could imagine track spacing and surface slope are an issue?

**The reviewer is right that summer laser altimetry data would be ideal, at least to correct a final vertical shift. However, especially at low latitudes, the track spacing is such that few to no laser elevation measurements would intersect with the Pléiades footprint (max 20 km wide but often much less given the shape of our sites).**

The author's response is likely incorrect. While I don't think integrating altimetry is necessary for this paper, I encourage the authors to explore the available coverage from the 5+ year ICESat-2 archive. At lower latitudes, systematic ICESat-2 off-pointing fills gaps between reference ground tracks, and even with missing data due to clouds, there should be sufficient coverage (1000s of ATL06 points) for a typical PGO DEM footprint. Try running SlideRule ATL06-SR query for one of your low-latitude footprints.

**Follow up on track-changes and author responses**

Line numbers from revised manuscript with track changes, and screenshots from response document

56 - Minor point - these are not "raw," they are Level-1B or "Primary" products, with many corrections by Airbus before delivery

70 - suggest "stereo images can be acquired in an along-track pair"

226 - delete "the demcoreg package, as described above"

228-230 and responses doc, regarding the 5th order polynomial fit to cross-track residuals

While I think the current approach is fine for the manuscript and initial release of the products, I maintain that this 5th order polynomial cross-track correction introduces unnecessary bias, and encourage the authors to carefully reconsider their methods for future corrections. I say this because I know that many in the community will use the PGO dh maps to compute geoedetic mass balance without careful inspection.

210-221: I don't understand why a 5th-order polynomial is needed for the cross-track direction, and there is no justification provided here. I just don't see these artifacts in Figure 4, and I worry about over-fitting and introducing bias over adjacent glaciers from a limited sample. If there is a root cause related to the CCD geometry model in the Pleaides-HR camera model, or systematic "cross-track jitter" during collection of stereo pairs, then that provides justification for high-order polynomial correction, but I'm not sure that's the case.

**We confirm that across track biases are generally smaller than along track biases. Below, we show several examples where the spatially-structured biases should, we think, be corrected. A fifth-order polynomial performed suitably well to remove bias as used in Gardelle et al. (2013, http://dx.doi.org/10.5194/tc-7-1263-2013). We note that the corrections are all graphically (.jpg or .png) illustrated in the data products downloaded by the users. In case those generic corrections are not satisfying (and this is indeed sometimes the case), the user can easily come back to the uncorrected DEM and apply the relevant correction.**

[Figure]

**Figure R1. Three examples where a 5th order polynomial fit to the residuals off glacier was applied to reduce the uncertainties in the PGO DEM.**
**The examples are for product Gulkana_ALA_2016-09-01_2128501_2021-09-09_2122269_SGM_2m (upper left), Lombardy_CEU_2017-08-01_1032561_2022-08-21_1033056_SGM_2m (upper right) and Olsen_GRL_2017-07-17_1358160_2022-07-22_1413096_SGM_2m (lower left).**

Based on the text, it sounds like this 5th order polynomial cross-track correction is performed first, then the spline along-track correction is performed second. **The order of these corrections is important, so if this is not the case, please clearly state the order in the text.** The order stated in the PGO paper (cross-track then along-track) is different than Gardelle et al (2013) "Then, we compute the elevation differences along and across the satellite track on stable areas and when necessary correct the bias using a 5th order polynomial fit."

The PGO sample I downloaded (Makalu_ASE_2018-10-16_0500119_2023-10-23_0500156) illustrates the problem…

[Figure]

The right plot shows the non-uniform, anisotropic distribution of "stable" surfaces, and the clear residual along-track jitter artifacts over these surfaces. By first computing a cross-track correction, the non-uniform spatial distribution of the static control surfaces leads to preferential sampling of the jitter bias (which dominates the residuals), so the area to the west appears to have slightly positive bias, and the area to the east appears to have negative bias. Fitting the cross-track correction to these residuals, before the along-track jitter correction, biases some columns by over 0.3 m in this case, which will propagate to all glacier surfaces in those columns.

[Figure]

The along-track jitter correction should be performed first, as this captures most of the observed residual variability.  I maintain that lower-order polynomials (planar, at most 2nd order) are sufficient to capture expected cross-track artifacts for linescan stereo sensors like Pleiades (assuming the detector array geometry is good, which it is for PHR1A and 1B).

Following along-track correction, the Gulkana residuals (below, mentioned in the response doc) could be modeled with a linear correction.

[Figure]

The Olsen residuals (also cited in the response doc) suffer from the same issues as Makalu above, with limited stable terrain, and non-uniform sampling of the jitter artifacts leading to apparent cross-track artifacts. The current PGO correction is unnecessarily introducing +/-0.2 to 0.3 m bias over adjacent glaciers.

[Figure]

Again, this is not a "must do" for this paper, but please consider this carefully for future processing efforts and future PGO product versions.

264-267 - Thanks for the detailed response regarding the systematic geolocation bias with latitude. Interesting to see the plots in the response document. Hopefully the source of this issue becomes clear. Personally, I would exercise caution during interpretations for different sensors, as there is considerable spread in the translations at all latitudes.

I appreciate the tests with different reference DEMs.  Just to confirm, this was all done by first transforming the reference DEM to match the local UTM zone of the PGO DEM?

While it shouldn't be an issue within a UTM zone, I'm still wondering about projection distortion issue with latitude. Distance, direction, and pixel area are all important properties for Nuth and Kaab, and these will be distorted by different amounts depending on projection choice and latitude. One way to test this is to repeat the co-registration using a local planar projection (like orthographic, or stereographic with clat,clon defined by center of DEM).

267-272 and responses doc, regarding the 2.4 m vertical bias and refraction/aberration correction

While some of the systematic 2.4 m vertical error could be due issues around penetration through snow in the GLO-30 products, I'm suspicious that this could be systematic error in the ASP v3.0 output (note similar vertical bias magnitude for WV/GE documented in Shean et al., 2016 and Fig 4 in Shean et al (2019) PIG paper in TC). This could likely be related to the (implementation and/or lack of) atmospheric refraction and velocity aberration correction for the version of ASP you are running, and/or some of the 2.0 m uncertainty associated with definition of the WGS84 Ensemble (EPSG:4326) vs a specific ITRF/WGS84 realization (e.g., https://spatialreference.org/ref/epsg/7664/ - see WKT2). Note that the latest ASP includes new, validated refraction/aberration corrections using the USGS Community Sensor Model, and support for PROJ9 and unambiguous output product 3D CRS definitions using WKT2.

**We computed one of the DEMs using the last version of ASP at the time of writing (v3.4) and found almost no changes (within 1-2 cm in height). We computed the mean vertical error for Pléiades 1A and 1B and found it was more negative (-3.9 m) for 1B than for 1A (-1.1 m). Hence, a**

**sensor specific correction may be partly responsible for the systematic vertical error and this is also under investigation at CNES. See revised section 3.1.1.**

Glad you tried this, but unfortunately, the refraction and aberration corrections were not enabled by default for Pleiades.  See release notes:
https://stereopipeline.readthedocs.io/en/latest/news.html#release-3-4-0-june-19-2024

WorldView (DigitalGlobe) cameras (Section 5):

- The WorldView linescan model got moved to a CSM implementation. The transitional option `--dg-use-csm` was removed. The new implementation is about 5x faster for ground-to-image projections.
- Re-enabled correcting velocity aberration and atmospheric refraction. These corrections are now implemented in the CSM camera model, and, unlike before, play nicely with bundle adjustment (Section 5.9).
- The options `--enable-correct-velocity-aberration` and `--enable-correct-atmospheric-refraction` got removed.

[Figure]

- Non-DG cameras do not use these corrections, as a case for that has not been made.

I reviewed appendix C in the Pleiades user guide (https://content.satimagingcorp.com/media/pdf/User_Guide_Pleiades.pdf) and it appears that we need to enable these corrections for the rigorous model in the Pleiades primary products (and perhaps also the RPCs). "The value of the atmospheric refraction correction depends on the incidence (null when viewing at nadir, **approximately 2m when incidence equals 15 degrees**)." - the magnitude is consistent with your observed vertical bias, and the sensor source (1A vs. 1B) may be less relevant than the relative geometry of each pair.

I raised this issue with ASP devs, and we will revisit. The PGO team can also repeat the ASP v3.4.0 test the CSM linescan model implementation for Pleiades with these corrections enabled.

319: lidar data source

> 280-281 - Must provide citations for these ALS data (with information on how they were collected and processed), and ideally links to open data archive for the specific products/versions used. I believe these were all ACO products in Canada, not sure about Norway. Details on instrument, altitude, etc will help interpret the observed differences with PGO DEMs over surfaces with variable roughness (e.g., crevasses). While most ALS data should be significantly more accurate/detailed than PGO DEMs, there is a huge range in ALS data quality, often related to instrument specs and processing approach.
>
> **The equipment and processing steps used to produce the ALS data are the same as those described in Pelto et al. (2019), now cited in the paper. Work is in progress to release the entire ALS data obtained over glaciers in western Canada but that manuscript is not yet finished.**

Thanks for including Pelto reference for the BC lidar surveys. It is disappointing to see that **no corresponding information was included about the Norway lidar collections in the text or the responses**. According to the author contributions section, presumably co-authors JMCB and LMA provided the airborne lidar data for Norway and can track this down? Hopefully someone can include citations or some text with information about those surveys, even if the lidar data are not publicly available. As a community, we need to be more careful assuming "lidar is truth" and "all airborne lidar data have the same accuracy."

324-327 - regarding lidar data preparation and vegetation masking

> 327-328: While ALS does obtain returns from beneath the canopy, if you are not isolating first/only or ground returns in the ALS point clouds, then ASP is going to produce some weighted average of all available ALS point elevations in each grid cell. You will not end up with a DTM (and I'm not sure that a citation to Piermattei should be included here).
>
> My recommendation would be to mask the vegetation, as this affects your stats, and you need to create a DSM, and need to deal with height percentiles (e.g., 90th percentile) when gridding lidar points for comparison with stereo products. This is tricky, and an important gap that requires more attention from our community in my opinion - ALS is great, but it needs to be carefully processed when used as "truth" for stereo evaluations.
>
> **We think it is actually important to show the influence that vegetation can have on two different types of measurements (stereo vs. Lidar). It is maybe obvious for specialists of Lidar but probably not to all users of the PGO database. Even near-contemporaneous acquisitions can exhibit spatially structured biases due to different interactions with the surface.**

I think the authors missed my point about the vegetation. The lidar point clouds should be filtered to preserve first returns to produce a DSM, or vegetation should be masked before co-registration.

I agree that showing and analyzing offsets over vegetation is valuable, but the current approach will potentially lead to incorrect co-registration and bias output DEMs.

When creating a DEM using ASP's point DEM, you are using points from all lidar returns (top of canopy, canopy, understory, ground). This will lead to systematic bias in the gridded values over vegetated areas - the final point2dem weighted average elevation for the output grid cell is not the top of canopy and not the ground, but somewhere in between (excatly where depends on point density).

The example in Figure 7 A/B shows vegetation bias in blue with relatively small areal coverage (maybe 5%), so it shouldn't impact the robust co-registration approaches used (automatically eliminate outliers). But using the same strategy will have a much bigger impact for other sites with greater vegetation cover (>20-30%).

I'm not suggesting reprocessing everything, but please be careful of this in the future - either isolate first returns from the lidar point cloud to create a DSM, or mask vegetation during co-registration.  You can then perform the same analysis using the unmasked products, which will show the important dh difference over vegetation.  Hopefully that makes sense.

Fig 7 - NMAD for BM and SGM compared to lidar for Hellstugubreen

[Figure]

I'm a little surprised by the observed differences between c and d if the only difference is the input PGO DEM using BM (c) vs. SGM (d). This potentially indicates that the independent co-registration approach (co-registering BM to lidar and independently co-registering SGM to lidar) is introducing bias that could be incorrectly interpreted as due to the correlation approach (or terrain properties). **We observe many local artefacts (Figure 7). However the bias (see column "Median Dh on glac**

16

**(m)" in Table 3) differ by only 0.05 m between the two correlation approaches so we do not really the point of the referee.**

I think the authors misunderstood my earlier comment. It appears that there is larger residual horizontal geolocation error in the BM product compared to the SGM product, likely introduced during the independent co-registration process for each. Please check the translation vectors for the dh maps used to produce 7c and 7d - I can't do this because the products involving lidar data are not available (which is fine, but limits reproducibility).

I'm having a hard time understanding how the on-glacier NMAD values in Table 3 for the Hellstugubreen **BM** DEM (0.12 m, Fig 7c) is lower than the on-glacier NMAD for the

Hellstugubreen **SGM** DEM (0.15 m, Fig 7d). Just looking at these difference maps (with identical color ramp), there appears to be more spread in the BM dh product than the SGM dh product (reproduced above for clarity). I'm wondering if there was potentially a mistake in the analysis/plotting code or labeling in the figure? Maybe double-check, and if necessary, modify the statement on linked 364-366:

364 the uncertainty on glaciers is a conservative approach. Interestingly the choice of the correlation
    algorithm (BM or SGM) has a different influence on and off glaciers. SGM results in lower NMAD is
366 superior off glaciers whereas using BM leads to reduced NMAD on glaciers.

492 (Fig 10 caption) - "mass balance estimates" instead of "mass balances"
469 - same

---

## Author Response (AR2)

Etienne Berthier
14 av Ed Belin
31400 Toulouse
etienne.berthier@univ-tlse3.fr

September 2024

Dear Editor,

Please find enclosed a revised version of our manuscript (MS) entitled, "The Pléiades Glacier Observatory: high resolution digital elevation models and ortho-imagery to monitor glacier change". To facilitate your assessment, we uploaded a clean and track-change version of the revised MS.

We thank David Shean for its further technical comments on our manuscript. Below, you will find a copy of all comments and, in blue, a point-by-point response to them. When relevant, the revised text is provided in italics.

We hope that these corrections/clarifications make our paper suitable for publication in *The Cryosphere*.

Yours sincerely,

Etienne Berthier on behalf of the co-authors

**Reply to David Shean**

Many thanks to the authors for considering all of my detailed comments, and providing a thorough response document. I know how much time is involved to do this. The revised manuscript reads well, and I believe is ready for publication.

There are a few items below which I encourage the authors to consider in their final revision/proofs, as well as ongoing product distribution, and future efforts to update PGO products and use for analysis. I am recommending acceptance, and I don't require a detailed response to this second round of comments, but I know the PGO team will understand these points, and I am happy to follow up on any of this, provide further explanation, and discuss corrections outside of the review process. Hoping we can get these things right, to establish best practices and deliver the best possible products for the community.

**We thank Dr. Shean for his continued help to improve our paper. Despite his statement that "I don't require a detailed response" we, nevertheless, think a response is still useful for the editor (and later all readers), in particular to discuss why we did not incorporate some suggestions provided by Dr. Shean in the current paper (these suggestions will be included in future releases of the PGO).**

Product availability

I was able to log in and download sample products! I noted that transfer rates to Seattle, WA were ~300-500 KB/s. The file I chose was 2023-11-11_0503585_Mera_ASE.tgz is 579 MB, so this required 15-30 minutes to transfer - not ideal, but fine.

After decompressing the archive, I noticed that the tif files were not tiled or compressed. I ran my gdal_compress.sh utility (https://github.com/dshean/gdal_tools/blob/master/gdal_compress.sh) using lossless LZW compression for the 4 tif files, which significantly reduced file size:

● 2023-11-11_0503585_Mera_ASE_1B_DEM_BM_2m.tif from 673M to 203M

● 2023-11-11_0503585_Mera_ASE_1B_DEM_BM_20m.tif from 6.7M to 3.0M

I created a new tar.gz archive with these files, the total archive size was reduced from 579 MB to 387 MB. I recommend that the PGO team runs similar gdal_translate commands to tile and compress the tif files, creates new archives, and replaces the existing archives on the distribution site before official release. This will reduce archive storage and file transfer times for the community.

**We are relieved to read that the reviewer could access the data and we have informed the team managing the distribution platform (A2S) about the transfer rate. They answered that the server itself has a much higher transfer rate than indicated by D. Shean and they suggested that the slow rates of transfer could rather come from the end user. We downloaded the same file as D. Shean (2023-11-11_0503585_Mera_ASE.tgz) from home in 15 seconds.**

**Regarding file compression. We made the same test as the reviewer on a PGO DEM and ended with contrasted results:**
○       **uncompressed tif 562 MB,**
○       **compressed using D. Shean suggestions (gdal_translate -co COMPRESS=LZW) = 342MB**
○       **compressed as done in the PGO (tar -czvf) = 317 MB**

**Still it seems wise to compress the tif files as suggested by the reviewer and this is something we will implement for forthcoming products.**

Bias correction using ICESat-2 data : "The author's response is likely incorrect. While I don't think integrating altimetry is necessary for this paper, I encourage the authors to explore the available coverage from the 5+ year ICESat-2 archive. At lower latitudes, systematic ICESat-2 off-pointing fills gaps between reference ground tracks, and even with missing data due to clouds, there should be sufficient coverage (1000s of ATL06 points) for a typical PGO DEM footprint. Try running SlideRule ATL06-SR query for one of your low-latitude footprints.

**The use of ICESat-2 data will require some further analysis. We would like to point out that for some of the PGO sites (for example San Quintin Glacier or Grey Glacier in Patagonia), the fraction of OFF glacier terrain in some of the Pléiades data is low. Hence, the number of actual laser footprint is extremely low. The coregistration could be tested ON glaciers using near-contemporaneous ICESat-2 data (to increase the number of ICESat footprints), but this will be appropriate if the time difference between Pléiades and ICESat-2 is only a few days; otherwise the signal of elevation changes will be too strong.**

56 - Minor point - these are not "raw," they are Level-1B or "Primary" products, with many corrections by Airbus before delivery

**We concur. "raw" was incorrect and we deleted the term as the level of production "primary" is already mentioned L148.**

70 - suggest "stereo images can be acquired in an along-track pair". **Changed as suggested.**

226 - delete "the demcoreg package, as described above". **Deleted.**

228-230 and responses doc, regarding the 5th order polynomial fit to cross-track residuals.

**The reviewer raise many good points here**

**We checked the Gardelle et al., 2013 IDL scripts and can confirm that they followed the same order as us, first the cross track correction and then the along-track one. Unfortunately, the order is not properly reflected in their description in the article.**

**We followed the same order (now clarified in the revised article) but did not perform any analysis to further justify the order. This analysis is clearly something to explore in the future. The Makalu example proposed by D. Shean would be a good case study to start with. We anticipate only a minor influence overall.**

**We added (L222ff) "*We note that the order of these corrections (first across-track then along-track) were taken from Gardelle et al. (2013) but were not studied further and could be the topic of future analysis.*"**

**For the Gulkana case study, we note that a linear correction would have been relevant but the 5th order correction did not introduce any unexpected bias.**

**However, we agree with the reviewer (and observe) that in some cases (mostly when stable terrain is rare or badly distributed), this high order fits do not work well. Hence we included a note on that in the revised text (L224ff): "*We also emphasise that the quality of these bias corrections depends on the availability of sufficient and well-distributed stable terrain. We therefore strongly encourage users to check the relevance of these automatic corrections using the plots associated with each elevation difference map and, if necessary, generate themselves the elevation change map using the PGO DEMs.*"**

264-267 - I appreciate the tests with different reference DEMs. Just to confirm, this was all done by first transforming the reference DEM to match the local UTM zone of the PGO DEM?

**Yes, the reference DEM is reprojected using gdal_warp to match the projection (same UTM zone) and extent of the PGO DEM.**

While it shouldn't be an issue within a UTM zone, I'm still wondering about projection distortion issue with latitude. Distance, direction, and pixel area are all important properties for Nuth and Kaab, and these will be distorted by different amounts depending on projection choice and latitude. One way to test this is to repeat the co-registration using a local planar projection (like orthographic, or stereographic with clat,clon defined by center of DEM).

**We tested the potential impact of such a latitudinal distortion in Iceland (>60°N). We co-registered nine Pléiades DEMs generated in the local projection system (ISN93, EPSG:3057) to a lidar DEM, and next tested re-projecting the same DEMs to UTM (UTM27N) and then perform the coregistration to the lidar. The difference of the shift vectors is at most less than 1 m (max 0.7 m in easting, 0.9 m in northing, 0.3 m in Z). The average difference is close to 0 (0.2 m in easting, 0.2 m in northing, 0.0 m in Z). This is an order of magnitude smaller than the latitude shifts discussed in our previous response letter so this appears not to be the cause of the northward shift at high latitudes.**

267-272 and responses doc, regarding the 2.4 m vertical bias and refraction/aberration correction

**We thank D. Shean for digging so deeply in the release of ASP. We plan to test the CSM linescan model implementation for Pleiades in the future. In our present work, however, we decided not to implement this test because, thanks to the systematic coregistration to Cop30, the PGO products are not affected by the bias.**

319: lidar data source. Thanks for including Pelto reference for the BC lidar surveys. It is disappointing to see that no corresponding information was included about the Norway lidar collections in the text or the responses. According to the author contributions section, presumably co-authors JMCB and LMA provided the airborne lidar data for Norway and can track this down? Hopefully someone can include citations or some text with information about those surveys, even if the lidar data are not publicly available. As a community, we need to be more careful assuming "lidar is truth" and "all airborne lidar data have the same accuracy."

**Good suggestions. The lidar surveys were made by TerraTec AS (2018, 2019a, 2019b). We have now added references to these reports. Unfortunately, they are only in Norwegian. NVE retrieves the data in UTM projections in the local zone (32 or 34 N).**

324-327 - regarding lidar data preparation and vegetation masking.

I think the authors missed my point about the vegetation. The lidar point clouds should be filtered to preserve first returns to produce a DSM, or vegetation should be masked before co-registration.

I agree that showing and analyzing offsets over vegetation is valuable, but the current approach will potentially lead to incorrect co-registration and bias output DEMs.

When creating a DEM using ASP's point DEM, you are using points from all lidar returns (top of canopy, canopy, understory, ground). This will lead to systematic bias in the gridded values over vegetated areas - the final point2dem weighted average elevation for the output grid cell is not the top of canopy and not the ground, but somewhere in between (excatly where depends on point density).

The example in Figure 7 A/B shows vegetation bias in blue with relatively small areal coverage (maybe 5%), so it shouldn't impact the robust co-registration approaches used (automatically

eliminate outliers). But using the same strategy will have a much bigger impact for other sites with greater vegetation cover (>20-30%).

I'm not suggesting reprocessing everything, but please be careful of this in the future - either isolate first returns from the lidar point cloud to create a DSM, or mask vegetation during co-registration. You can then perform the same analysis using the unmasked products, which will show the important dh difference over vegetation. Hopefully that makes sense.

**We take good note to be careful about this aspect in the future. The Canadian team (from ACO) usually generates the DEMs only from the ground returns. Because Pléiades DEMs typically map the summit of the canopy (or an horizon between the canopy and the ground), we processed the lidar pointcloud differently here. Figure 7 shows that Lidar and submeter stereo "see" the vegetated terrain very differently (we note that these differences will vary between seasons). We agree that a global vegetation mask could be systematically (and relatively easily) added to the coregistration. Added to the PGO improvement list.**

Fig 7 - NMAD for BM and SGM compared to lidar for Hellstugubreen

I think the authors misunderstood my earlier comment. It appears that there is larger residual horizontal geolocation error in the BM product compared to the SGM product, likely introduced during the independent co-registration process for each. Please check the translation vectors for the dh maps used to produce 7c and 7d - I can't do this because the products involving lidar data are not available (which is fine, but limits reproducibility).

I'm having a hard time understanding how the on-glacier NMAD values in Table 3 for the Hellstugubreen BM DEM (0.12 m, Fig 7c) is lower than the on-glacier NMAD for theHellstugubreen SGM DEM (0.15 m, Fig 7d). Just looking at these difference maps (with identical color ramp), there appears to be more spread in the BM dh product than the SGM dh product (reproduced above for clarity). I'm wondering if there was potentially a mistake in the analysis/plotting code or labeling in the figure? Maybe double-check, and if necessary, modify the statement on linked 364-366:

**We checked the NMADs (because we were also surprised by this result) and can confirm our numbers. This is the illustration that a color scale/representation can be misleading or, rather, not reflect the statistics. Below we zoom in on a very smooth area of Hellstugubreen, away from any crevasse field. These subsets illustrate very similar noise levels in the two correlation algorithms. As these smooth areas occupy most of the glacier, this likely explains why the NMAD are similar.**

[Figure]

| **Zoom on the block matching (BM) elevation change map for Hellstugubreen** | **Zoom on the semi-global matching (SGM) elevation change map for Hellstugubreen** |

492 (Fig 10 caption) - "mass balance estimates" instead of "mass balances"

469 - same

**Corrected twice.**

**New References**
Terratec AS: Rapport for luftbåren laserskanning. Langfjordjøkelen 2018, 14 pp, 2018.
Terratec AS: Rapport for luftbåren laserskanning. Gråsubreen 2019, 14 pp, 2019a
Terratec AS: Rapport for luftbåren laserskanning. Hellstugubreen-Memurubreen 2019, 14 pp, 2019b.